# Find tailored step example for next step: A Targeted Step-wise Retrieval Framework for Guiding LLM Reasoning

## Abstract

Large language models (LLMs) have shown strong performance in mathematical reasoning, supported by approaches such as In-Context Learning (ICL) and Retrieval-Augmented Generation (RAG). However, existing methods often provide entire problems as examples, which is too coarse-grained for multi-step reasoning. Many steps in a retrieved problem may not align with the reasoning trajectory of the target problem, and some may even mislead the inference process. To address this limitation, we propose TSS (Tailored Step Search), a framework that enhances LLM reasoning through targeted step-level retrieval. TSS enables a model to dynamically decide when to retrieve a single logically consistent next step, using the current problem and its intermediate state as the query. The framework consists of two main components. First, we design structured training data and develop a Step Retriever, trained with a contrastive learning strategy to capture the logical flow between consecutive steps. Second, we train a Generator with a two-phase curriculum: it first learns to predict whether retrieval is necessary, and then learns to generate step-by-step reasoning in a structured format. Experiments on four mathematical reasoning datasets, across four backbone LLMs and multiple few-shot settings, demonstrate that TSS substantially improves reasoning accuracy and reliability. Our code is available at https://anonymous.4open.science/r/TSS-D930/.

## 1 Introduction

Large language models (LLMs) have made substantial advances in mathematical reasoning (Hendrycks et al., 2021; Cobbe et al., 2021), largely enabled by Chain-of-Thought (CoT) prompting, which guides models to generate intermediate steps and handle complex problems. Although this paradigm yields strong performance on many benchmarks, inference errors remain common and often propagate through multi-step reasoning. To address these issues, In-Context Learning (ICL) (Dong et al., 2022) has been widely adopted, providing example cases to guide model reasoning. Current ICL approaches include metric-based selection (e.g., perplexity scores (**?**)), retrieval-based ranking (Wu et al., 2022), and iterative refinement methods (**?**) that introduce additional examples during inference. Traditional ICL methods typically retrieve entire documents (Asai et al., 2024) or large text segments (Zhang et al., 2025) based on semantic similarity to the query. However, in mathematical reasoning, this often yields *problem-level* examples that appear similar to the query but diverge in their reasoning steps.

As shown in Figure 1, to solve the target mathematical problem, the LLM uses ICL to retrieve semantically similar problems along with their solutions as examples, which are then used to guide its reasoning process, referred to here as the actual reasoning. The right trajectory represents the reference solution provided by human experts. Under traditional ICL, the retrieval process selects example problems and their solutions based on semantic similarity to the target problem. Although these problems may appear semantically similar, the solution trajectory of the retrieved example is structurally misaligned with the right trajectory of the target problem. For instance, their initial steps follow entirely different approaches. The gold solution begins by deriving the circle equation, whereas the retrieved example starts with a coordinate-based analysis of the overlapping region. Moreover, many steps in the retrieved solution are irrelevant and provide little direct utility. As

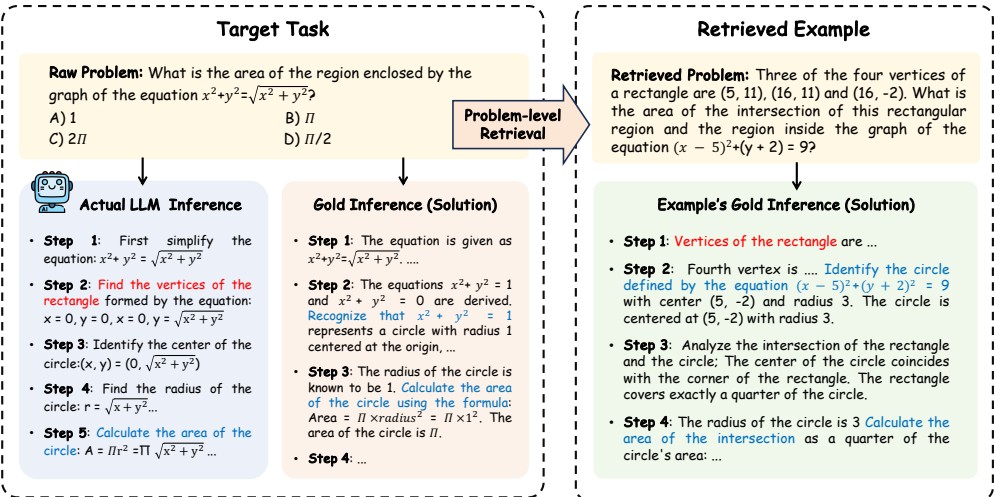

Figure 1: Problem-level ICL examples. Actual LLM Inference shows the actual trajectory of Raw Problem. Gold Inference and Example's Gold Inference show the standard solution of Raw Problem and Retrieved Problem retrieved for Raw Problem respectively. The red part represents reasoning steps that are structurally misaligned and misleading with respect to the target. The Step 1 of Example's Gold Inference is marked in red because the target problem requires deriving a circle equation, not analyzing rectangle vertices. This mismatch misguides the LLM(Step 2 of Actual LLM Inference is the start of error). The blue trajectory represents reasoning steps that are logically aligned and useful for the target problem.

shown in Figure 1, only the second step, deriving the circle equation, and the fourth step, applying the area formula, are directly useful. More critically, the first step, identifying the rectangle's vertices, and the third step, analyzing its positional relationship with the circle, negatively influence the reasoning of the target problem.

To systematically quantify these shortcomings of *problem-based* examples, we introduce two key metrics for assessing the utility of question-level examples. The Same Order Relevance Rate (SRR) measures the proportion of steps that are logically aligned, and the Step Utilization Rate (LUR) measures the proportion of steps that are directly useful. A higher SRR indicates that the retrieved examples follow a reasoning order more consistent with the target problem, while a higher LUR reflects that more steps in the retrieved example directly contribute to solving the target problem. As shown in Table 1, both SRR and LUR remain consistently low (typically below 20%), which means that most retrieved steps are either misaligned or irrelevant. Increasing the number of problem-level examples (e.g., from $k = 1$ to $k = 3$) does not alleviate this misalignment and can even degrade performance by introducing additional irrelevant steps. Consequently, when SRR and LUR are low, error rates remain high (generally above 60%), confirming that poor step alignment and low step utility significantly undermine reasoning reliability.

The limitations of problem-level examples naturally raise a question: how do humans solve similar problems efficiently? Human problem-solvers often benefit from a single relevant hint rather than a complete worked example. This is because human reasoning typically follows a logical chain of *condition-process-conclusion*. When a step is provided, it effectively conveys this logical chain to the mind. People select examples as hints by analyzing the *condition-process-conclusion* structure and then imitate the relevant reasoning process.

Based on the above consideration, we propose *step-level* method to **find tailored step example for next step** , **Tailored Step Search(TSS)**, a novel framework that empowers LLMs with more human-like, process-oriented problem-solving capabilities. First, the model must be able to perform logical reasoning, by linking *condition-process-conclusion* similar to humans. We decompose each solution into a sequence of `[CONDITION, PROCESS, CONCLUSION]` steps to simulating human's *condition-process-conclusion* reasoning. Second, existing semantic encoders should be trained to find logically relevant subsequent steps based on queries generated from the current question and steps, rather than relying solely on semantic similarity matching. So we develop a **Step Retriever** trained use the current step to obtain next step. Finally, the model must learn whether the next step need information retrieval or not during the reasoning process. A **Generator** is trained to learn to

| Model | 1-shot | | | | 2-shot | | | | 3-shot | | | |
|---|---|---|---|---|---|---|---|---|---|---|---|---|
| | (SRR) | (ER-SRR) | (LUR) | (ER-LUR) | (SRR) | (ER-SRR) | (LUR) | (ER-LUR) | (SRR) | (ER-SRR) | (LUR) | (ER-LUR) |
| Llama2$_{7B}$ | 19.51 | 86.49 | 12.65 | 86.59 | 15.81 | 91.14 | 5.16 | 87.91 | 10.50 | 93.10 | 3.82 | 93.55 |
| Llama3$_{8B}$ | 20.17 | 75.68 | 13.42 | 75.31 | 16.84 | 71.43 | 5.59 | 68.13 | 8.86 | 73.56 | 4.28 | 70.65 |
| Qwen2.5$_{3B}$ | 19.92 | 68.92 | 12.51 | 64.63 | 17.38 | 5.25 | 59.21 | 58.24 | 13.14 | 63.86 | 4.36 | 60.64 |
| Qwen2.5$_{7B}$ | 19.64 | 67.57 | 12.90 | 66.25 | 16.41 | 5.25 | 66.23 | 65.93 | 13.32 | 63.86 | 3.93 | 62.77 |

Table 1: The SRR and LUR performance of our four models under k-shot conditions (k=1, 2, 3). ER-SRR and ER-LUR represent the probability of errors occurring when the SRR or LUR falls below a certain threshold (We set at 0.3). More information can be found in Appendix A.6.

predict the need for retrieval and generate *condition-process-conclusion* reasoning processes. Our contributions are:

- We discovered and explored the granularity mismatch problem when the entire problem or the history of the solution or a partial solution is used as an example, and used this as the motivation to propose our framework.

- We introduce **TSS**, a framework that enhances LLM reasoning by mimicking the human cognitive process of seeking targeted, step-wise guidance. We propose a **Step Retriever** trained with a novel contrastive objective that explicitly models the logical progression between reasoning steps and design a **two-stage training curriculum** for the generator. Our framework consistently improves reasoning accuracy and coherence on complex mathematical tasks.

- We contribute a knowledge base and training dataset that can retrieve step-level examples to obtain examples for the next step during inference. Through experiments on four mathematical datasets, four large language model backbones, and multiple few-shot learning settings, we validated the performance improvements of our framework.

## 2 RELATED WORK

### 2.1 MATHEMATICAL REASONING

Mathematical reasoning tasks have long been at the core of artificial intelligence research, and their development process profoundly reflects the evolution of machine intelligence research(Zhang et al., 2019; Liu et al., 2023). In the early stages of research, the field relied mainly on rule systems, template or statistical machine learning methods for simple math word problems (Feigenbaum & Feldman, 1963; Fletcher, 1985). With the rise of large language models (LLMs), people attempt to improve mathematical capability by fine-tuning with more high-quality mathematical data (Shao et al., 2024; Yang et al., 2024; Lewkowycz et al., 2022; Yue et al., 2023; Xu et al., 2024); Many people also focus on using retrieval-augmented generation (Wei et al., 2023) technology to inject relevant examples through prompt engineering. Moreover, many studies employ tree searching strategies like Tree of Thoughts (ToT) (Yao et al., 2023; Besta et al., 2024) or Monte Carlo Tree Search (Zhang et al., 2024; Chen et al., 2024a; Feng et al., 2023; Zhu et al., 2022) to obtain better solution.

### 2.2 IN-CONTEXT LEARNING

In-context Learning (ICL) guides large language models (LLMs) to learn and reason based on given examples (Dong et al., 2022). Existing methods mainly try to solve this problem from different perspectives. Similar-ICL methods (Liu et al., 2021; Luo et al., 2023; Zhang et al., 2023; Fu et al., 2022) aims to screen the examples that are most similar to the semantic representation of the test sample. The commonly used similarity calculation methods include TF-IDF, BM25 (Robertson et al., 2009), T5 encoding (Raffel et al., 2020), BGE-M3 (Chen et al., 2024b). There are also methods to select the context by calculating and evaluating the influence of each example on the prediction results, and the influence calculation methods involve influence functions (Van et al., 2024; Chang & Jia, 2022), perplexity (**?**), mutual information (Sorensen et al., 2022), or even feedback from LLMs (Nguyen & Wong, 2023; **?**). LM3 (Liu et al., 2024) quantifies the impact of example selection on ICL inference performance and identified effective methods for selecting examples.

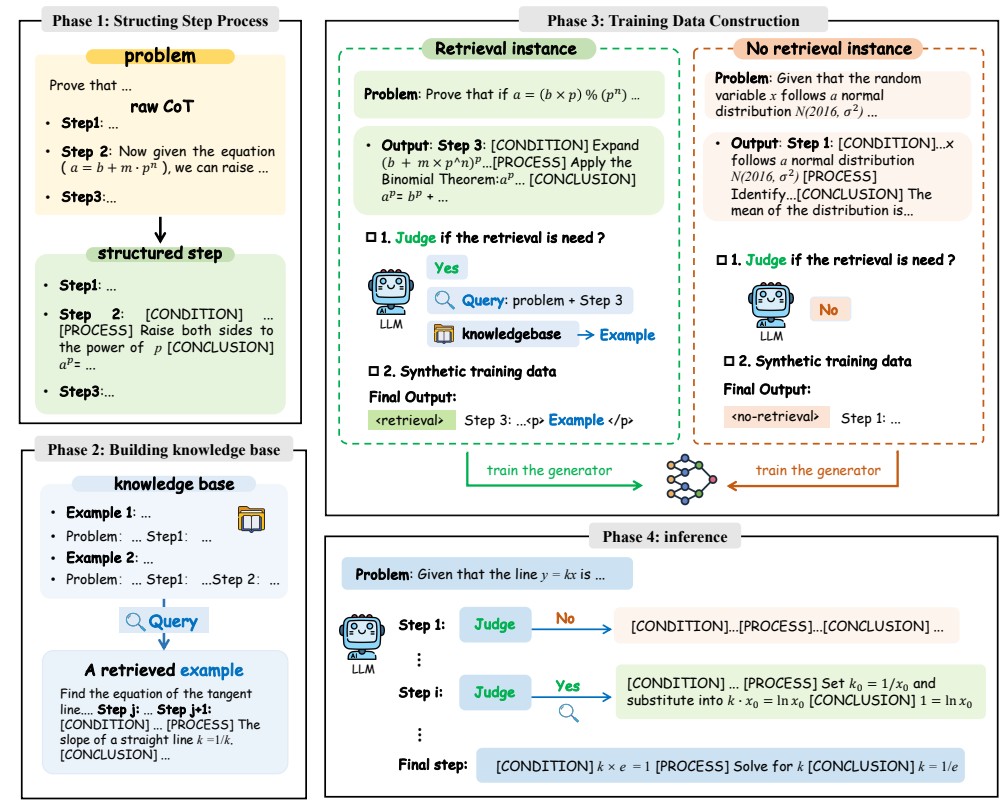

Figure 2: Overview of our framework. The top left panel shows our process for constructing structured step-by-step data, the bottom left panel illustrates an example of building a knowledge base, the top right panel shows how we construct training data, and the bottom right panel demonstrates our reasoning process. The algorithm of Inference can been seen in Appendix D.

## 2.3 RETRIEVAL ARGUMENT GENERATION

Retrieval-Augmented Generation (RAG) expands the input space of the language model by introducing retrieved text fragments. Recent research progress has increasingly focused on enhancing the reasoning capabilities of RAG frameworks. For example, SelfRAG (Asai et al., 2024) and Auto-RAG (Yu et al., 2024) utilize automated data synthesis techniques to enhance the inference process in retrieval-augmented frameworks. AirRAG (Feng et al., 2025) combines Monte Carlo tree search and self-consistency methods. LATS (Zhou et al., 2023) search for relevant information employing LM-generated value scores to generate with tree search.

## 3 PRELIMINARIES

### 3.1 PROBLEM FORMULATION

In mathematical reasoning, a generative model $\mathcal{M}$ typically produces a solution as a sequence of steps $[S_1, \ldots, S_n]$ for a given problem $X$. Our key insight is to structure each step $S_{i-1}$ to encapsulate all necessary historical information. This simplifies the process into a first-order Markovian dependency, where the next step only depends on the previous one:

$$S_i = \arg\max_S P_{\mathcal{M}}\left(S \mid X, S_{i-1}\right). \tag{1}$$

While this simplifies the dependency, the model may still require external guidance to ensure logical progression. To address this, our framework incorporates an on-demand retrieval mechanism. When assistance is needed, the model retrieves a relevant example $E_i = \text{Retriever}(X, S_{i-1}) = (X', S'_{j-1}, S'_j)$ based on the current context. The generation of step $S_i$ is thus conditioned on this

new evidence:

$$S_i = \arg\max_S P_{\mathcal{M}} \left( S \mid X, S_{i-1}, E_i \right). \qquad (2)$$

## 3.2 STRUCTURED STEPS

To structure each step to simulating human's *condition-process-conclusion* reasoning, we do as the following: given the solution sequence $Y = [S_1, S_2, \ldots, S_n]$, each step $S_i$ is a structured instance represented as a triple $(C_i, P_i, Q_i)$, which is $(Condition, Process, Conclusion)$. [*Condition*] is the known facts and previous conclusions at the start of step $i$, [*Process*] is the reasoning process or calculations performed in step $i$, and [*Conclusion*] is the result derived from step $i$. Detailed definition can be seen in Appendix B.1 As shown in the top left corner of Figure 2, the two green rounded rectangles on the left show the process of structuring the steps. For example, we can see the three step of our structured step, the *[CONDITION]* contain the conditions this step need, which are conclusion (i.e., calculation results of Set A and Set B) of above two steps.

## 3.3 KNOWLEDGE BASE CONSTRUCTION AND INDEXING

To accomplish our goal to get a step example, we construct and index a structured knowledge base using the structured steps described above. We select problems from the NumiMath-CoT dataset (LI et al., 2024). The sources of the dataset range from Chinese high school math exercises to US and international mathematics olympiad competition problems,including, which is very suitable as corpus. The process begins by decomposing the solutions from a large corpus of solved problems into our structured step instances $S'$. From each decomposed solution $Y' = [S'_1, S'_2, \ldots, S'_n]$, we then generate a comprehensive set of key-value pairs. Specifically, for each step $S'_j$ (where $j > 0$), we create a key $k'_j = (X', S'_{j-1})$ representing the context, and a corresponding value $v'_j = S'_j$ representing the content to be retrieved as shown in the example in the knowledge base on the right side of the figure. The specific construction method and prompt used are in the Appendix. Finally, for offline indexing, we use our trained retriever model $e(\cdot)$ to pre-compute the dense vector embeddings for all keys in the knowledge base. These vectors are then stored in an efficient approximate nearest neighbor (ANN) index, such as FAISS (Johnson et al., 2019), to facilitate fast retrieval of the most relevant keys for any given query at inference time.

## 4 METHOD

### 4.1 CONTRASTIVE TRAINING DATA CONSTRUCTION

We first decompose problem solutions into structured step instances $(S_1, S_2, \ldots)$. We then construct training data for a contrastive learning objective, inspired by SimCSE (Gao et al., 2021), by generating positive and hard negative pairs for each query. This forces the retriever to learn not just semantic similarity, but the fine-grained distinctions of a valid logical progression. We design one kind of positive pair and four kinds of negative pair. After constructing contrastive training data, we fine-tune a pretrained sentence embedding model, bge-large-en-v1.5 (Xiao et al., 2023), using the InfoNCE (Parulekar

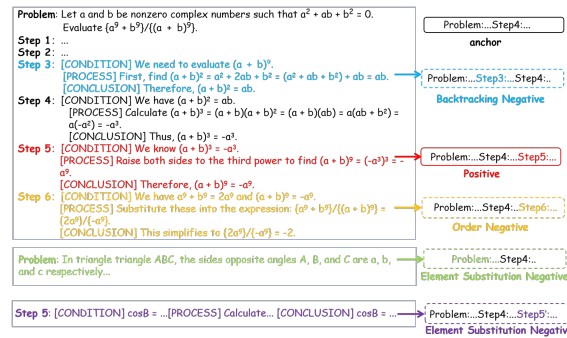

Figure 3: An example of constructing positive and negative examples for retrieval.

et al., 2023) loss function (detailed information is recorded in Appendix A.7).

**Order Negatives** These data pair the query with a distant future step ($S_{i+k}$ where $k > 1$) to enforce logical sequence. This is exemplified by pairing query $(X, S_4)$ with the distant step $S_6$, which disrupts the direct logical flow.

**Positive Pair** For a given query $q_{i-1} = (X, S_{i-1})$, the positive example is constructed using its ground-truth successor step, $S_i$. For instance, given the query $(X, S_4)$ in Figure 3, the positive key-value pair is formed with $S_5$ (highlighted in red).

**Backtracking Negatives** These data pair the query with a preceding step, such as pairing query $(X, S_4)$ with the context of step $S_3$ (blue in Figure 3). This teaches the model to seek forward-progressing, informative steps and understand the direction of reasoning.

**Element Substitution Negatives (replacing the problem stem)** Element Substitution Negatives have two types, replacing the problem stem and replacing the targeted step, respectively. They are created by replacing the the original problem $X$ in positive example $(X, S_{i-1}, S_i)$ with other problem like the content in the green part of the Figure 3, or replacing the target step $S_i$ with counterparts drawn from a different, randomly selected problem $X'$ (as shown in the purple part of the Figure 3.

## 4.2 THE GENERATOR TRAINING

### 4.2.1 TRAINING DATA CONSTRUCTION

We use the same structured step decomposition process described in section 3.2. The training data comes from PRM800K(Lightman et al., 2023). To create training data for our dual-task objective, we used a large language model (GPT-4o-mini) and automatically annotated the dataset with carefully designed prompts. These prompts instruct the model to analyze whether the reasoning from the previous step to the current step in the current problem requires examples for reference or understanding. If the model determines that its reasoning is biased or could benefit from examples, it annotates the corresponding step with a special `<retrieval>` token; otherwise, the step is labeled `<no_retrieval>`. We propose evaluation criteria for both types of labeling. For example, to`<retrieval>` token, we require the model to label the following situations: Non-obvious Strategy or Knowledge(this strategy does not occur in the above inference), Significant Logical Jump,Conceptual Flaw (e.g., distributing a function like f(a+b) = f(a)+f(b)). After labeling the special tokens, we will use the pre-trained retriever and knowledge base to obtain examples for demonstration purposes if the output is `<retrieval>`. The detailed process can be found in the Appendix B.4. This process yields training instances in one of two formats as shown in the top right corner of Figure 2.

Now we get data with `<retrieval>` token and step example: `<retrieval>` $S_i$ `<p>` $E_j$ `</p>` and data with `<no-retrieval>` token: `<no-retrieval>` $S_i$ `<p>` `</p>` . Here, $S_i$ represents the target structured step to be generated. The content for $E_j$ is the retrieved example instance $(X', S'_{j-1}, S'_j)$ from the knowledge base. We use tags `<p>` and `</p>` surrounding the retrieved content. When we retrieve the target step from the knowledge base to construct data, we will use gpt-4o for evaluation. We divide it into five levels: Perfect Analogy, Method Demonstration, Fair, Poor and Irrelevant and each level will give an example to score, the specific prompt words and examples can be seen in the Appendix F.3.2.

### 4.2.2 TWO-PHASE TRAINING FRAMEWORK

Our training regimen employs a two-phase approach designed to decouple retrieval decision-making from reasoning generation.

**Phase 1: Retrieval Prediction Training.** In this phase, the model learns *when* to retrieve. It receives the problem instance $X$ and the current step $S_i$ without any retrieved context. The objective is to predict the retrieval necessity label $\ell_{i+1} \in \{$`<retrieval>`, `<no-retrieval>`$\}$ for the subsequent step. This phase uses masked retrieval conditioning, systematically omitting the example, to force the model to develop intrinsic heuristics for its decisions. The loss is:

$$\mathcal{L}_1 = -\mathbb{E}_{(X, S_i, \ell_{i+1}) \sim \mathcal{D}} \log P(\ell_{i+1} \mid X, S_i) \tag{3}$$

**Phase 2: learns to generate reasoning processes and *condition-process-conclusion* formats.** Here, the model learns to generate reasoning processes using ou data using structure data. It processes the problem $X$, the current step $S_i$. The training objective is to generate the next reasoning step $S_{i+1}$. The loss is:

$$\mathcal{L}_2 = -\mathbb{E}_{(X, S_i, S_{i+1}) \sim \mathcal{D}} \log P(S_{i+1} \mid X, S_i) \tag{4}$$

| Backbone | Method | AIME | GSM8K | MATH500 | STEP | |
|---|---|---|---|---|---|---|
| | | (ACC) | (ACC) | (ACC) | (ACC) | (DSM) |
| Llama-2-7B | Vanilla *w/o retrieval* | 0.79 ± 0.41 | 14.84 ± 0.05 | 2.45 ± 0.12 | 13.51 ± 0.63 | 3.09 ± 0.29 |
| | TSS *w/o retrieval* | 0.99 ± 0.46 | 17.29 ± 0.06 | **5.44 ± 0.11**\* | 14.20 ± 0.69 | 3.51 ± 0.31 |
| | Vanilla | 1.54 ± 0.73 | 15.01 ± 0.03 | 3.79 ± 0.14 | 11.05 ± 0.75 | 4.14 ± 0.22 |
| | SPELL | 0.79 ± 0.34 | 15.16 ± 0.06 | 3.17 ± 0.13 | 7.11 ± 0.50 | 3.79 ± 0.34 |
| | IDS | 2.38 ± 0.69 | 15.99 ± 0.05 | 2.63 ± 0.12 | 9.38 ± 0.63 | 3.24 ± 0.11 |
| | **TSS (Ours)** | **2.53 ± 0.59**\* | **18.20 ± 0.06**\* | 4.97 ± 0.10 | **15.29 ± 0.48**\* | **4.21 ± 0.21**\* |
| Llama-3-8B | Vanilla *w/o retrieval* | 3.23 ± 0.65 | 80.43 ± 0.05 | 27.79 ± 0.12 | 27.87 ± 0.43 | **14.93 ± 0.52**\* |
| | TSS *w/o retrieval* | 4.02 ± 0.64 | 80.59 ± 0.03 | 27.21 ± 0.16 | 26.62 ± 0.65 | 13.27 ± 0.20 |
| | Vanilla | 5.66 ± 0.64 | 78.99 ± 0.05 | 26.19 ± 0.14 | **28.76 ± 0.63**\* | 10.06 ± 0.31 |
| | SPELL | 6.39 ± 0.46 | 82.20 ± 0.06 | 22.60 ± 0.15 | 26.77 ± 0.51 | 8.01 ± 0.32 |
| | IDS | 7.35 ± 0.58 | 80.83 ± 0.05 | **30.37 ± 0.14**\* | 23.64 ± 0.61 | 10.86 ± 0.17 |
| | **TSS (Ours)** | **9.04 ± 0.58**\* | **82.79 ± 0.04**\* | 28.25 ± 0.17 | 28.03 ± 0.85 | 14.63 ± 0.29 |
| Qwen2.5-3B | Vanilla *w/o retrieval* | 6.56 ± 0.31 | 87.20 ± 0.05 | 63.81 ± 0.12 | 35.05 ± 0.82 | 14.92 ± 0.32 |
| | Our Step *w/o retrieval* | 2.51 ± 0.52 | 85.81 ± 0.06 | 64.60 ± 0.15 | 34.06 ± 0.51 | 15.31 ± 0.34 |
| | Vanilla | 4.73 ± 0.55 | 87.33 ± 0.05 | 64.78 ± 0.18 | 36.05 ± 0.75 | **19.59 ± 0.41**\* |
| | SPELL | 4.07 ± 0.56 | 88.55 ± 0.05 | 55.59 ± 0.16 | **43.37 ± 0.85**\* | 18.34 ± 0.55 |
| | IDS | **6.81 ± 0.60**\* | 88.64 ± 0.04 | 65.19 ± 0.11 | 32.89 ± 0.63 | 18.22 ± 0.30 |
| | **TSS (Ours)** | 5.87 ± 0.63 | **89.85 ± 0.06**\* | **66.82 ± 0.16**\* | 39.14 ± 0.93 | 18.89 ± 0.39 |
| Qwen2.5-7B | Vanilla *w/o retrieval* | 13.27 ± 0.44 | 92.72 ± 0.05 | 82.01 ± 0.12 | 35.12 ± 0.54 | 25.50 ± 0.39 |
| | TSS *w/o retrieval* | 12.24 ± 0.63 | **95.45 ± 0.06**\* | 74.17 ± 0.15 | 36.93 ± 0.91 | 26.42 ± 0.32 |
| | Vanilla | 13.17 ± 0.69 | 88.11 ± 0.06 | 77.22 ± 0.12 | 33.15 ± 0.58 | 19.87 ± 0.46 |
| | SPELL | 13.75 ± 0.55 | 94.92 ± 0.05 | 85.38 ± 0.12 | 41.24 ± 0.65 | 15.59 ± 0.25 |
| | IDS | 13.88 ± 0.57 | 93.49 ± 0.06 | 85.78 ± 0.14 | 41.20 ± 0.63 | 17.53 ± 0.39 |
| | **TSS (Ours)** | **16.50 ± 0.51**\* | 95.06 ± 0.05 | **87.35 ± 0.13**\* | **45.42 ± 0.81**\* | **27.64 ± 0.36**\* |

Table 2: Overview of one-shot experiment results. We report accuracy on the AIME, GSM8K, and MATH500 benchmark datasets. On our custom dataset, we report both accuracy and dynamic stride matching (DSM) metrics. Numbers marked with an asterisk (*) indicate the best performance within each model category, and underlined numbers indicate the second-best performance.

## 5 EXPERIMENTS

### 5.1 EXPERIMENTS SETTINGS

We perform evaluations of our **TSS** and various baselines on a range of tasks about math reasoning. We compare the performance of our method on the four datasets: *AIME* (Patel et al., 2024), *GSM8K*(Cobbe et al., 2021), *Math500*(Hendrycks et al., 2021) and *STEP* . *STEP* Dataset is the dataset we construct for testing the models' performance in generating steps. The problem of our dataset comes from PRM800K(Lightman et al., 2023). We choose some of the test dataset in it and decompose the problem solutions into structured steps using a GPT-4o-mini model. The specific construction process and prompts are stored in the Appendix. We also introduce the detail of our dataset in Appendix A.2.

We selected four base models, Qwen2.5-3B-Instruct (Team, 2024), Qwen2.5-Math-7B-Instruct (Yang et al., 2024), Llama-2-7b-chat-hf (Touvron et al., 2023) and Llama-3-8B-Instruct (Dubey et al., 2024). We conducted experiments using zero-shot and standard RAG processes on these four base models. We also performed experiments using methods SPELL (Gonen et al., 2023) IDS (Qin et al., 2024) (we provide a more detailed introduction in A.9), and our TSS method on these four base models. For experiments requiring examples, we used four different settings (1-shot, 2-shot, 3-shot, and 4-shot). We also conducted zero-shot experiments on the four base models after fine-tuning them with our dataset. During inference, we use the instructions in Appendix A.4. The instruction can be seen in Appendix A.3. For the baseline data that needs to be retrieved, we adopt bge-en-1.5v as the retriever. The knowledge base used for retrieval adopts the one we used in Section 3.3. For our TSS,we use our finetuned bge-en-v1.5 as our base embedder . More details about the training settings are given in the Appendix A.1. We define three evaluation metrixs for our task, details and equations can been seen in Appendix A.5

**Dynamic Step Matching (DSM)** To evaluate the intermediate reasoning process, we propose Dynamic Step Matching (DSM). This metric assesses step-wise accuracy while being robust to variations in reasoning length.

**Trajectory Correction Number (TCN)** The Trajectory Correction Number is designed to quantify the number of wrong trajectory directly corrected by our retrieval mechanism. TCN delves into the

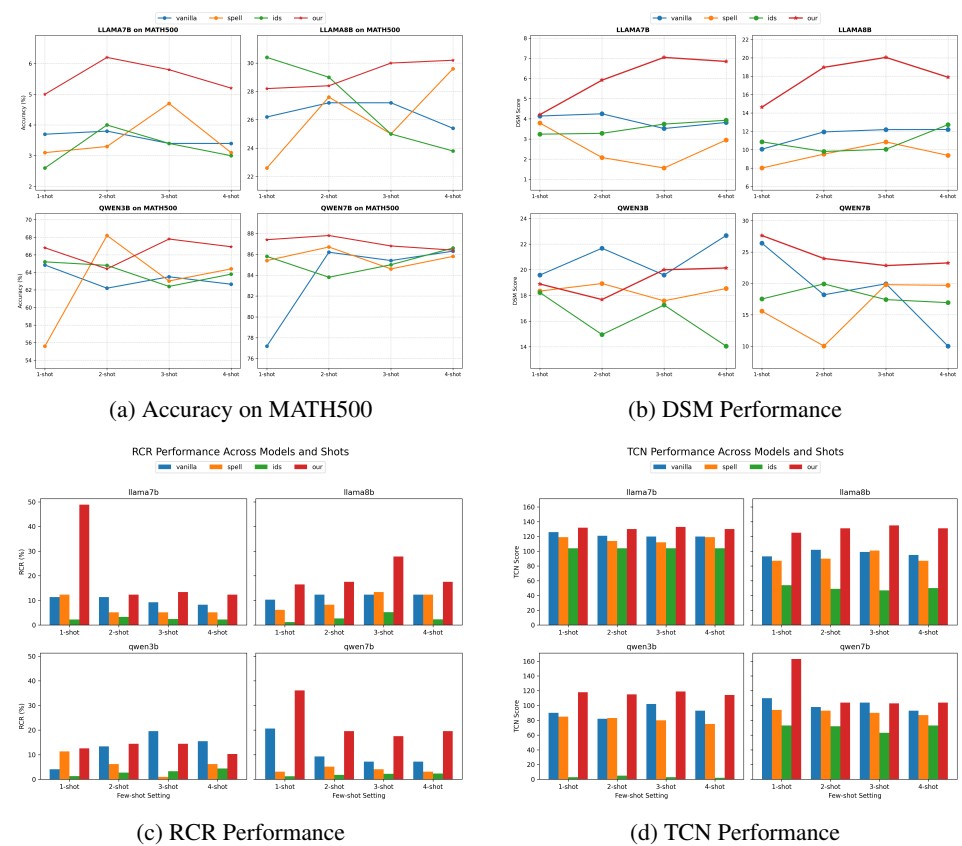

(a) Accuracy on MATH500        (b) DSM Performance

(c) RCR Performance        (d) TCN Performance

Figure 4: Overall performance comparison across four metrics. (a) shows the answer accuracy on MATH500. (b), (c), and (d) show performance on intermediate reasoning metrics: DSM, RCR, and TCN respectively. Answer accuracy on other dataset can be seen in Appendix A.8

model's internal reasoning trajectory to measure its capacity for mid-path self-correction at critical junctures.

**Retrieval-Corrected Rate (RCR)** The Retrieval-Corrected Rate is designed to measure the breadth of problems over which our retrieval mechanism successfully intervenes to correct the reasoning trajectory. While TCN counts the total number of corrected steps, it does not reveal their distribution. RCR complements this by focusing on the proportion of problems that have had at least one trajectory correction.

## 5.2 MAIN RESULTS

Table 2 summarizes the experimental results of our performance metrics (ACC and DSM) using four base models on four datasets under the setting of one-shot.

**Comparison against baselines without retrieval** First, we compare our TSS models against their respective base models without retrieval. The results unequivocally show that integrating our on-demand retrieval mechanism yields substantial performance gains across nearly all settings. For instance, on the AIME (Patel et al., 2024) dataset, Our $TSS_{Llama8B}$ achieves a score of 9.02, 4 more correct answers than the base Llama-3-8B's score of 5.74. Similarly, $TSS_{qwen7B}$ elevates the Math500 (Hendrycks et al., 2021) score to 87.40, while the base model is 82.00. This demonstrates that by providing targeted, step-wise guidance, TSS effectively mitigates the common failure modes of LLMs in reasoning.

**Comparison against baselines with retrieval** We also benchmarked our framework against a standard RAG setup. As shown in Table2, our TSS models consistently and significantly outperform the

standard RAG baselines. Compared with SPELL and IDS, we remain our advantages. On MATH500 (Hendrycks et al., 2021), our TSS score 87.40 while SPELL score 85.40 using Qwen2.5-Math-7B-Instruct(Yang et al., 2024). On GSM8K (Cobbe et al., 2021) and base Llama-3-8B-Instruct(Dubey et al., 2024), our method outperformed the IDS method by 1.97 percentage points. In A critical insight from this comparison is that conventional RAG may degrade the performance of step-by-step reasoning, sometimes even proving detrimental. A stark example is on the AIME dataset with the Qwen2.5-3B (Team, 2024) model, where adding a standard BGE retriever caused performance to plummet from **6.56** to **4.52**. Our TSS did not exhibit such a significant performance drop under most experimental settings. This result strongly supports our central hypothesis: for a model engaged in a delicate logical deduction, retrieving a semantically similar but procedurally irrelevant document may derail the reasoning process.

### 5.3 ANALYSIS

**Ablation studies** To isolate the contributions of our framework's components, we conducted a crucial ablation study by evaluating our specially trained generator without the support of the retriever (`TSS w/o retrieval`). In most cases, its performance is similar to that of the base model without retrieval, and in some cases, the generator alone exhibits inconsistent performance, and may underperforms the original base model (e.g., on AIME with Qwen2.5-3B, dropping from 6.58 to 2.67). However, the moment this generator is paired with our Step Retriever, its performance is dramatically and consistently amplified.

**Robustness Across Diverse Datasets.** Our framework demonstrates strong robustness by delivering consistent gains across datasets with varying characteristics. On AIME and Math500, our method achieves the most dramatic improvements, underscoring its strength in error correction and trajectory guidance. On the Step Dataset, which is specifically designed to evaluate step-wise logical coherence, our models also show the largest performance gains (e.g., TSS improves from 35.05 to 45.36).

**Changes in Accuracy and Dynamic Step Matching under different k-shot settings** The upper part of Figure 4 shows the performance of different methods on MATH500 Hendrycks et al. (2021) (ACC) and STEP (DSM) under different k-shot settings (k=1, 2, 3, 4). We found an interesting phenomenon: the performance is not always better with a larger k. In most cases, the best performance is achieved with 2-shot or 3-shot settings. We believe that when the model receives many examples, the decreasing relevance of later examples with increasing numbers of examples may introduce noise, thus negatively impacting the model's inference performance.

**Changes in Trajectory Correction Numbe and Retrieval-Corrected Rate under different k-shot settings** The lower part of Figure 4 shows the performance of different methods on TCN and RCR metrics under different k-shot settings (k=1, 2, 3, 4). Our TSS method consistently outperforms other methods in improving both the number of correct reasoning paths and the number of correctly solved problems. We also observed that, for these metrics, larger k values do not always lead to better performance; excessive noise can interfere with the model's reasoning. Our TSS method, however, remains the most stable across different k-shot settings.

## 6 CONCLUSION

In this paper, we introduce TSS, a novel framework designed to enhance the multi-step reasoning abilities of large language models (LLMs) by simulating the cognitive process of humans seeking guidance step by step. We focus on how to find suitable examples for the next step, using them as context to boost mathematical capabilities. By reconstructing the mathematical problem-solving process into a structured format and developing specialized logic-step retrievers and a two-stage trained generator, our system learns when to retrieve and how to utilize individual logical subsequent steps as guiding examples. This on-demand, logically aware retrieval mechanism goes beyond traditional RAG, providing the model with fine-grained process supervision that helps improve overall problem-solving accuracy. We candidly acknowledge certain limitations in the current work, such as exploring the application of our ideas without training and extending the tasks to a broader range. These will also serve as directions for our future research focus.

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

# A  EXPERIMENTAL DETAILS

## A.1  MORE DETAILS OF TRAINING

We use 2 Nvidia A800 with 80GB memory to train our models. When training the retriever, we fine-tune all parameters of bge-en-v1.5. We train for 5 epochs, with a batch size of 8 per device, 16 gradient accumulation steps, an initial learning rate of 1e-5, and a warmup ratio of 0.1. When training the generator, all models are fine-tuned using LLaMA-Factory (Zheng et al., 2024). For Llama2-7b-chat-hf, we trained for 4 epochs, with a batch size of 4 per device, 16 gradient accumulation steps, and an initial learning rate of 2e-5; for Llama3-8B-Instruct, we trained for 4 epochs, with a batch size of 2 per device, 32 gradient accumulation steps, and an initial learning rate of 2e-5; for Qwen2.5-3B-Instrcut, we trained for 4 epochs, with a batch size of 8 per device, 16 gradient accumulation steps, and an initial learning rate of 1e-5; for Qwen2.5-7B-Instruct, we trained for 4 epochs, with a batch size of 4 per device, 16 gradient accumulation steps, and an initial learning rate of 2e-5.

## A.2  BRIEF INTRODUCTION OF DATASETS

Here we introduce four datasets we use:

*AIME*(Patel et al., 2024) Dataset includes questions from the American Invitational Mathematics Competition. *AIME* is a renowned high school mathematics competition, famous for its challenging mathematical problems. Here we choose questions from 2020 to 2024. The number of the AIME(Patel et al., 2024) is 122.

*GSM8K*(Cobbe et al., 2021) Dataset is a dataset of 8.5K high quality linguistically diverse grade school math word problems. We collect 1319 questions for test.

*Math500*(Hendrycks et al., 2021) Dataset focused on high school-level competition mathematics with 500 problems.

*STEP* Dataset is the dataset we construct for testing the models' performance in generating steps. The problem of our dataset comes from PRM800K(Lightman et al., 2023). PRM800K is a process supervision dataset containing 800,000 step-level correctness labels for model-generated solutions to problems. We choose some of the test dataset in it and decompose the problem solutions into structured steps using a GPT-4o-mini model. We select 194 problems. Other Information can be seen in Appendix C.1. We contruct the step of the raw CoT with our structure step. The process and prompts of building structure step are stored in the Appendix B.1.

## A.3  INSTRUCTION TUNING TEMPLATE

The figure 5 shows the instruction template we use for instruction fine-tuning, including spe cial tokens and steps for prediction.

Template for generating special token

You are a planner for a math solving agent. Your task is to decide if you need to retrieve an example for the next step. Based on the problem and the previous step, your response MUST be one of two tags and nothing else: `<retrieval>` or `<no-retrieval>`.

Template for generating solution

You are an expert math solver. Your goal is to generate the next step to solve a math problem. When you reach the final answer, include it with \\boxed{answer} format.

Figure 5: The template for instruction fine-tuning.

### A.4 TEMPLATE FOR INFERENCE

> **Instructions**
>
> You are a planner for a math solving agent. Your task is to decide if you need to retrieve an example for the next step. Based on the problem and the previous step, your response MUST be one of two tags and nothing else: '¡retrieval¿' or '¡no retrieval¿'

Table 3: Template for TSS inference to generate special token to decide whether to retrieve.

> **Instructions**
>
> You are an expert math solver. Your goal is to inference the next step to solve a math problem. When you reach the final answer, include it in the CONCLUSION section with
> boxedanswer format. Here are some examples:$\{retrieved examples\}$

Table 4: Template for TSS inference.

> **Instructions**
>
> You are an expert math solver. Your goal is to inference step by step to solve a math problem. When you reach the final answer, include it in the CONCLUSION section with
> boxedanswer format. Here are some examples:$\{retrieved examples\}$

Table 5: Template for baseline inference.

### A.5 OUR EVALUATION METRIC

**DSM**  It introduces a penalty factor that accounts for discrepancies between the number of generated steps and the number of steps in the gold-standard solution. The DSM score is computed as follows:

$$M = \sum_{k=1}^{\min(L_p, L_g)} \mathbb{I}(\text{match}_k), \rho = \sqrt{\frac{\min(L_p, L_g)}{\max(L_p, L_g)}}, \text{DSM} = \frac{M}{L_g} \cdot \rho, \tag{5}$$

Here, $L_p$ is the number of predicted steps and $L_g$ is the number of gold-standard steps. We perform a greedy, one-to-one matching between the predicted and gold steps. $M$ is the total number of correctly matched pairs, where $\mathbb{I}(\text{match}_k)$ is an indicator function that returns 1 if the $k$-th pair of steps are judged as logically equivalent by an LLM. The penalty factor $\rho \in (0, 1]$ scales the score down as the length mismatch between the predicted and gold solutions increases, rewarding solutions that are both correct and concise.

**TCN**  The Trajectory Correction Number is designed to quantify the number of wrong trajectory directly corrected by our retrieval mechanism. TCN delves into the model's internal reasoning

trajectory to measure its capacity for mid-path self-correction at critical junctures. A high TCN value indicates that our framework's interventions are both effective and frequent, proactively repairing potentially failing reasoning paths. We define a correctable step error as an instance where the model generates an erroneous step $s'_{i+1}$ following a correct step $s_i$. A step correction occurs if the model, from the same node $s_i$, generates a logically correct subsequent step $s_{i+1}$. The TCN is the total count of such corrections across the test set. Let $\mathcal{M}_o$ denote the model without retrieval and $\mathcal{M}_w$ denote the model with retrieval. Let $s'_{i+1}$ and $s_{i+1}$ be the subsequent steps generated by $\mathcal{M}_o$ and $\mathcal{M}_w$ respectively. The function $\delta(s)$ is a boolean error indicator, returning true if step $s$ is logically flawed. The TCN is then formulated as:

$$\text{TCN} = \sum_{Q \in \mathcal{D}_{\text{test}}} \sum_{i=1}^{N_Q - 1} \mathbb{I}(\delta(s'_{i+1}) \wedge \neg \delta(s_{i+1})), \tag{6}$$

**RCR** The Retrieval-Corrected Rate is designed to measure the breadth of problems over which our retrieval mechanism successfully intervenes to correct the reasoning trajectory. While TCN counts the total number of corrected steps, it does not reveal their distribution. RCR complements this by focusing on the proportion of problems that have had at least one trajectory correction. A high RCR indicates that the corrective capability of TSS is generalizable and widely applicable across many different problems, rather than being confined to intensive corrections on a few. This metric assesses the breadth of the model's impact on improving the reasoning process itself. We define a corrected problem as any problem in which at least one step in its reasoning trajectory has been successfully corrected by the model. The RCR is formally defined as the proportion of problems, among all those where the model without examples made at least one logical error, that model with examples successfully corrected at least once.

$$\text{RCR} = \frac{|\{Q \in \mathcal{D}_{\text{error}} \mid \exists i, \text{is\_corrected}(Q, i)\}|}{|\mathcal{D}_{\text{error}}|}, \tag{7}$$

where $\mathcal{D}_{\text{error}}$ is the set of all problems in which the baseline model (without retrieval) made at least one logical error during its reasoning process. The function $\text{is\_corrected}(Q, i)$ returns true if a trajectory correction occurred at step $i$ for problem $Q$ (i.e., the baseline erred at step $i + 1$ while TSS did not). The term $\exists i, \text{is\_corrected}(Q, i)$ thus identifies problems with at least one such correction.

### A.6 DETAILS OF COMPUTING SRR AND LUR

SRR calculates the proportion of steps in the ground-truth solution that are ordinally aligned with steps in the example. A higher SRR indicates that the example's structure closely mirrors the reasoning path required for the target problem; LUR measures the proportion of steps within the example that are directly relevant to the reasoning process for the target solution. A high LUR suggests that the example is concise and contains a high density of useful information. The prompts to compute SRR and LUR are reported in Table 22

### A.7 RETRIEVER TRAINING OBJECTIVE

we fine-tune a pretrained sentence embedding model, bge-large-en-v1.5,using the contrastive learning objective (Xiao et al., 2023). Let $\boldsymbol{e}(\cdot)$ denote the embedding function of our retriever, which maps a query or a key to a dense vector. The training objective is to maximize the similarity between a query's embedding and its positive key's embedding, while simultaneously minimizing its similarity to a set of negative keys' embeddings. For a given query $q_i = (X, S_{i-1})$ within a mini-batch, we consider its one positive key $k_i^+$ and a set of $N$ hard negative keys $\{k_j^-\}_{j=1}^N$. The training is guided by InfoNCE loss function:

$$\mathcal{L}_i = -\log \frac{\exp(\text{sim}(\boldsymbol{e}(q_i), \boldsymbol{e}(k_i^+))/\tau)}{\exp(\text{sim}(\boldsymbol{e}(q_i), \boldsymbol{e}(k_i^+))/\tau) + \sum_{j=1}^N \exp(\text{sim}(\boldsymbol{e}(q_i), \boldsymbol{e}(k_j^-))/\tau)}, \tag{8}$$

where $\text{sim}(\cdot, \cdot)$ is the cosine similarity function and $\tau$ is a trainable temperature hyperparameter.

### A.8 ANSWER ACCURACY ON AIME, GSM8K, STEP

We report answer accuracy on AIME, GSM8K, STEP in Figure 6.

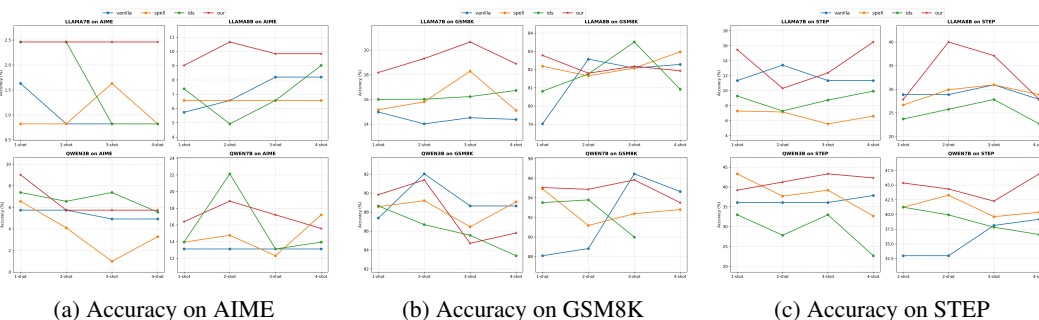

(a) Accuracy on AIME        (b) Accuracy on GSM8K        (c) Accuracy on STEP

Figure 6: Overall performance comparison across three metrics. (a) shows the answer accuracy on AIME. (b) shows the answer accuracy on GSM8K (c) show performance on STEP.

### A.9 SPELL AND IDS

SPELL(Gonen et al., 2023) assumes that prompts with lower perplexity can lead to better performance. It calculates the perplexity of prompt words formed by multiple candidate examples, and then selects the k examples with the lowest perplexity as prompts for reasoning. In contrast, TSS retrieves specific reasoning steps on-demand during inference to guide the model's logic.

IDS(Qin et al., 2024) uses zero-shot chain-of-thought (Zero-shot-CoT) to generate an initial reasoning path for the test question, and based on this, retrieves k semantically similar 'question-answer' pairs from the training set to serve as examples for the next iteration. This process involves iterative reasoning. In contrast, our TSS operates at the step-level granularity, dynamically retrieving only the specific next step required by the current reasoning state, thereby providing more precise and fine-grained supervision, and it also does not require iteration.

## B  DATA FORMULATION AND PROCESSING

**Details of GPT-4o-mini-based data collections**  We use the instruction and demonstration to prompt GPT-4o-mini, listed in Section F. We use the temperature 0.1, $top_p$ to be 0.9 and set the maximum output token counts to be 2048. We discard instances where GPT-4o-mini does not follow the designated output formats.

### B.1  STRUCTURE GENERATION

To convert raw Chain-of-Thought (CoT) solution steps into our structured [CONDITION, PROCESS, CONCLUSION] (CPC) format. For each step $S_i$ in a given solution, we provided the model with the full problem statement $X$, the complete raw CoT solution, and the specific target step $S_i$. The model was instructed to generate the structured CPC triple based on the following definitions:

**[CONDITION]** Extract all necessary information known *before* the current step begins. This must include any relevant numbers or formulas from the original problem statement ($X$) and the explicit conclusions from all preceding steps ($S_1, \ldots, S_{i-1}$).

**[PROCESS]** Describe the core logical or mathematical operation performed in the current step $S_i$. This description should be a high-level summary of the method used (e.g., "Substitute the linear equation into the circle equation," "Apply the Pythagorean theorem"), rather than the detailed calculation itself.

**[CONCLUSION]** State the direct result or outcome of the process, typically a new equation, a simplified expression, or a numerical value.

| Raw Data | **Problem:** Given sets $A = \{x \mid x^2 + 2x - 3 < 0\}$ and $B = \{x \mid |x - 1| \geq 2\}$, find $A \cap B$.

Set A is defined by the inequality $x^2 + 2x - 3 < 0$. We can factor the quadratic expression as $(x + 3)(x - 1) < 0$. The solution to this inequality lies between the roots $x = -3$ and $x = 1$. Thus, set A is $A = \{x \mid -3 < x < 1\}$.
Set B is defined by the absolute value inequality $|x - 1| \geq 2$. This inequality can be split into two separate cases: $x - 1 \geq 2$ or $x - 1 \leq -2$. Solving for $x$ in each case, we get $x \geq 3$ or $x \leq -1$. Thus, set B is $B = \{x \mid x \geq 3 \text{ or } x \leq -1\}$.
We are looking for the overlap between the interval $(-3, 1)$ and the region $(-\infty, -1] \cup [3, \infty)$. The only common region is where $-3 < x$ and $x \leq -1$. The intersection is $A \cap B = \{x \mid -3 < x \leq -1\}$. |
|---|---|
| Structured Data | **Problem:** Given sets $A = \{x \mid x^2 + 2x - 3 < 0\}$ and $B = \{x \mid |x - 1| \geq 2\}$, find $A \cap B$.
**Step 1:**
[CONDITION] Set $A = \{x \mid x^2 + 2x - 3 < 0\}$. [PROCESS] Factor the quadratic: $x^2 + 2x - 3 = (x + 3)(x - 1)$, and find intervals where the product is negative. [CONCLUSION] $A = \{x \mid -3 < x < 1\}$.
**Step 2:**
[CONDITION] Set $B = \{x \mid |x - 1| \geq 2\}$. [PROCESS] Solve the absolute value inequality: $x - 1 \geq 2$ or $x - 1 \leq -2$. [CONCLUSION] $B = \{x \mid x \geq 3 \text{ or } x \leq -1\}$.
**Step 3:**
[CONDITION] $A = \{x \mid -3 < x < 1\}$ and $B = \{x \mid x \geq 3 \text{ or } x \leq -1\}$. [PROCESS] Find the intersection $A \cap B$ by considering the intervals. [CONCLUSION] $A \cap B = \{x \mid -3 < x \leq -1\}$. |

Table 6: List of the example of our structure steps

We adopt the structured step to construct our knowledge base, the retriever training data, and the generator training data. We record he statistic of these data in Appendix C.1.

### B.2 POSITIVE AND NEGATIVE CONSTRUCTING DETAILS

To train our retriever to discern fine-grained logical distinctions, we developed a perturbation strategy to create challenging negatives. For a given positive example, we generate several types of logically flawed but semantically plausible negative examples. Table 7 illustrates our primary perturbation methods. These hard negatives force the retriever to move beyond simple keyword matching and learn the underlying logical validity of a reasoning step.

| Raw Data | **Problem:** Let $a$ and $b$ be nonzero complex numbers such that $a^2 + ab + b^2 = 0$. Evaluate $\frac{a^9+b^9}{(a+b)^9}$. |
|---|---|
| | **Step 1:** [CONDITION] Given the equation $a^2 + ab + b^2 = 0$. [PROCESS] Factor the equation as $(a - b)(a^2 + ab + b^2) = 0$. [CONCLUSION] This implies $a^3 - b^3 = 0$, hence $a^3 = b^3$. **Step 2:** [CONDITION] From $a^3 = b^3$, we have $b^9 = a^9$. [PROCESS] Substitute $b^9$ with $a^9$ in the expression $a^9 + b^9$. [CONCLUSION] Thus, $a^9 + b^9 = a^9 + a^9 = 2a^9$. **Step 3:** [CONDITION] We need to evaluate $(a + b)^9$. [PROCESS] First, find $(a + b)^2 = a^2 + 2ab + b^2 = (a^2 + ab + b^2) + ab = ab$. [CONCLUSION] Therefore, $(a + b)^2 = ab$ **Step 4:** [CONDITION] We have $(a + b)^2 = ab$. [PROCESS] Calculate $(a + b)^3 = (a + b)(a + b)^2 = (a + b)(ab) = a(ab + b^2) = a(-a^2) = -a^3$. [CONCLUSION] Thus, $(a + b)^3 = -a^3$ **Step 5:** [CONDITION] We know $(a + b)^3 = -a^3$. [PROCESS] Raise both sides to the third power to find $(a + b)^9 = (-a^3)^3 = -a^9$. [CONCLUSION] Therefore, $(a + b)^9 = -a^9$. **Step 6:** [CONDITION] We have $a^9 + b^9 = 2a^9$ and $(a+b)^9 = -a^9$. [PROCESS] Substitute these into the expression: $\frac{a^9+b^9}{(a+b)^9} = \frac{2a^9}{-a^9}$. [CONCLUSION] This simplifies to $\frac{2a^9}{-a^9} = -2$. |
| **Input** | **Problem:** Let $a$ and $b$ be nonzero complex numbers such that $a^2 + ab + b^2 = 0$. Evaluate $\frac{a^9+b^9}{(a+b)^9}$. **Step 4:** [CONDITION] We have $(a + b)^2 = ab$. [PROCESS] Calculate $(a + b)^3 = (a + b)(a + b)^2 = (a + b)(ab) = a(ab + b^2) = a(-a^2) = -a^3$. [CONCLUSION] Thus, $(a + b)^3 = -a^3$ |
| **Positive Example (Correct)** | **Step 5:** [CONDITION] We know $(a + b)^3 = -a^3$. [PROCESS] Raise both sides to the third power to find $(a + b)^9 = (-a^3)^3 = -a^9$. [CONCLUSION] Therefore, $(a + b)^9 = -a^9$. |
| **Backtracking Negatives** | **Step 3:** [CONDITION] We need to evaluate $(a + b)^9$. [PROCESS] First, find $(a + b)^2 = a^2 + 2ab + b^2 = (a^2 + ab + b^2) + ab = ab$. [CONCLUSION] Therefore, $(a + b)^2 = ab$ |
| **Order Negatives** | **Step 6:** [CONDITION] We have $a^9 + b^9 = 2a^9$ and $(a+b)^9 = -a^9$. [PROCESS] Substitute these into the expression: $\frac{a^9+b^9}{(a+b)^9} = \frac{2a^9}{-a^9}$. [CONCLUSION] This simplifies to $\frac{2a^9}{-a^9} = -2$. |
| **Element Substitution Negatives (replacing the problem stem)** | Replacing the problem stem with the folowing: In triangle $\triangle ABC$, the sides opposite angles $A$, $B$, and $C$ are $a$, $b$, and $c$ respectively, with $a = \sqrt{7}$, $b = 2$, and $A = \frac{\pi}{4}$. Find $\cos B$. |
| **Element Substitution Negatives (replacing the target step)** | Replacing the target step 5 with following: [CONDITION]] $\cos^2 B = 1 - \left(\frac{\sqrt{14}}{7}\right)^2$. [PROCESS] Calculate $1 - \frac{14}{49}$. [CONCLUSION] $\cos^2 B = \frac{35}{49}$. |

Table 7: Here are the example of how we generate positive and negative samples.

## B.3 KNOWLEDGE BASE TRAINING DATA EXAMPLE.

Here, we list our knowledge base training data example. We have broken down the problem into components as shown in the table 9.

| Data Type | Content |
| --- | --- |
| **Raw Data** | **Problem:** A sphere is cut into six equal wedges. The circumference of the sphere is $18\pi$ inches. After cutting, each wedge's volume is doubled artificially. What is the volume of one of these augmented wedges? Express your answer in terms of $\pi$.
**Step 1:** [CONDITION] The circumference of the sphere is $18\pi$ inches. [PROCESS] Using the formula for circumference $2\pi r = 18\pi$, solve for $r$. [CONCLUSION] $r = 9$ inches.
**Step 2:** [CONDITION] The radius of the sphere is $r = 9$ inches. [PROCESS] Calculate the volume using $V = \frac{4}{3}\pi r^3$. [CONCLUSION] $V = \frac{4}{3}\pi(9^3) = 972\pi$ cubic inches.
**Step 3:** [CONDITION] The volume of the sphere is $972\pi$ cubic inches. [PROCESS] Divide the volume by 6 to find the volume of one wedge: $\frac{972\pi}{6}$. [CONCLUSION] The volume of one wedge is $162\pi$ cubic inches.
**Step 4:** [CONDITION] The volume of one wedge is $162\pi$ cubic inches. [PROCESS] Double the volume of the wedge: $2 \cdot 162\pi$. [CONCLUSION] The volume of one augmented wedge is $324\pi$ cubic inches. |
| **Knowledge base Data** | **Problem:** A sphere is cut into six equal wedges. The circumference of the sphere is $18\pi$ inches. After cutting, each wedge's volume is doubled artificially. What is the volume of one of these augmented wedges? Express your answer in terms of $\pi$.
**Step 1:** [CONDITION] The circumference of the sphere is $18\pi$ inches. [PROCESS] Using the formula for circumference $2\pi r = 18\pi$, solve for $r$. [CONCLUSION] $r = 9$ inches. |
| **Knowledge base Data 1** | **Problem:** A sphere is cut into six equal wedges. The circumference of the sphere is $18\pi$ inches. After cutting, each wedge's volume is doubled artificially. What is the volume of one of these augmented wedges? Express your answer in terms of $\pi$.
**Step 1:** [CONDITION] The circumference of the sphere is $18\pi$ inches. [PROCESS] Using the formula for circumference $2\pi r = 18\pi$, solve for $r$. [CONCLUSION] $r = 9$ inches.
**Step 2:** [CONDITION] The radius of the sphere is $r = 9$ inches. [PROCESS] Calculate the volume using $V = \frac{4}{3}\pi r^3$. [CONCLUSION] $V = \frac{4}{3}\pi(9^3) = 972\pi$ cubic inches. |
| **Knowledge base Data 2** | **Problem:** A sphere is cut into six equal wedges. The circumference of the sphere is $18\pi$ inches. After cutting, each wedge's volume is doubled artificially. What is the volume of one of these augmented wedges? Express your answer in terms of $\pi$.
**Step 1:** [CONDITION] The circumference of the sphere is $18\pi$ inches. [PROCESS] Using the formula for circumference $2\pi r = 18\pi$, solve for $r$. [CONCLUSION] $r = 9$ inches. |

Table 8: – Continued

| Data Type | Content |
|---|---|
| **Knowledge base Data 3** | **Problem:** A sphere is cut into six equal wedges. The circumference of the sphere is $18\pi$ inches. After cutting, each wedge's volume is doubled artificially. What is the volume of one of these augmented wedges? Express your answer in terms of $\pi$.
**Step 1:** [CONDITION] The circumference of the sphere is $18\pi$ inches. [PROCESS] Using the formula for circumference $2\pi r = 18\pi$, solve for $r$. [CONCLUSION] $r = 9$ inches.
**Step 2:** [CONDITION] The radius of the sphere is $r = 9$ inches. [PROCESS] Calculate the volume using $V = \frac{4}{3}\pi r^3$. [CONCLUSION] $V = \frac{4}{3}\pi(9^3) = 972\pi$ cubic inches.
**Step 3:** [CONDITION] The volume of the sphere is $972\pi$ cubic inches. [PROCESS] Divide the volume by 6 to find the volume of one wedge: $\frac{972\pi}{6}$. [CONCLUSION] The volume of one wedge is $162\pi$ cubic inches. |
| **Knowledge base Data 4** | **Problem:** A sphere is cut into six equal wedges. The circumference of the sphere is $18\pi$ inches. After cutting, each wedge's volume is doubled artificially. What is the volume of one of these augmented wedges? Express your answer in terms of $\pi$.
**Step 1:** [CONDITION] The circumference of the sphere is $18\pi$ inches. [PROCESS] Using the formula for circumference $2\pi r = 18\pi$, solve for $r$. [CONCLUSION] $r = 9$ inches.
**Step 2:** [CONDITION] The radius of the sphere is $r = 9$ inches. [PROCESS] Calculate the volume using $V = \frac{4}{3}\pi r^3$. [CONCLUSION] $V = \frac{4}{3}\pi(9^3) = 972\pi$ cubic inches.
**Step 3:** [CONDITION] The volume of the sphere is $972\pi$ cubic inches. [PROCESS] Divide the volume by 6 to find the volume of one wedge: $\frac{972\pi}{6}$. [CONCLUSION] The volume of one wedge is $162\pi$ cubic inches.
**Step 4:** [CONDITION] The volume of one wedge is $162\pi$ cubic inches. [PROCESS] Double the volume of the wedge: $2 \cdot 162\pi$. [CONCLUSION] The volume of one augmented wedge is $324\pi$ cubic inches. |

Table 9: List of the example of our knowledge base steps

## B.4   GENERATOR TRAINING DATA

### B.4.1   DATA WITH SPECIAL TOKEN

| Data Type | Content |
|---|---|
| **Problem** | Given an arithmetic sequence $\{a_n\}$ that satisfies

$$(a_1 + a_2) + (a_2 + a_3) + \cdots + (a_n + a_{n+1}) = 2n(n+1)$$

for $n \in \mathbb{N}^*$.
    1. Find the general formula for the sequence $\{a_n\}$.

    2. Find the sum of the first $n$ terms, $S_n$, of the sequence $\left\{\frac{a_n}{2^{n-1}}\right\}$. |

Table 10: – Continued

| Data Type | Content |
| --- | --- |
| **Step 1** | **Input:** [CONDITION] $(a_1 + a_2) + (a_2 + a_3) + \cdots + (a_n + a_{n+1}) = 2n(n + 1)$. [PROCESS] Denote this as equation (1) and consider $(a_1 + a_2) + (a_2 + a_3) + \cdots + (a_{n-1} + a_n) = 2(n-1)n$ as equation (2). [CONCLUSION] We have equations (1) and (2) for further analysis.
**Gpt-4o-mini Output:** `<retrieval>`
**Final Step 1**
`<retrieval>` [CONDITION] $(a_1 + a_2) + (a_2 + a_3) + \cdots + (a_n + a_{n+1}) = 2n(n + 1)$. [PROCESS] Denote this as equation (1) and consider $(a_1 + a_2) + (a_2 + a_3) + \cdots + (a_{n-1} + a_n) = 2(n - 1)n$ as equation (2). [CONCLUSION] We have equations (1) and (2) for further analysis. |
| **Step 2** | **Input:** [CONDITION] Equations (1) and (2). [PROCESS] Subtract equation (2) from equation (1). [CONCLUSION] $a_n + a_{n+1} = 4n$, denoted as equation (3).
**Gpt-4o-mini Output:** `<retrieval>`
**Final Step 2**
`<retrieval>`: [CONDITION] Equations (1) and (2). [PROCESS] Subtract equation (2) from equation (1). [CONCLUSION] $a_n + a_{n+1} = 4n$, denoted as equation (3). |
| **Step 3** | **Input:** [CONDITION] Equation (3). [PROCESS] Set $n = n - 1$ to obtain $a_{n-1} + a_n = 4(n - 1)$, denoted as equation (4). [CONCLUSION] We have equation (4) for further analysis.
**Gpt-4o-mini Output:** `<no-retrieval>`
**Final Step 3**
`<no-retrieval>`: [CONDITION] Equation (3). [PROCESS] Set $n = n - 1$ to obtain $a_{n-1} + a_n = 4(n - 1)$, denoted as equation (4). [CONCLUSION] We have equation (4) for further analysis. |
| **Step 4** | **Input:** [CONDITION] Equations (3) and (4). [PROCESS] Subtract equation (4) from equation (3). [CONCLUSION] $2d = 4$, thus $d = 2$.
**Gpt-4o-mini Output:** `<retrieval>`
**Final Step 4**
`<retrieval>`: [CONDITION] Equations (3) and (4). [PROCESS] Subtract equation (4) from equation (3). [CONCLUSION] $2d = 4$, thus $d = 2$. |
| **Step 5** | **Input:** [CONDITION] $d = 2$ and $a_1 + a_2 = 4$. [PROCESS]] Solve for $a_1$ using $a_1 + (a_1 + d) = 4$. [CONCLUSION] $a_1 = 1$.
**Gpt-4o-mini Output:** `<retrieval>`
**Final Step 5**
`<retrieval>`: [CONDITION] $d = 2$ and $a_1 + a_2 = 4$. [PROCESS]] Solve for $a_1$ using $a_1 + (a_1 + d) = 4$. [CONCLUSION] $a_1 = 1$. |
| **Step 6** | **Input:**
[CONDITION] $a_1 = 1$ and $d = 2$. [PROCESS] Write the general formula for the sequence. [CONCLUSION] $a_n = 1 + 2(n - 1) = 2n - 1$.
**Gpt-4o-mini Output:** `<no-retrieval>`
**Final Step 6**
`<no-retrieval>`: [CONDITION] $a_1 = 1$ and $d = 2$. [PROCESS] Write the general formula for the sequence. [CONCLUSION] $a_n = 1 + 2(n - 1) = 2n - 1$. |

Table 10: – Continued

| Data Type | Content |
|---|---|
| **Step 7** | **Input:**
[CONDITION] $a_n = 2n - 1$. [PROCESS] Express $\frac{a_n}{2^{n-1}} = (2n - 1) \cdot \left(\frac{1}{2}\right)^{n-1}$. [CONCLUSION] We have the expression for the terms of the sequence in part (2).
**Gpt-4o-mini Output:** `<no-retrieval>`
**Final Step 7**
`<no-retrieval>` [CONDITION] $a_n = 2n - 1$. [PROCESS] Express $\frac{a_n}{2^{n-1}} = (2n - 1) \cdot \left(\frac{1}{2}\right)^{n-1}$. [CONCLUSION] We have the expression for the terms of the sequence in part (2). |
| **Step 8** | **Input:**
[CONDITION] $S_n = 1 \cdot \left(\frac{1}{2}\right)^0 + 3 \cdot \left(\frac{1}{2}\right)^1 + 5 \cdot \left(\frac{1}{2}\right)^2 + \cdots + (2n - 1) \cdot \left(\frac{1}{2}\right)^{n-1}$. [PROCESS] Write the expression for $S_n$. [CONCLUSION] $S_n$ is expressed as a series involving powers of $\frac{1}{2}$.
**Gpt-4o-mini Output:** `<no-retrieval>`
**Final Step 8**
`<no-retrieval>`: [CONDITION] $S_n = 1 \cdot \left(\frac{1}{2}\right)^0 + 3 \cdot \left(\frac{1}{2}\right)^1 + 5 \cdot \left(\frac{1}{2}\right)^2 + \cdots + (2n-1) \cdot \left(\frac{1}{2}\right)^{n-1}$. [PROCESS] Write the expression for $S_n$. [CONCLUSION] $S_n$ is expressed as a series involving powers of $\frac{1}{2}$. |
| **Step 9** | **Input:**
[CONDITION] $S_n$ expression. [PROCESS] Multiply $S_n$ by $\frac{1}{2}$ and shift the series. [CONCLUSION] $\frac{1}{2}S_n = 1 \cdot \left(\frac{1}{2}\right)^1 + 3 \cdot \left(\frac{1}{2}\right)^2 + 5 \cdot \left(\frac{1}{2}\right)^3 + \cdots + (2n - 1) \cdot \left(\frac{1}{2}\right)^n$.
**Gpt-4o-mini Output:** `<no-retrieval>`
**Final Step 9**
`<no-retrieval>`: [CONDITION] $S_n$ expression. [PROCESS] Multiply $S_n$ by $\frac{1}{2}$ and shift the series. [CONCLUSION] $\frac{1}{2}S_n = 1 \cdot \left(\frac{1}{2}\right)^1 + 3 \cdot \left(\frac{1}{2}\right)^2 + 5 \cdot \left(\frac{1}{2}\right)^3 + \cdots + (2n - 1) \cdot \left(\frac{1}{2}\right)^n$. |
| **Step 10** | **Input:**
[CONDITION] $\frac{1}{2}S_n$ expression. [PROCESS] Rearrange and simplify the series. [CONCLUSION] $\frac{1}{2}S_n = 3 - (2n + 3) \cdot \left(\frac{1}{2}\right)^n$.
**Gpt-4o-mini Output:** `<retrieval>`
**Final Step 10**
`<retrieval>`: [CONDITION] $\frac{1}{2}S_n$ expression. [PROCESS] Rearrange and simplify the series. [CONCLUSION] $\frac{1}{2}S_n = 3 - (2n + 3) \cdot \left(\frac{1}{2}\right)^n$. |
| **Step 11** | **Input:** [CONDITION] $\frac{1}{2}S_n = 3 - (2n + 3) \cdot \left(\frac{1}{2}\right)^n$. [PROCESS] Multiply both sides by 2 to solve for $S_n$. [CONCLUSION] $S_n = 6 - (2n + 3) \cdot \left(\frac{1}{2}\right)^{n-1}$.
**Gpt-4o-mini Output:** `<no-retrieval>`
**Final Step 11**
`<no-retrieval>`: [CONDITION] $\frac{1}{2}S_n = 3 - (2n + 3) \cdot \left(\frac{1}{2}\right)^n$. [PROCESS] Multiply both sides by 2 to solve for $S_n$. [CONCLUSION] $S_n = 6 - (2n + 3) \cdot \left(\frac{1}{2}\right)^{n-1}$. |

Table 11: List of the example of our special tokens

### B.4.2 DATA WITH RETRIEVED DATA

| Data Type | Content |
|---|---|
| **Problem** | Given an arithmetic sequence $\{a_n\}$ that satisfies $$(a_1 + a_2) + (a_2 + a_3) + \cdots + (a_n + a_{n+1}) = 2n(n + 1)$$ for $n \in \mathbb{N}^*$. 
 1. Find the general formula for the sequence $\{a_n\}$. 
 2. Find the sum of the first $n$ terms, $S_n$, of the sequence $\left\{\frac{a_n}{2^{n-1}}\right\}$. |
| **Step** | **Input:** `<retrieval>:` `[CONDITION]` $\frac{1}{2}S_n$ expression. `[PROCESS]` Rearrange and simplify the series. `[CONCLUSION]` $\frac{1}{2}S_n = 3 - (2n + 3) \cdot \left(\frac{1}{2}\right)^n$. |
| **Retrived Example** | **Problem**: Given the sequence $\{a_n\}$, where $a_1 = 4$, and $a_n = a_{n-1} + 2^{n-1} + 3$ for $n \geqslant 2, n \in \mathbb{N}^*$. 
 1. Prove that the sequence $\{a_n - 2^n\}$ is an arithmetic sequence and find the general term formula for $\{a_n\}$. 
 2. Let $b_n = \frac{a_n}{2^n}$, find the sum of the first $n$ terms, denoted as $S_n$. |
| **Final Step** | **Step 9** `[CONDITION]` $\frac{1}{2}T_n$ established. `[PROCESS]` Simplify the right-hand side. `[CONCLUSION]` $\frac{1}{2}T_n = \frac{5}{2} - \frac{3n+5}{2^{n+1}}$. 
 **Step 10** `[CONDITION]]` $T_n$ derived from $\frac{1}{2}T_n$. `[PROCESS]` Multiply by 2. `[CONCLUSION]` $T_n = 5 - \frac{3n+5}{2^n}$. 
 `<retrieval>` `[CONDITION]` $(a_1 + a_2) + (a_2 + a_3) + \cdots + (a_n + a_{n+1}) = 2n(n + 1)$. `[PROCESS]` Denote this as equation (1) and consider $(a_1 + a_2) + (a_2 + a_3) + \cdots + (a_{n-1} + a_n) = 2(n - 1)n$ as equation (2). `[CONCLUSION]` We have equations (1) and (2) for further analysis. |

Table 13: List of the example of our full generator training data

## C DATASET DETAILS

### C.1 DATASET STATISTICS

| Data | Number | Retrieval Token Number | Step Number | Average Step of a Problem |
|---|---|---|---|---|
| Retrieve Training Data | 35832 | – | – | – |
| Generator Training Data | 10467 | 28038 | 54086 | 5.17 |
| Knowledge Base Data | 125402 | – | – | – |
| STEP Dataset | 194 | – | 962 | 4.96 |

Table 14: Details of Our Constructed Data.

### C.2 EVALUATION FOR GENERATOR TRAINING DATA

We also recode the cause of retrieval in Table 15. We recode four reason deciding whether to give a special retrieval token to the steo: Significant Logical Jump, Non-obvious Theorem/Formula, Clever Trick/Strategy and Conceptual Calculation Error. We report the proportion of them.

| Cause | proportion (%) |
|---|---|
| Significant Logical Jump | 74.90 |
| Non-obvious Theorem/Formula | 21.27 |
| Clever Trick/Strategy | 2.83 |
| Conceptual Calculation Error | 1.00 |

Table 15: Cause of retrieval and its proportion.

## D INFERENCE

During inference, the TSS framework operates in an auto-regressive loop. For each step $i$, the generator first takes the problem $X$ and the previously generated step $S_{i-1}$ as input. Its initial task is to predict only the decision token for the upcoming step, $\ell_i \in \{$<retrieval>, <no-retrieval>$\}$. If the decision is <retrieval>, the retriever is activated. A query is formed from the problem stem and current step, $q_i = (X, S_{i-1})$, and sent to our pre-built FAISS index. The retriever fetches the top-k most relevant example, $E_i = (X', S'_{j-1}, S'_j)$. If the decision is <no-retrieval>, this example $E_i$ is set to null. Finally, the generator produces the structured reasoning step $S_i$. It conditions its generation on the context: the problem $X$, the previous step $S_{i-1}$, and the (potentially null) retrieved example $E_i$. This loop continues, with each newly generated step $S_i$ serving as the context for the next decision $\ell_{i+1}$, until the model generates a final answer or a termination token.

---

**Algorithm 1** The Simplified TSS Inference Algorithm

---

**Require:** Problem instance $X$, Maximum number of steps $N_{\max}$
**Ensure:** Final solution $Y = [S_1, \ldots, S_T]$
1: Initialize $Y \leftarrow \emptyset$, $S_0 \leftarrow$ null
2: **for** $i = 1$ to $N_{\max}$ **do**
3:     **if** ShouldRetrieve$(\mathcal{M}_{\text{gen}}, X, S_{i-1})$ **then**                    ▷ Decide and retrieve
4:         $E_i \leftarrow \mathcal{R}_{\text{ret}}(X, S_{i-1})$
5:     **else**
6:         $E_i \leftarrow$ null
7:     **end if**
8:     $S_i \leftarrow \mathcal{M}_{\text{gen}}(X, S_{i-1}, E_i)$                    ▷ Generate next step
9:     Append $S_i$ to $Y$
10:     **if** IsTermination$(S_i)$ **then break**
11:     **end if**
12: **end for**
13: **return** $Y$

---

## E CASE STUDIES

Table 16: Our Model Inference Process

| Data Type | Content |
|---|---|
| **Problem** | Given that the line $y = kx$ is a tangent to the function $y = \ln x$, find the value of $k$. |
| **Gold Answer** | $\dfrac{1}{e}$ |

*Continued on next page*

Table 16: Our Model Inference Process (– Continued)

| Inference | Process |
|---|---|
| **Decide whether to retrieve** | **Output:** retrieval
**Retrieval Query:**
Given that the line $y = kx$ is a tangent to the function $y = \ln x$, find the value of $k$.
**Retrieved Example:**
Problem: If the line $y = kx + b$ is a tangent line to the curve $y = \ln x + 2$ and also a tangent line to the curve $y = \ln(x+1)$, find the value of $b$.
Step 1: [CONDITION] Tangent points are $(x_1, kx_1 + b)$ for $y = \ln x + 2$ and $(x_2, kx_2 + b)$ for $y = \ln(x+1)$ [PROCESS] Set up the relationship between the tangent points: $k = \frac{1}{x_1} = \frac{1}{x_2+1}$ [CONCLUSION] $x_1 = x_2 + 1$ |
| **Inference** | **Output:**
Step 1: [CONDITION] Line $y = kx$ is tangent to $y = \ln x$ [PROCESS] Set $kx = \ln x$ for tangency [CONCLUSION] $kx = \ln x$ |
| **Decide whether to retrieve** | **Output:** no-retrieval
**Retrieval Query:** None
**Retrieved Example:** None |
| **Inference** | **Output:**
Step 2: [CONDITION] $kx = \ln x$ [PROCESS] Differentiate $y = \ln x$ to find slope: $\frac{dy}{dx} = \frac{1}{x}$ [CONCLUSION] Slope of tangent at point of tangency is $\frac{1}{x}$ |
| **Decide whether to retrieve** | **Output:** retrieval
**Retrieved Query:**
Given that the line $y = kx$ is a tangent to the function $y = \ln x$, find the value of $k$.
Step 2: [CONDITION] $kx = \ln x$ [PROCESS] Differentiate $y = \ln x$ to find slope: $\frac{dy}{dx} = \frac{1}{x}$ [CONCLUSION] Slope of tangent at point of tangency is $\frac{1}{x}$
**Retrieved Example:**
Problem: Find the equation of the tangent line to the function $y = \frac{\ln x}{x} + x$ at the point $(1, 1)$.
Step 2: [CONDITION] $f'(x) = \frac{1-\ln x}{x^2} + 1$ [PROCESS] Calculate $f'(1)$: $f'(1) = \frac{1-\ln 1}{1^2} + 1$ [CONCLUSION] $f'(1) = 2$
Step 3: [CONDITION] Slope of tangent line $k = 2$, point $(1, 1)$ [PROCESS] Use point-slope form: $y - 1 = 2(x - 1)$ [CONCLUSION] $y - 1 = 2x - 2$ |
| **Inference** | **Output:**
Step 3: [CONDITION] Slope of tangent is $\frac{1}{x}$ [PROCESS] Set $k = \frac{1}{x}$ and substitute into $kx = \ln x$ [CONCLUSION] $1 = \ln x$ |
| **Decide whether to retrieve** | **Output:** no-retrieval
**Retrieved Query:** None
**Retrieved Example:** None |
| **Inference** | **Output**:
Step 4: [CONDITION] $1 = \ln x$ [PROCESS] Solve for $x$ [CONCLUSION] $x = e$ |
| **Decide whether to retrieve** | **Output:** no-retrieval
**Retrieved Query:** None
**Retrieved Example:** None |

*Continued on next page*

Table 16: Our Model Inference Process (– Continued)

| Inference | Process |
|---|---|
| **Inference** | **Output**:
Step 5: [CONDITION] $x = e$ [PROCESS] Substitute $x$ back into the equation $kx = \ln x$ [CONCLUSION] $ke = \ln e = 1$ |
| **Decide whether to retrieve** | **Output:** no-retrieval
**Retrieved Query:** None
**Retrieved Example:** None |
| **Inference** | **Output**:
Step 6: [CONDITION] $ke = 1$ [PROCESS] Solve for $k$ [CONCLUSION] $k = \frac{1}{e}$ |

### E.1 SUCCESSFUL RETRIEVAL CASE

Comparison of different RAG approaches: (a)Search for examples based on the question before making inferences. This method with the example does not solve the error in the second step of the model that the derivative of any horizontal coordinate is calculated as the derivative of a specific point. (b)Search the question stem and historical steps according to the question stem and the steps you have tried during the reasoning process. A long example of the question and the historical steps of the question was still retrieved which include the equation of the intersection of the curves but can only be used to correct the step that has been made because it is not necessarily related to the next step. The reasoning in the next step is wrong because the model does not know what to do after listing the equation of the intersection of the two curves. (c)Searching for the next step based on the question stem and the current step can be used as an example step in the reasoning process, guiding the model to make its own subsequent reasoning. After obtaining the derivative in step 2, we use the information from the problem stem and step 2 to obtain an example for step 3 (composing the tangent equation for a specific point $x_0$) (composing the tangent equation for the point $(1, 1)$) and guide model to generate the right inference.

## F PROMPTS

### F.1 PROMPT FOR SRR AND LUR

---

**Instructions**

**same_order_relevance**
> System Prompt: You are an expert comparing step-by-step solutions. Compare 'Target Steps' and 'Example Steps' pairwise. Return a JSON object: `{"num_common_steps": T, "num_related_steps": R}` where T is the number of pairs compared (length of the shorter list) and R is the count of logically related pairs.

**logic_utilization**
> System Prompt: You are an expert comparing step-by-step solutions. Determine how many 'Target Steps' are logically covered by *any* of the 'Example Steps'. Return a JSON object: `{"num_target_steps": T, "num_utilized_steps": U}` where T is the total number of target steps and U is the count of target steps with a logical equivalent in the example.

---

Table 17: Instructions and demonstrations for evaluating SRR and LUR.

1404
1405
1406
1407
1408
1409
1410
1411
1412
1413
1414
1415
1416
1417
1418
1419
1420
1421
1422
1423
1424
1425
1426
1427
1428
1429
1430
1431
1432
1433
1434
1435
1436
1437
1438
1439
1440
1441
1442
1443
1444
1445
1446
1447
1448
1449
1450
1451
1452
1453
1454
1455
1456
1457

## F.2 PROMPT FOR CONSTRUCTING STRUCTURED STEPS

---

**Instructions**

---

You are an expert in mathematical reasoning and pedagogy. Your task is to analyze the reasoning process from a "Raw Data" to create a corresponding "Knowledge base Data" entry. The knowledge base data is structured as a key-value pair, where the key is the context (problem + previous steps) and the value is the target next step.

---

**Part 1: Instructions**

1. Analyze the provided "Raw Data", which contains a complete problem and its step-by-step solution.

2. For **each step** in the solution (from Step 2 onwards), create one key-value pair for the knowledge base.

3. The **Key** MUST contain the full problem description AND all steps preceding the target step.

4. The **Value** MUST be the single, target step that immediately follows the context provided in the key.

---

**Part 2: Example**

**Raw Data: Problem:** A sphere is cut into six equal wedges. The circumference of the sphere is $18\pi$ inches. After cutting, each wedge's volume is doubled artificially. What is the volume of one of these augmented wedges? Express your answer in terms of $\pi$.
**Step 1:** [CONDITION] The circumference of the sphere is $18\pi$ inches. [PROCESS] Using the formula for circumference $2\pi r = 18\pi$, solve for $r$. [CONCLUSION] $r = 9$ inches.
**Step 2:** [CONDITION] The radius of the sphere is $r = 9$ inches. [PROCESS] Calculate the volume using $V = \frac{4}{3}\pi r^3$. [CONCLUSION] $V = 972\pi$ cubic inches.
**Step 3:** [CONDITION] The volume of the sphere is $972\pi$ cubic inches. [PROCESS] Divide the volume by 6 to find the volume of one wedge: $\frac{972\pi}{6}$. [CONCLUSION] The volume of one wedge is $162\pi$.
**Step 4:** [CONDITION] The volume of one wedge is $162\pi$ cubic inches. [PROCESS] Double the volume of the wedge: $2 \cdot 162\pi$. [CONCLUSION] The volume of one augmented wedge is $324\pi$.

**Knowledge base Data: Key:**
**Problem:** A sphere is cut into six equal wedges. The circumference of the sphere is $18\pi$ inches. After cutting, each wedge's volume is doubled artificially. What is the volume of one of these augmented wedges? Express your answer in terms of $\pi$.
**Step 1:** [CONDITION] The circumference of the sphere is $18\pi$ inches. [PROCESS] Using the formula for circumference $2\pi r = 18\pi$, solve for $r$. [CONCLUSION] $r = 9$ inches.
**Step 2:** [CONDITION] The radius of the sphere is $r = 9$ inches. [PROCESS] Calculate the volume using $V = \frac{4}{3}\pi r^3$. [CONCLUSION] $V = 972\pi$ cubic inches.

**Value:**
**Step 3:** [CONDITION] The volume of the sphere is $972\pi$ cubic inches. [PROCESS] Divide the volume by 6 to find the volume of one wedge: $\frac{972\pi}{6}$. [CONCLUSION] The volume of one wedge is $162\pi$.

---

> **Part 3: Your Task** Based on the instructions and the example, process the "Raw Data" below to generate the corresponding "Knowledge base Data".
> **Raw Data:**
> {*raw_data_content*}
> **Knowledge base Data:**
> {*knowledge_base_data_output*}

Table 18: Instructions and demonstrations for constructing structure data.

## F.3 PROMPT FOR CONSTRUCTING GENERATOR TRAINING DATA

### F.3.1 SPECIAL TOKEN

> **Instructions**
>
> You are an expert in mathematical reasoning and pedagogy. Your task is to analyze the reasoning leap from a `previous_step` to a `current_step` within the context of a given `problem`. Based on this analysis, you will decide if a student would need to retrieve a guiding example to understand or perform this specific step.
>
> ---
>
> **Part 1: Your Evaluation Criteria** Your entire judgment is based on the transition from the previous step (or the problem description) to the current step.
> - You should recommend retrieval if the transition to the current step represents a significant intellectual challenge or a common pitfall. Examples include:
>
>   1. **Non-obvious Strategy or Knowledge:** The step requires a theorem, formula, or problem-solving trick that is not immediately apparent from the preceding context.
>   2. **Significant Logical Jump:** The step represents a major leap in reasoning that combines multiple smaller, unstated steps.
>   3. **Conceptual Flaw / Error-Prone Step:** The step involves a process where a conceptual misunderstanding is likely or has occurred. This is a form of **flawed logic**. Examples:
>      – Using an incorrect formula for the situation.
>      – Misapplying a complex procedure (e.g., integration by parts).
>      – Incorrectly using a mathematical property (e.g., distributing a function like $f(a + b) = f(a) + f(b)$).
>
> - You should recommend no-retrieval if the current step is a direct and simple consequence of the previous step, such as:
>
>   1. A straightforward calculation or algebraic manipulation.
>   2. The direct application of a very common and obvious formula.
>   3. A simple arithmetic slip (e.g., $2 + 3 = 6$) or a transcription error. These are considered minor execution errors, not conceptual gaps requiring retrieval.
>
> ---
>
> **Part 2: Your Output Format** Your response MUST be a JSON object with two keys: "tag" and "category".
>
>   1. "tag": Your decision from Part 1 retrieval or no-retrieval .
>
>   2. "category":
>
>      - If "tag" is no-retrieval , the value for "category" MUST be "N/A".
>      - If "tag" is retrieval , you MUST choose one of the following categories that best describes your reasoning:

"Non-obvious Theorem/Formula"
"Clever Trick/Strategy"
"Significant Logical Jump"
"Conceptual Calculation Error"
"Other"

Table 19: Instructions and demonstrations for generator training data.

### F.3.2 EVALUATION

**Instructions**

You are an expert in mathematical reasoning. Your task is to evaluate if a "Retrieved Step" provides a logically sound example for deriving a "Gold Step".
Both steps are self-contained with `[CONDITION]`, `[PROCESS]`, and `[CONCLUSION]`. Your evaluation must focus on the similarity of the logical leap in the `[PROCESS]`, given the context in `[CONDITION]`.

---

**Part 1: Scoring Rubric**

**5 (Perfect Analogy):** The Retrieved Step solves a problem of the *exact same type* using the *exact same method*. A student can directly map their problem onto the example by just changing numbers.

**4 (Method Demonstration):** The Retrieved Step demonstrates the *core method or formula* required, but the *problem setup or context is different*. The student needs an insight to see how the method applies to their specific problem.

**3 (Fair):** The Retrieved Step demonstrates a partially related concept or a sub-step, but not the main logical leap.

**2 (Poor):** The Retrieved Step is on the same general topic but uses an inapplicable or incorrect method for the goal. It's more likely to confuse than help.

**1 (Irrelevant):** The logic and mathematical domain are completely unrelated.

---

**Part 2: Examples for Calibration**

EXAMPLE FOR SCORE 5 (PERFECT ANALOGY)

***Gold Step:*** "`[CONDITION]` There are 10 red balls and 10 blue balls in a bag. We draw two balls without replacement. `[PROCESS]` The probability of the first being red is $10/20$. The probability of the second being red is then $9/19$. The total probability is $(10/20) \times (9/19)$. `[CONCLUSION]` The probability is $9/38$."

***Retrieved Step:*** "`[CONDITION]` A batch has 50 items, 5 of which are defective. We pick two items without replacement. `[PROCESS]` The probability of the first being defective is $5/50$. The probability of the second being defective is $4/49$. The total probability is $(5/50) \times (4/49)$. `[CONCLUSION]` The probability is $2/245$."

***Reasoning:*** Both problems are of the exact same type (conditional probability without replacement). Perfect 5.

EXAMPLE FOR SCORE 4 (METHOD DEMONSTRATION)

***Gold Step:*** "`[CONDITION]` The radius of a circle is 5 and the distance from the center to a chord is 3. This forms a right triangle with the radius as the hypotenuse. `[PROCESS]` Use the Pythagorean theorem to find half the chord: semi-chord $= \sqrt{5^2 - 3^2}$. `[CONCLUSION]` The semi-chord is 4."

***Retrieved Step:*** "[CONDITION] A standard right triangle has a hypotenuse of 13 and one leg of 5. [PROCESS] To find the other leg, use the Pythagorean theorem: leg $= \sqrt{13^2 - 5^2}$. [CONCLUSION] The other leg is 12."

***Reasoning:*** Demonstrates the necessary *method* (Pythagorean theorem) but not in the same *problem context* (circle geometry vs. basic triangle). Strong 4.

EXAMPLE FOR SCORE 3 (FAIR)

***Gold Step:*** "[CONDITION] We need to solve the inequality $x^2 - 4x + 3 < 0$. [PROCESS] First, find the roots of $x^2 - 4x + 3 = 0$, which are $x = 1$ and $x = 3$. Then test the intervals. [CONCLUSION] The solution is $1 < x < 3$."

***Retrieved Step:*** "[CONDITION] We need to solve the equation $x^2 - 5x + 6 = 0$. [PROCESS] Factor it as $(x - 2)(x - 3) = 0$. [CONCLUSION] The roots are $x = 2$ and $x = 3$."

***Reasoning:*** Demonstrates a necessary sub-step (finding roots) but fails to address the main goal (solving the inequality). Fair 3.

EXAMPLE FOR SCORE 2 (POOR)

***Gold Step:*** "[CONDITION] We need the volume of a cylinder with radius $r = 3$ and height $h = 10$. [PROCESS] The volume formula is $V = \pi \cdot r^2 \cdot h = \pi \cdot 3^2 \cdot 10$. [CONCLUSION] The volume is $90\pi$."

***Retrieved Step:*** "[CONDITION] We need a property of a cylinder with radius $r = 3$ and height $h = 10$. [PROCESS] The surface area formula is $A = 2\pi r h + 2\pi r^2 = 60\pi + 18\pi$. [CONCLUSION] The surface area is $78\pi$."

***Reasoning:*** The topic (cylinder) is the same, but the retrieved step uses the wrong formula (surface area instead of volume). This is confusing and unhelpful. Clear 2.

EXAMPLE FOR SCORE 1 (IRRELEVANT)

***Gold Step:*** "[CONDITION] We need the probability of drawing two aces from a 52-card deck. [PROCESS] $P(A \text{ and } B) = (4/52) \cdot (3/51)$. [CONCLUSION] The probability is $1/221$."

***Retrieved Step:*** "[CONDITION] The function is $f(x) = x^3$. [PROCESS] Apply the power rule for derivatives. [CONCLUSION] The derivative is $f'(x) = 3x^2$."

***Reasoning:*** Completely unrelated mathematical domains (probability vs. calculus). Obvious 1.

---

**Part 3: Your Task**  Analyze the two steps below. Your response **MUST** be a single integer from 1 to 5.
**Gold Step:**
{*gold_step_content*}
**Retrieved Step:**
{*retrieved_step_content*}
**Your Score (1-5):**

Table 20: Instructions and demonstrations for special tokens.

## F.4 EXPERIMENT

---

**Instructions**

**step_binary_accuracy**

System Prompt: You are a strict math logic evaluator. Compare the 'Generated Step' and 'Gold Step'. Your task is to determine if the generated step is logically and numerically equivalent to the gold step. Return a single JSON object: `{"is_correct": true}` or `{"is_correct": false}`.

---

Table 21: Instructions and demonstrations for evaluating whether the two steps are correct, which is used in DSM, TCN, RCR.

---

**Instructions**

**SRR**

System Prompt: You are an expert comparing step-by-step solutions. Compare 'Target Steps' and 'Example Steps' pairwise. Return a JSON object: `{"num_common_steps": `$T$`, "num_related_steps": `$R$`}` where $T$ is the number of pairs compared (length of the shorter list) and $R$ is the count of logically related pairs.

**LUR**

System Prompt: You are an expert comparing step-by-step solutions. Determine how many 'Target Steps' are logically covered by *any* of the 'Example Steps'. Return a JSON object: `{"num_target_steps": `$T$`, "num_utilized_steps": `$U$`}` where $T$ is the total number of target steps and $U$ is the count of target steps with a logical equivalent in the example.

---

Table 22: Instructions and demonstrations for evaluating whether the two steps are correct, which is used in DSM, TCN, RCR.

## G  LLM USAGE STATEMENT

The LLM was used as a tool to assist with polishing the writing and did not directly contribute to theresearch findings or results. All content generated by the LLM was thoroughly reviewed and editedby the authors to ensure its relevance, accuracy, and scientific integrity.

