# OpenReview forum: "Find tailored step example for next step: A Targeted Step-wise Retrieval Framework for Guiding LLM Reasoning"
_ICLR.cc/2026/Conference — Submitted to ICLR 2026_

### Official Review · Reviewer_Fy6i · 2025-10-30

**Soundness:** 3
**Presentation:** 2
**Contribution:** 2
**Rating:** 2
**Confidence:** 4

**Summary:**

This paper focuses on retrieving the examples for mathematical reasoning via Large Language Models (LLMs). Existing methods often provide entire problems as examples, which may not align with the reasoning trajectory of the target problem, and some may even mislead the inference process. To this end, the authors propose Tailored Step Search (TSS), a framework that dynamically decides when to retrieve a single logically consistent next step (still, I do not think that the authors have given a clear definition to the ``logically consistent''). Experiments on four mathematical reasoning datasets demonstrate that they can improve reasoning accuracy and reliability compared to multiple baseline methods.

**Strengths:**

1. The proposed method outperforms multiple baseline methods in the experiments.

**Weaknesses:**

1. The writing could be improved. From lines 258 to 280, the line break is confusing. Element substitution negatives for replacing the problem stem and the targeted step can be presented in a single paragraph for better understanding. In Figure 1, the meaning of the red and blue font colors is not explained in the caption.

2. Some important information is missing from the paper. From 370 to 372, for the baseline methods, the authors should give a brief introduction to SPELL and IDS methods, at least their key differences in working mechanism compared to the proposed method.

3. The motivation is not clear enough to me. Based on Figure 1, I still do not understand what it looks like for a perfectly retrieved example. Regarding the given retrieved example in Figure 1, what do you mean by "correspond one-to-one with the corresponding steps in the golden steps"?

**Questions:**

1. As retrieving good examples for ICL is common in different applications of LLMs, how can the method be generalized to other tasks like question answering?

2. What is the computational cost for the extra procedure for judging and retrieving the examples?

---

> ### Author Response · Authors · 2025-11-25
> **Response to W1 & W2: Revision of article flaws**
>
> Thank you for your feedback on our article! Regarding W1 & W2, we have completed the revisions in the newly uploaded paper.
>
> **1.About the line break issue and the text explanation in Figure 1 (W1)**
>
> Thank you very much for pointing out the unclear parts in our writing. We sincerely apologize for the inconvenience these issues may have caused you. Based on your suggestions, we have adjusted the page layout and merged the paragraphs from lines 258 to 280.
>
> We have also revised the caption of Figure 1 to clearly explain the meaning of the different colored fonts. Here, we will restate it once more. In Figure 1,  Actual LLM Inference shows the actual trajectory of Raw Problem. Gold Inference and Example’s Gold Inference show the standard solution of Raw Problem and  Retrieved Problem retrieved for Raw Problem respectively. The red part represents reasoning steps that are structurally misaligned and misleading with respect to the target. The Step 1 of Example’s Gold Inference is marked in red because the target problem requires deriving a circle equation, not analyzing rectangle vertices. This mismatch misguides the LLM(Step 2 of Actual LLM Inference is the start of error). The blue trajectory represents reasoning steps that are logically aligned and useful for the target problem.
>
> **2.A brief introduction to benchmark methods (W2)**
>
> SPELL  assumes that prompts with lower perplexity can lead to better performance. It calculates the perplexity of prompt words formed by multiple candidate examples, and then selects the k examples with the lowest perplexity as prompts for reasoning. In contrast, TSS retrieves specific reasoning steps on-demand during inference to guide the model's logic.
>
> IDS uses zero-shot chain-of-thought (Zero-shot-CoT) to generate an initial reasoning path for the test question, and based on this, retrieves k semantically similar 'question-answer' pairs from the training set to serve as examples for the next iteration. This process involves iterative reasoning. In contrast, our TSS operates at the step-level granularity, dynamically retrieving only the specific next step required by the current reasoning state, thereby providing more precise and fine-grained supervision, and it also does not require iteration.

---

> ### Author Response · Authors · 2025-11-25
> **Response to W3: Clarification of Research Motivation**
>
> Thanks for your insight! We regret that Figure 1 failed to clearly convey our motivation. Our core point is that traditional RAG may retrieve examples that are semantically related but follow a completely different problem-solving logic path. An 'ideal' retrieved example should provide a next-step illustration that is logically similar (for instance, using the same theorem or technique) when encountering difficulty at the current step, even if it comes from a superficially entirely different problem. We will redesign Figure 1 or add a more persuasive example to clarify this point. Regarding the 'one-to-one correspond,' we mean that the ideal guiding step should align logically with the next step in the gold solution, whereas the steps in the problem-solving solutions retrieved by existing methods are often misaligned and irrelevant to the sequence of steps in the target task. Here we provide a example to help you understand our motivation:
>
> To clarify these concepts and our motivation, we present a side-by-side comparison of three scenarios:
> 1.  **Target Problem**: The problem to be solved.
> 2.  **Imperfect Example (Baseline)**: What standard semantic retrieval typically finds (high textual similarity, logical mismatch).
> 3.  **Perfect Example (Ideal)**: What an ideal retrieval looks like (structural alignment).
>
> **Illustrative Case: Quadratic Minimization**
>
>   Target Problem: Find the minimum of $f(x) = (x-2)^2 + 3$ on the interval $[5, 10]$.
>
>   Key Logic: The vertex ($x=2$) is **outside** the interval. We must evaluate the boundary.
>
> **Step-by-Step Correspondence Analysis**
>
> | Step | (A) Target Problem(Reasoning required) | (B) Imperfect Example(Standard Retrieval) | (C) Perfect Example(Ideal example) |
> | :--- | :--- | :--- | :--- |
> | Problem Setup | Find min of $f(x)=(x-2)^2+3$on interval *$[5, 10]$ | Find min of $f(x)=(x-2)^2+3$ (Unconstrained) | Find min of $g(x)=(x-1)^2+5$ on interval $[3, 6]$ |
> | Why Retrieved? | N/A | High Semantic Similarity.Identical equation. Retrievers rank this first. | Structural Similarity.Different numbers, but same logical constraints. |
> | Step 1(Find Axis) | Find axis of symmetry:$x = 2$. | Find axis of symmetry:$x = 2$. | Find axis of symmetry:$x = 1$. |
> | Alignment | | MATCH: $\checkmark$ | MATCH: $\checkmark$ |
> | Step 2(Check Position) | Check interval:Is $2 \in [5, 10]$? No.Vertex is to the left. | Direct Conclusion:Since $a>0$, the minimum is at the vertex. | Check interval:Is $1 \in [3, 6]$? No.Vertex is to the left. |
> | Alignment | | CRITICAL MISMATCH: $\times$Example teaches: "Ignore interval, take vertex." | MATCH: $\checkmark$Example teaches: "Check if vertex is inside. If not, check boundary." |
> | Step 3(Calculation) | Boundary Evaluation:Min is at boundary $x=5$.$y=(5-2)^2+3=12$. | Vertex Evaluation:Min is at vertex $x=2$.$y=(2-2)^2+3=3$. | Boundary Evaluation:Min is at boundary $x=3$.$y=(3-1)^2+5=9$. |
> | Result | Correct Answer: 12 | Wrong Answer: 3 | Correct Logic |
>
> **Clarification Based on the Analysis:**
>
> 1.  What is a "Perfectly Retrieved Example"?
>     Column (C) represents a "perfect" example. Although the numbers ($1, 3, 6$) differ from the target, the sequence of logical operations (Step 1 $\to$ Step 2 $\to$ Step 3) maps **one-to-one** to the Target Problem. This enables the LLM to successfully imitate the reasoning process (structural isomorphism).
>
> 2.  Why the Baseline Fails (Motivation for Figure 1):
>     Standard retrieval overwhelmingly prefers Column (B) because the function expression is identical to the target (high semantic overlap). However, as shown in Step 2, this leads to a Logical Mismatch. The guidance provided by (B) ("Minimum is at the vertex") is mathematically valid for (B) but false for (A). This misleads the model into an incorrect trajectory, as illustrated in Figure 1.
>
> 3.  How our **TSS** Solves This:
>     Finding a complete problem like (C) is difficult in actual knowledge base,but there are quite a few steps involved in similar operations. **TSS** addresses this problem by dynamically retrieving possible next-step examples. Before step 2, if TSS deems retrieval necessary (based on a learned ability to judge whether retrieval is needed), it will find an example like step 2 in (c). This is thanks to our structured steps, which allow our process not to be limited to the current computation but also to incorporate conclusions derived from previous steps as the basis for the current step. When querying the next step, the model considers not only semantics but also historical conditions, which is further facilitated by our trained retriever capable of querying the next step based on the current step.
>
> Is there anything else you don't understand?

---

> ### Author Response · Authors · 2025-11-27
> **Response to Q1: Explore how to extend to other tasks**
>
> Thanks for your question.This is a very important question. We add this part in our conclusion of our latest paper as potential future work.
>
> Although this article focuses on mathematics, the core idea of TSS,which is providing dynamic, fine-grained logical guidance in complex, multi-step reasoning tasks, has strong potential for generalization. Our guidance focuses on the current state and aims to guide the future, which is not consistent with methods that rely on feedback. We do not use feedback methods but instead aim to implement this context-learning approach for future guidance because in mathematical reasoning, if one reasons first and then looks for similar examples to correct it, using semantic retrieval will also retrieve content similar to the incorrect reasoning, making it difficult to correct. In fact, this is similar across many tasks. For example, in code generation, when encountering difficulties while implementing a specific function, one can retrieve code snippets of functions with similar logic; in complex question-and-answer scenarios that require logical reasoning, one can retrieve similar facts or logical reasoning to guide the next step of the reasoning chain.

---

> ### Author Response · Authors · 2025-11-27
> **Response to Q2: Report average token length (1-shot)**
>
> We thank you for concerning the token cost. The reason we did not include them in the paper before is that, as the first context learning system focusing on future next-step examples, our focus was on example selection and performance improvement. But as you concern, this is an important part. Now we report this part.
>
> Output is the number of tokens generated by the model, and input is the number of tokens fed into the model. It can be seen that our method generally consumes fewer tokens in output compared to the Spell and IDS methods, and in many cases, even fewer than the Vanilla method. This indicates that our [condition][process][conclusion] structure does not lead to excessive token consumption. Regarding input tokens, our method are significantly fewer than those of Method SPELL, and sometimes fewer than Method IDS. However, it is interesting that our `[CONDITION][PROCESS][CONCLUSION]` output structure not only does not increase the number of output tokens, but in many cases also produces slightly fewer tokens, making it more concise. Because of limitation of characters, here we report one-shot, and we will report two-shot, three-shot and four-shot in other review. Thank you!
>
> **1-shot**
>
> | method    | model     | aime_output | aime_input | gsm8k_output | gsm8k_input | math500_output | math500_input | step_output | step_iutput |
> | --------- | --------- | ----------- | ---------- | ------------ | ----------- | -------------- | ------------- | ----------- | ----------- |
> | Vanilla   | llama2-7b | 614.99      | 866.18     | 322.52       | 552.45      | 419.34         | 638.42        | 336.58      | 748.53      |
> | SPELL     | llama2-7b | 658.36      | 16525.52   | 346.02       | 10897.77    | 490.37         | 12837.34      | 516.97      | 13932.07    |
> | IDS       | llama2-7b | 1834.65     | 3595.86    | 1254.37      | 2154.38     | 1647.28        | 2532.96       | 1298.38     | 2245.59     |
> | TSS(Ours) | llama2-7b | 743.14      | 5738.89    | 335.21       | 1976.92     | 545.76         | 3440.9        | 557.53      | 3788.22     |
> | Vanilla   | llama3-8b | 665.96      | 776.24     | 280.54       | 458.39      | 444.49         | 567.05        | 503.87      | 1054.69     |
> | SPELL     | llama3-8b | 735.46      | 14943.45   | 341.73       | 9097.23     | 522.68         | 11462.48      | 580.59      | 12648.33    |
> | IDS       | llama3-8b | 1935.46     | 3563.75    | 1046.47      | 5894.63     | 1764.39        | 1575.47       | 1564.73     | 3375.73     |
> | TSS(Ours) | llama3-8b | 732.35      | 5642.63    | 240.51       | 1631.18     | 447.61         | 3007.13       | 481.56      | 3858.99     |
> | Vanilla   | qwen-3b   | 888.45      | 1712.79    | 290.80       | 505.55      | 554.13         | 608.49        | 586.26      | 1248.78     |
> | SPELL     | qwen-3b   | 848.77      | 15464.91   | 288.05       | 9703.64     | 551.59         | 11901.73      | 600.04      | 13007.92    |
> | IDS       | qwen-3b   | 3464.36     | 5173.73    | 753.73       | 4864.26     | 1353.64        | 4773.37       | 483.32      | 4675.13     |
> | TSS(Ours) | qwen-3b   | 771.98      | 5707.93    | 228.53       | 1856.62     | 479.60         | 3002.87       | 557.05      | 4914.97     |
> | Vanilla   | qwen-7b   | 814.50      | 1637.84    | 304.56       | 809.12      | 532.32         | 607.49        | 590.32      | 661.52      |
> | SPELL     | qwen-7b   | 828.44      | 15531.96   | 306.21       | 9697.96     | 535.70         | 11916.37      | 565.16      | 13037.32    |
> | IDS       | qwen-7b   | 3457.22     | 5387.34    | 954.66       | 2598.55     | 1587.46        | 1893.68       | 2694.35     | 19646.74    |
> | TSS(Ours) | qwen-7b   | 707.83      | 5491.45    | 239.08       | 1542.43     | 468.43         | 3035.87       | 452.90      | 3775.52     |

---

> ### Author Response · Authors · 2025-11-27
> **Response to Q2: Report average token length (2-shot)**
>
> **2-shot**
>
> | method    | model     | aime_output | aime_input | gsm8k_output | gsm8k_input | math500_output | math500_input | step_output | step_iutput |
> | --------- | --------- | ----------- | ---------- | ------------ | ----------- | -------------- | ------------- | ----------- | ----------- |
> | Vanilla   | llama2-7b | 621.90      | 1571.16    | 347.43       | 989.55      | 456.58         | 1150.07       | 347.43      | 989.55      |
> | SPELL     | llama2-7b | 634.86      | 17087.90   | 344.56       | 11364.71    | 472.46         | 13345.99      | 522.29      | 14478.97    |
> | IDS       | llama2-7b | 1854.38     | 4673.95    | 1035.76      | 3024.37     | 1356.58        | 3752.83       | 1434.53     | 2093.88     |
> | TSS(Ours) | llama2-7b | 835.20      | 7822.24    | 344.42       | 2127.55     | 596.41         | 4667.35       | 492.85      | 3563.48     |
> | Vanilla   | llama3-8b | 691.20      | 1404.43    | 278.56       | 815.99      | 460.04         | 1020.43       | 460.03      | 1020.43     |
> | SPELL     | llama3-8b | 712.22      | 15471.05   | 312.95       | 9555.87     | 499.73         | 11955.1       | 563.32      | 13188.91    |
> | IDS       | llama3-8b | 1834.64     | 2254.64    | 1035.46      | 6795.45     | 1674.74        | 1476.23       | 1365.35     | 3046.56     |
> | TSS(Ours) | llama3-8b | 621.73      | 6245.19    | 242.62       | 1759.86     | 466.49         | 3801.58       | 436.97      | 4267.75     |
> | Vanilla   | qwen-3b   | 832.53      | 1479.85    | 279.73       | 890.12      | 546.89         | 1079.96       | 594.64      | 1175.57     |
> | SPELL     | qwen-3b   | 860.25      | 15975.17   | 288.19       | 10193.21    | 553.91         | 12403.89      | 596.39      | 14130.79    |
> | IDS       | qwen-3b   | 2576.57     | 4354.67    | 676.74       | 5365.75     | 1654.45        | 3254.62       | 1764.36     | 3537.93     |
> | TSS(Ours) | qwen-3b   | 795.92      | 7302.85    | 258.47       | 2078.40     | 479.59         | 3604.95       | 457.05      | 4998.46     |
> | Vanilla   | qwen-7b   | 759.53      | 1399.16    | 306.59       | 888.55      | 528.62         | 1078.96       | 565.72      | 1174.57     |
> | SPELL     | qwen-7b   | 860.25      | 15975.17   | 288.19       | 10193.21    | 553.91         | 12403.89      | 596.39      | 14130.79    |
> | IDS       | qwen-7b   | 2883.24     | 3546.63    | 10375.58     | 2874.94     | 1754.78        | 3365.25       | 1764.76     | 3556.62     |
> | TSS(Ours) | qwen-7b   | 673.59      | 6827.57    | 238.31       | 1570.39     | 458.62         | 3648.779      | 423.71      | 3356.66     |

---

> ### Author Response · Authors · 2025-11-27
> **Response to Q2: Report average token length (3-shot)**
>
> **3-shot**
>
> | method    | model     | aime_output | aime_input | gsm8k_output | gsm8k_input | math500_output | math500_input | step_output | step_iutput |
> | --------- | --------- | ----------- | ---------- | ------------ | ----------- | -------------- | ------------- | ----------- | ----------- |
> | Vanilla   | llama2-7b | 568.59      | 2232.44    | 342.36       | 1429.59     | 452.69         | 1645.84       | 452.69      | 1645.83     |
> | SPELL     | llama2-7b | 648.31      | 17671.40   | 319.40       | 11813.67    | 457.686        | 13843.73      | 521.45      | 15009.88    |
> | IDS       | llama2-7b | 146.38      | 6949.78    | 783.93       | 4388.77     | 1357.97        | 5027.49       | 1384.54     | 4994.39     |
> | TSS(Ours) | llama2-7b | 802.02      | 9142.50    | 332.27       | 2118.93     | 578.59         | 5426.03       | 511.34      | 3959.45     |
> | Vanilla   | llama3-8b | 691.16      | 1997.41    | 271.87       | 1177.07     | 452.18         | 1459.28       | 452.18      | 1459.28     |
> | SPELL     | llama3-8b | 693.92      | 15992.24   | 308.83       | 9992.26     | 506.18         | 12938.05      | 525.07      | 13669.58    |
> | IDS       | llama3-8b | 1965.74     | 5987.89    | 876.92       | 3365.36     | 1354.35        | 4365.73       | 1356.87     | 4563.62     |
> | TSS(Ours) | llama3-8b | 637.83      | 7685.86    | 240.53       | 1856.59     | 471.89         | 4648.93       | 434.06      | 5261.31     |
> | Vanilla   | qwen-3b   | 851.59      | 2092.87    | 273.04       | 1277.39     | 530.97         | 1536.64       | 598.35      | 1662.07     |
> | SPELL     | qwen-3b   | 869.82      | 16571.19   | 283.12       | 10661.75    | 546.97         | 12914.37      | 585.33      | 14052.52    |
> | IDS       | qwen-3b   | 2564.73     | 6343.66    | 878.96       | 3675.39     | 1654.62        | 4745.19       | 1865.84     | 5064.88     |
> | TSS(Ours) | qwen-3b   | 771.90      | 8707.93    | 344.95       | 3386.08     | 493.62         | 4737.53       | 537.30      | 5342.75     |
> | Vanilla   | qwen-7b   | 830.1       | 2049.6     | 304.96       | 1580.31     | 526.77         | 1535.64       | 576.70      | 1661.07     |
> | SPELL     | qwen-7b   | 786.06      | 16644.84   | 303.87       | 10658.94    | 526.63         | 12971.09      | 570.25      | 14082.76    |
> | IDS       | qwen-7b   | 3468.57     | 6436.25    | 912.15       | 4682.26     | 1578.33        | 4563.25       | 1674.46     | 5064.42     |
> | TSS(Ours) | qwen-7b   | 686.48      | 8706.33    | 237.47       | 1600.82     | 459.15         | 4316.45       | 416.41      | 4970.45     |

---

> ### Author Response · Authors · 2025-11-27
> **Response to Q2: Report average token length (4-shot)**
>
> Thank you very much for your patience! We are now reporting the results for the 4-shot setting, which is the last one.
>
> **4-shot**
>
> | method    | model     | aime_output | aime_input | gsm8k_output | gsm8k_input | math500_output | math500_input | step_output | step_iutput |
> | --------- | --------- | ----------- | ---------- | ------------ | ----------- | -------------- | ------------- | ----------- | ----------- |
> | Vanilla   | llama2-7b | 568.60      | 2921.02    | 342.37       | 1868.42     | 452.69         | 2164.69       | 482.45      | 2335.61     |
> | SPELL     | llama2-7b | 628.26      | 18257.76   | 322.56       | 12593.96    | 454.76         | 14342.77      | 508.98      | 15535.92    |
> | IDS       | llama2-7b | 1694.94     | 7392.93    | 1098.26      | 5395.18     | 1493.36        | 6373.62       | 1495.65     | 7062.68     |
> | TSS(Ours) | llama2-7b | 859.80      | 11979.99   | 337.05       | 2242.61     | 578.16         | 5956.66       | 476.76      | 4071.24     |
> | Vanilla   | llama3-8b | 691.16      | 2612.07    | 271.87       | 1537.63     | 452.18         | 1919.19       | 452.18      | 1919.19     |
> | SPELL     | llama3-8b | 730.79      | 16547.68   | 297.99       | 10418.43    | 495.13         | 12918.73      | 534.29      | 14173.43    |
> | IDS       | llama3-8b | 1938.27     | 7327.38    | 963.85       | 7474.38     | 1675.82        | 5938.73       | 1364.73     | 5974.83     |
> | TSS(Ours) | llama3-8b | 655.19      | 8959.95    | 241.87       | 1966.41     | 463.48         | 5101.51       | 477.80      | 6801.65     |
> | Vanilla   | qwen-3b   | 851.59      | 2733.95    | 273.03       | 1663.90     | 530.97         | 2016.27       | 598.35      | 2183.59     |
> | SPELL     | qwen-3b   | 875.01      | 17129.89   | 286.04       | 11125.77    | 546.41         | 13423.23      | 609.61      | 14554.55    |
> | IDS       | qwen-3b   | 2574.27     | 9357.83    | 783.53       | 4973.28     | 1595.27        | 3826.27       | 1684.83     | 6472.73     |
> | TSS(Ours) | qwen-3b   | 757.88      | 11603.93   | 246.55       | 4462.43     | 484.35         | 5624.36       | 563.13      | 5914.98     |
> | Vanilla   | qwen-7b   | 830.10      | 2647.33    | 304.96       | 1662.75     | 526.77         | 2015.27       | 576.70      | 2182.59     |
> | SPELL     | qwen-7b   | 775.75      | 17236.22   | 304.76       | 11124.87    | 526.69         | 13472.44      | 569.13      | 14569.82    |
> | IDS       | qwen-7b   | 2445.34     | 8375.48    | 1067.19      | 5234.92     | 1636.28        | 6591.26       | 1593.68     | 6385.93     |
> | TSS(Ours) | qwen-7b   | 708.55      | 10734.20   | 242.57       | 1665.09     | 457.14         | 5000.19       | 432.22      | 5856.34     |

---

> ### Author Response · Authors · 2025-11-27
> **Official Comment by Authors**
>
> Dear Reviewer,
>
> Thank you very much for your time and expertise in reviewing our submission. As the rebuttal deadline is approaching, please let us know if you have any questions or need any clarification. We are very happy to discuss any aspect of the work.

---

### Official Review · Reviewer_Bzfe · 2025-11-01

**Soundness:** 3
**Presentation:** 3
**Contribution:** 2
**Rating:** 4
**Confidence:** 3

**Summary:**

This paper proposes TSS (Tailored Step Search), a framework for improving LLM mathematical reasoning through targeted step-level retrieval. Unlike traditional RAG methods that retrieve entire problems, TSS dynamically retrieves single next-step examples during reasoning. The system consists of: (1) a Step Retriever trained with contrastive learning to capture logical flow between steps, and (2) a Generator trained in two phases to predict retrieval necessity and generate structured reasoning steps in [CONDITION, PROCESS, CONCLUSION] format. The paper evaluates on AIME, GSM8K, MATH500, and a custom STEP dataset  across four LLM backbones (Llama-2-7B, Llama-3-8B, Qwen2.5-3B, Qwen2.5-7B).  They introduce new metrics (DSM, TCN, RCR) for evaluating intermediate
reasoning quality.

**Strengths:**

1. The paper identifies a real limitation of problem-level retrieval in mathematical reasoning - many retrieved steps are misaligned.
2. The [CONDITION, PROCESS, CONCLUSION] decomposition provides an interpretable structure for step-wise reasoning, which could benefit explainability and error analysis.
3. The introduction of DSM (Dynamic Step Matching), TCN (Trajectory Correction Number), and RCR (Retrieval-Corrected Rate) offers fine-grained evaluation of intermediate reasoning quality beyond final answer accuracy.
4. The method shows improvements across multiple model backbones and datasets, suggesting some degree of generalizability.

**Weaknesses:**

1. The paper lacks comparison with directly  relevant prior work: 1) Self-RAG: Also performs adaptive retrieval with similar decide-then-retrieve mechanism. This is a glaring omission; 2) Problem-level retrieval using the SAME knowledge base: Without this comparison, the claimed superiority of step-level over problem-level retrieval is not convincingly demonstrated; 3) Simple retrieval methods (e.g., BM25) as lower bounds.
2, On AIME (122 problems), improvements of 0.82→2.46 represent only ~1-2 additional correct answers, which may lack statistical significance.
3. Missing comparisons on existing benchmarks like MMLU-Math, TheoremQA.
4. No analysis of retrieval quality (precision/recall of relevant steps)

**Questions:**

1. Why is Self-RAG (2024) not included as a baseline? It directly addresses adaptive retrieval in LLM reasoning and should be the primary comparison point. Can you provide results comparing TSS with Self-RAG?
2. Can you provide a controlled comparison where both problem-level and step-level retrieval use the EXACT SAME knowledge base?
3. What is the accuracy of the retrieval decision mechanism during inference? (i.e., when the model predicts <retrieval> vs <no-retrieval>, how often is this decision correct?)
4. How does the automatic annotation quality (using GPT-4o-mini) affect results?
5. How sensitive is the method to the number of negative examples in contrastive learning?

---

> ### Author Response · Authors · 2025-11-26
> **Response to W1 & Q1: Report the mean and standard deviation of multiple runs and supplement with comparison experiments with Self-RAG**
>
> Thanks for your excellent question. We did not previously report the results of Self-RAG[1] in the paper because Self-RAG[1] is trained on QA data. Now we report it below.
>
> **1.Results from multiple runs (mean ± standard deviation) （W1）**
> We understand your perspective. Our results are based on selecting the most stable data after multiple runs. As per your request, we have updated the data in Table 2 of the latest submitted paper to reflect the results of multiple runs (mean ± standard deviation).
>
> **2.Supplementary Self-RAG experiments（W1&Q1）**
>
> Self-RAG is a framework trained to carry out selective retrieval, reasoning, and evaluation processes. It uses retrieval tokens and multiple evaluation tokens for reasoning and judgment.
>
> We will report the results on AIME,GSM8K,MATH500 and STEP on 1-shot, 2-shot, 3-shot and 4-shot using Llama-2-7B and Llama2-13B as used in self-RAG.  The accuracy is measured on a percentage scale, precise to two decimal places. To make it easier for you to view, we have also recorded the results of our method here for easy comparison.
>
> **On AIME**
>
> | Base Model | Method   | K-Shot | Accuracy% |
> | ---------- | -------- | ------ | --------- |
> | Llama2-7B  | SELF-RAG | 1-shot | 0.82      |
> | Llama2-7B  | SELF-RAG | 2-shot | 1.63      |
> | Llama2-7B  | SELF-RAG | 3-shot | 0.82      |
> | Llama2-7B  | SELF-RAG | 4-shot | 0.82      |
> | Llama2-7B  | TSS      | 1-shot | 2.46      |
> | Llama2-7B  | TSS      | 2-shot | 2.46      |
> | Llama2-7B  | TSS      | 3-shot | 1.63      |
> | Llama2-7B  | TSS      | 4-shot | 1.63      |
> | Llama2-13B | SELF-RAG | 1-shot | 1.63      |
> | Llama2-13B | SELF-RAG | 2-shot | 2.46      |
> | Llama2-13B | SELF-RAG | 3-shot | 2.46      |
> | Llama2-13B | SELF-RAG | 4-shot | 1.63      |
> | Llama2-13B | TSS      | 1-shot | 4.09      |
> | Llama2-13B | TSS      | 2-shot | 3.28      |
> | Llama2-13B | TSS      | 3-shot | 2.46      |
> | Llama2-13B | TSS      | 4-shot | 3.28      |
>
> **On GSM8K**
>
> | Base Model | Method   | K-Shot | Accuracy% |
> | ---------- | -------- | ------ | --------- |
> | Llama2-7B  | SELF-RAG | 1-shot | 15.06     |
> | Llama2-7B  | SELF-RAG | 2-shot | 15.78     |
> | Llama2-7B  | SELF-RAG | 3-shot | 15.78     |
> | Llama2-7B  | SELF-RAG | 4-shot | 16.63     |
> | Llama2-7B  | TSS      | 1-shot | 18.18     |
> | Llama2-7B  | TSS      | 2-shot | 19.31     |
> | Llama2-7B  | TSS      | 3-shot | 20.65     |
> | Llama2-7B  | TSS      | 4-shot | 18.89     |
> | Llama2-13B | SELF-RAG | 1-shot | 19.81     |
> | Llama2-13B | SELF-RAG | 2-shot | 20.88     |
> | Llama2-13B | SELF-RAG | 3-shot | 20.74     |
> | Llama2-13B | SELF-RAG | 4-shot | 20.95     |
> | Llama2-13B | TSS      | 1-shot | 30.37     |
> | Llama2-13B | TSS      | 2-shot | 31.22     |
> | Llama2-13B | TSS      | 3-shot | 31.15     |
> | Llama2-13B | TSS      | 4-shot | 30.30     |
>
> **On MATH500**
>
> | Base Model | Method   | K-Shot | Accuracy% |
> | ---------- | -------- | ------ | --------- |
> | Llama2-7B  | SELF-RAG | 1-shot | 3.80      |
> | Llama2-7B  | SELF-RAG | 2-shot | 3.20      |
> | Llama2-7B  | SELF-RAG | 3-shot | 2.60      |
> | Llama2-7B  | SELF-RAG | 4-shot | 3.00      |
> | Llama2-7B  | TSS      | 1-shot | 5.00      |
> | Llama2-7B  | TSS      | 2-shot | 6.20      |
> | Llama2-7B  | TSS      | 3-shot | 4.80      |
> | Llama2-7B  | TSS      | 4-shot | 4.20      |
> | Llama2-13B | SELF-RAG | 1-shot | 5.60      |
> | Llama2-13B | SELF-RAG | 2-shot | 5.40      |
> | Llama2-13B | SELF-RAG | 3-shot | 5.20      |
> | Llama2-13B | SELF-RAG | 4-shot | 5.10      |
> | Llama2-13B | TSS      | 1-shot | 6.40      |
> | Llama2-13B | TSS      | 2-shot | 7.00      |
> | Llama2-13B | TSS      | 3-shot | 6.40      |
> | Llama2-13B | TSS      | 4-shot | 6.80      |
>
> **On our STEP Dataset**
>
> | Base Model | Method   | K-Shot | Accuracy% |
> | ---------- | -------- | ------ | --------- |
> | Llama2-7B  | SELF-RAG | 1-shot | 8.25      |
> | Llama2-7B  | SELF-RAG | 2-shot | 9.28      |
> | Llama2-7B  | SELF-RAG | 3-shot | 9.28      |
> | Llama2-7B  | SELF-RAG | 4-shot | 10.31     |
> | Llama2-7B  | TSS      | 1-shot | 15.46     |
> | Llama2-7B  | TSS      | 2-shot | 11.34     |
> | Llama2-7B  | TSS      | 3-shot | 13.40     |
> | Llama2-7B  | TSS      | 4-shot | 17.53     |
> | Llama2-13B | SELF-RAG | 1-shot | 9.28      |
> | Llama2-13B | SELF-RAG | 2-shot | 9.28      |
> | Llama2-13B | SELF-RAG | 3-shot | 10.31     |
> | Llama2-13B | SELF-RAG | 4-shot | 9.28      |
> | Llama2-13B | TSS      | 1-shot | 14.43     |
> | Llama2-13B | TSS      | 2-shot | 17.53     |
> | Llama2-13B | TSS      | 3-shot | 14.43     |
> | Llama2-13B | TSS      | 4-shot | 13.40     |
>
> In terms of results, our method outperforms self-rag in most settings. On GSM8K, Our frameworks are all significantly higher than Self-RAG[1]. On other dataset, our **TSS** also perform better, and our mechanism is designed to be simpler than Self-RAG[1].
>
> [1] SELF-RAG: Learning to Retrieve, Generate and Critique through Self-reflection. ICLR 2024.

---

> ### Author Response · Authors · 2025-11-26
> **Response to Q3, Q4 & Q5: Prediction accuracy of special tokens, evaluation on the automatic annotation quality (using GPT-4o-mini)  and sensitivity to the number of negative examples**
>
> Your suggestions are constructive! Thanks for your question. We will answer your questions below.
>
> * **Prediction accuracy of special tokens (Q3)**
>
> You mention that what is the accuracy of the retrieval decision mechanism during inference. Our design intention is to train the generator to develop a kind of 'self-awareness,' that is, to determine whether the current state is sufficient to proceed to the next step. Our goal is to enable the model to learn this ability. We are now evaluating whether the model has learned this capability. We will assess our fine-tuned generator's accuracy in predicting \<retrieval\> and \<no_retrieval\> during reasoning (compared to GPT-4o-mini's annotations), then evaluate the accuracy and recall of predictions\<retrieval\> and \<no_retrieval\> respectively. From the table below, our model's prediction accuracy and recall are both good, indicating that the model has learned the ability to decide whether retrieval is needed in the current state.
>
> |      model       | retrieval_precision | retrieval_recall | no-retrieval_precision | no-retrieval_recall |
> | ----------- | ------------------- | ---------------- | ---------------------- | ------------------- |
> | Llama-2-7b  | 0.8209              | 0.7331           | 0.7818                 | 0.8929              |
> | Llama-3-8B  | 0.8852              | 0.9421           | 0.9151                 | 0.8077              |
> | Qwen-2.5-3B | 0.8384              | 0.8348           | 0.8459                 | 0.8545              |
> | Qwen-2.5-7B  | 0.9176              | 0.9075           | 0.9406                 | 0.9021              |
>
> * **evaluation on the automatic annotation quality (using GPT-4o-mini) (Q4)**
>
> We randomly selected 100 samples from the Generator Training Data dataset and invited six highly educated evaluators (each with at least a bachelor's degree) to conduct the evaluation. The evaluation focused on two key dimensions: the rationality of the content in structured steps ([]CONDITION[PROCESS][CONCLUSION]) and the reasonableness of special tokens that require retrieval. As shown in the table below, we used Fleiss’ Kappa coefficient to assess the reliability of the annotations. The Kappa values for both the rationality of structured step content and the reasonableness of special tokens requiring retrieval exceeded the 0.41 reliability threshold, thereby confirming the reliability of our annotation results.
>
> **The Kappa values for the rationality of structured step content**
>
> |         source          | number | Fleiss' Kappa |
> | :---------------------: | :----: | :-----------: |
> | Generator Training Data |  100   |     0.66      |
>
>
> **The Kappa values for both the reasonableness of special tokens requiring retrieval**
>
> |         source          | number | Fleiss' Kappa |
> | :---------------------: | :----: | :-----------: |
> | Generator Training Data |  100   |     0.52      |
>
>
> * **sensitivity to the number of negative examples (Q5)**
>
> This is a good question about the robustness of our contrastive learning training. We will conduct a sensitivity analysis experiment. When training our retriever, we will vary the number of hard negative samples paired with each positive example (8, 16, 32, 64) and observe its impact on the retriever's performance (Recall@k), to see whether the retriever can recall the next step among the k retrieved candidates.
>
> | batch_size | Recall@1 | Recall@3 | Recall@5 |
> | ---------- | -------- | -------- | -------- |
> | 8          | 0.833147 | 0.8522   | 0.8903   |
> | 16         | 0.8394   | 0.8627   | 0.9064   |
> | 32         | 0.8450   | 0.8834   | 0.9294   |
> | 64         | 0.8492   | 0.8968   | 0.9395   |
>
> It can be observed that the recall rate is quite good, already exceeding 83% at k=3, and it can even reach 90% with k=5.

---

> ### Author Response · Authors · 2025-11-27
> **Response to W3: a method for evaluating impact of retrieved examples on the results**
>
> Thank you for these insightful questions !
>
> As the first system to consider step prediction examples, our focus is on how to implement the search for examples for the next step and  validate the feasibility of this approach. But We did conduct a quality assessment as below: .
>
> We define the metrics `adherence`, `adherence_precision` and  `adherence_recall`.
>
> We set an "Adherence Threshold" `ADHERENCE_THRESHOLD`. This means that if the average adherence score of a question is $\ge 3$, we consider that the model has "Effectively Learned (High Adherence)" the retrieved content; otherwise, it is considered "Ineffective Learning (Low Adherence)". We only analyze questions where retrieval actually occurred.
>
> **According to the Adherence Threshold and whether it is actually correct, we can get four types of results:**
>
> TP (True Positive,High Adherence & Correct): The model effectively learned the retrieved content and finally answered the question correctly. This is the ideal working state of the retrieval system.
>
> FP (False Positive, High Adherence & Wrong) : The model effectively learned the retrieved content, but eventually answered incorrectly. This usually means that the retrieved content itself is incorrect or misleading. This is the "bad teammate" effect, where the retriever found a wrong example.
>
> FN (False Negative, Low Adherence & Correct): The model did not effectively learn the retrieved content (or even completely ignored it), but it figured out the correct answer by itself. This shows that for this question, retrieval is useless or even potentially distracting. The model's own knowledge is sufficient.
>
> TN (True Negative, Low Adherence & Wrong): The model did not effectively learn the retrieved content, and it failed to figure out the correct answer by itself, resulting in a wrong answer. This shows that the model neither learned nor knew it.
>
> Now, we define the meaning of adherence_precision and adherence_recall:
>
> **1. adherence_precision**:  $ \frac{TP}{TP + FP} $
>
> Among all cases where the model claims to have "effectively learned" the retrieved content (High Adherence), what proportion eventually led to a correct result.
>
> **2. adherence_recall**: $  \frac{TP}{TP + FN} $
>
> Among all questions that were eventually answered correctly, what proportion was achieved through "effectively learning" the retrieved content.
>
> We use gpt-4o to evaluate, the prompt is:
>
> You are a learning ability analyst. Your task is to evaluate to what extent a model, after seeing a reference example (Retrieved Example), adopts the example's core logic or problem-solving method in its generated next step (Generated Step). Follow 5-point scale.
>
> 5: Perfectly adopted the example's core method and applied it correctly.
>
> 4: Mostly adopted the example's method, but the application was slightly inappropriate or incomplete.
>
> 3: Adopted some elements from the example, but the core method is different.
>
> 2: Appears to have attempted adoption, but used it completely incorrectly.
>
> 1: Completely ignored the reference example
>
> | model      | shot | retrieval_num | adherence | adherence_precision | adherence_recall |
> | ---------- | :--: | :-----------: | :-------: | :-----------------: | :--------------: |
> | Llama-2-7B |  1   |      233      |   3.75    |        70.64        |      68.68       |
> | Llama-2-7B |  2   |      228      |   3.96    |        64.17        |      63.33       |
> | Llama-2-7B |  3   |      235      |   3.77    |        71.11        |      75.00       |
> | Llama-2-7B |  4   |      221      |   3.72    |        72.18        |      67.14       |
> | Llama-3-8B |  1   |      324      |   3.60    |        76.98        |      80.18       |
> | Llama-3-8B |  2   |      328      |   3.93    |        75.71        |      78.95       |
> | Llama-3-8B |  3   |      318      |   3.38    |        76.73        |      69.23       |
> | Llama-3-8B |  4   |      331      |   3.26    |        77.27        |      83.33       |
> | Qwen2-3B   |  1   |      264      |   3.35    |        79.83        |      78.19       |
> | Qwen-3B    |  2   |      272      |   3.87    |        83.33        |      74.02       |
> | Qwen-3B    |  3   |      262      |   3.65    |        86.51        |      82.84       |
> | Qwen-3B    |  4   |      268      |   3.29    |        78.25        |      83.47       |
> | Qwen-7B    |  1   |      296      |   3.32    |        78.33        |      72.88       |
> | Qwen-7B    |  2   |      302      |   3.78    |        78.46        |      63.56       |
> | Qwen-7B    |  3   |      298      |   3.81    |        87.50        |      76.00       |
> | Qwen-7B    |  4   |      304      |   3.19    |        77.25        |      67.86       |
>
> retrieval_num represents the number of triggering retrieval.  The examples we retrieved basically exert a positive effect on the results, especially Qwen-2.5-3B and Llama-3-8B .

---

> ### Author Response · Authors · 2025-11-27
> **Response to W2: comparisons on existing benchmarks MMLU-college**
>
> Thank you for your suggestion. We will complete the supplementary experiments on MMLU-college having 100 college-level questions.
>
> **mmlu-college**
>
> | Base Model    | Method       | K-Shot                | Accuracy% |
> | ------------- | ------------ | --------------------- | --------- |
> | Qwen-3B       | No Retrieval | 0-shot (No Retrieval) | 35.00     |
> | Qwen-3B       | vallina      | 1-shot                | 38.00     |
> | Qwen-3B       | vallina      | 2-shot                | 37.00     |
> | Qwen-3B       | vallina      | 3-shot                | 34.00     |
> | Qwen-3B       | vallina      | 4-shot                | 31.00     |
> | Qwen-3B       | IDS          | 1-shot                | 44.00     |
> | Qwen-3B       | IDS          | 2-shot                | 43.00     |
> | Qwen-3B       | IDS          | 3-shot                | 43.00     |
> | Qwen-3B       | IDS          | 4-shot                | 37.00     |
> | Qwen-3B       | SPELL        | 1-shot                | 47.00     |
> | Qwen-3B       | SPELL        | 2-shot                | 46.00     |
> | Qwen-3B       | SPELL        | 3-shot                | 43.00     |
> | Qwen-3B       | SPELL        | 4-shot                | 41.00     |
> | **Qwen-3B**   | **TSS**      | **1-shot**            | **52.00** |
> | **Qwen-3B**   | **TSS**      | **2-shot**            | **49.00** |
> | **Qwen-3B**   | **TSS**      | **3-shot**            | **46.00** |
> | **Qwen-3B**   | **TSS**      | **4-shot**            | **45.00** |
> | Qwen-7B       | No Retrieval | 0-shot (No Retrieval) | 78.00     |
> | Qwen-7B       | vallina      | 1-shot                | 77.00     |
> | Qwen-7B       | vallina      | 2-shot                | 79.00     |
> | Qwen-7B       | vallina      | 3-shot                | 80.00     |
> | Qwen-7B       | vallina      | 4-shot                | 81.00     |
> | Qwen-7B       | IDS          | 1-shot                | 82.00     |
> | Qwen-7B       | IDS          | 2-shot                | 83.00     |
> | Qwen-7B       | IDS          | 3-shot                | 80.00     |
> | Qwen-7B       | IDS          | 4-shot                | 79.00     |
> | **Qwen-7B**   | **TSS**      | **1-shot**            | **86.00** |
> | **Qwen-7B**   | **TSS**      | **2-shot**            | **85.00** |
> | **Qwen-7B**   | **TSS**      | **3-shot**            | **82.00** |
> | **Qwen-7B**   | **TSS**      | **4-shot**            | **83.00** |
> | Qwen-7B       | SPELL        | 1-shot                | 76.00     |
> | Qwen-7B       | SPELL        | 2-shot                | 73.00     |
> | Qwen-7B       | SPELL        | 3-shot                | 75.00     |
> | Qwen-7B       | SPELL        | 4-shot                | 73.00     |
> | Llama2-7B     | No Retrieval | 0-shot (No Retrieval) | 24.00     |
> | Llama2-7B     | vallina      | 1-shot                | 13.00     |
> | Llama2-7B     | vallina      | 2-shot                | 14.00     |
> | Llama2-7B     | vallina      | 3-shot                | 15.00     |
> | Llama2-7B     | vallina      | 4-shot                | 13.00     |
> | Llama2-7B     | IDS          | 1-shot                | 16.00     |
> | Llama2-7B     | IDS          | 2-shot                | 14.00     |
> | Llama2-7B     | IDS          | 3-shot                | 13.00     |
> | Llama2-7B     | IDS          | 4-shot                | 15.00     |
> | Llama2-7B     | SPELL        | 2-shot                | 25.00     |
> | Llama2-7B     | SPELL        | 1-shot                | 24.00     |
> | Llama2-7B     | SPELL        | 3-shot                | 26.00     |
> | Llama2-7B     | SPELL        | 4-shot                | 22.00     |
> | **Llama2-7B** | **TSS**      | **1-shot**            | **21.00** |
> | **Llama2-7B** | **TSS**      | **2-shot**            | **27.00** |
> | **Llama2-7B** | **TSS**      | **3-shot**            | **27.00** |
> | **Llama2-7B** | **TSS**      | **4-shot**            | **25.00** |
> | Llama3-8B     | No Retrieval | 0-shot (No Retrieval) | 35.00     |
> | Llama3-8B     | vallina      | 1-shot                | 37.00     |
> | Llama3-8B     | vallina      | 2-shot                | 36.00     |
> | Llama3-8B     | vallina      | 3-shot                | 37.00     |
> | Llama3-8B     | vallina      | 4-shot                | 38.00     |
> | Llama3-8B     | IDS          | 1-shot                | 41.00     |
> | Llama3-8B     | IDS          | 2-shot                | 43.00     |
> | Llama3-8B     | IDS          | 4-shot                | 44.00     |
> | Llama3-8B     | IDS          | 3-shot                | 42.00     |
> | Llama3-8B     | SPELL        | 1-shot                | 47.00     |
> | Llama3-8B     | SPELL        | 2-shot                | 45.00     |
> | Llama3-8B     | SPELL        | 3-shot                | 43.00     |
> | Llama3-8B     | SPELL        | 4-shot                | 40.00     |
> | **Llama3-8B** | **TSS**      | **1-shot**            | **49.00** |
> | **Llama3-8B** | **TSS**      | **2-shot**            | **48.00** |
> | **Llama3-8B** | **TSS**      | **3-shot**            | **44.00** |
> | **Llama3-8B** | **TSS**      | **4-shot**            | **44.00** |

---

> ### Author Response · Authors · 2025-11-27
> **Response to W2: comparisons on existing benchmarks MMLU-highschool**
>
> Here we will complete the supplementary experiments on MMLU-highschool[1] having 270 high school-level questions
>
> **MMLU-highschool**
>
> | Base Model    | Method       | K-Shot                | Accuracy% |
> | ------------- | ------------ | --------------------- | --------- |
> | Qwen-3B       | No Retrieval | 0-shot (No Retrieval) | 66.67     |
> | Qwen-3B       | vallina      | 1-shot                | 67.41     |
> | Qwen-3B       | vallina      | 2-shot                | 68.52     |
> | Qwen-3B       | vallina      | 3-shot                | 67.41     |
> | Qwen-3B       | vallina      | 4-shot                | 68.15     |
> | Qwen-3B       | IDS          | 1-shot                | 68.89     |
> | Qwen-3B       | IDS          | 2-shot                | 69.63     |
> | Qwen-3B       | IDS          | 3-shot                | 72.22     |
> | Qwen-3B       | IDS          | 4-shot                | 70.00     |
> | Qwen-3B       | SPELL        | 1-shot                | 73.33     |
> | Qwen-3B       | SPELL        | 2-shot                | 74.07     |
> | Qwen-3B       | SPELL        | 3-shot                | 72.96     |
> | Qwen-3B       | SPELL        | 4-shot                | 72.59     |
> | **Qwen-3B**   | **TSS**      | **1-shot**            | **76.30** |
> | **Qwen-3B**   | **TSS**      | **2-shot**            | **76.67** |
> | **Qwen-3B**   | **TSS**      | **3-shot**            | **77.04** |
> | **Qwen-3B**   | **TSS**      | **4-shot**            | **75.93** |
> | Qwen-7B       | vallina      | 1-shot                | 87.41     |
> | Qwen-7B       | vallina      | 2-shot                | 87.78     |
> | Qwen-7B       | vallina      | 3-shot                | 87.04     |
> | Qwen-7B       | vallina      | 4-shot                | 86.67     |
> | Qwen-7B       | IDS          | 1-shot                | 88.32     |
> | Qwen-7B       | IDS          | 2-shot                | 88.89     |
> | Qwen-7B       | IDS          | 3-shot                | 88.15     |
> | Qwen-7B       | IDS          | 4-shot                | 87.41     |
> | Qwen-7B       | SPELL        | 1-shot                | 88.15     |
> | Qwen-7B       | SPELL        | 2-shot                | 87.87     |
> | Qwen-7B       | SPELL        | 3-shot                | 87.04     |
> | Qwen-7B       | SPELL        | 4-shot                | 87.78     |
> | **Qwen-7B**   | **TSS**      | **1-shot**            | **90.00** |
> | **Qwen-7B**   | **TSS**      | **2-shot**            | **89.63** |
> | **Qwen-7B**   | **TSS**      | **3-shot**            | **90.37** |
> | **Qwen-7B**   | **TSS**      | **4-shot**            | **89.26** |
> | Llama2-7B     | No Retrieval | 0-shot (No Retrieval) | 37.40     |
> | Llama2-7B     | vallina      | 1-shot                | 37.03     |
> | Llama2-7B     | vallina      | 2-shot                | 37.78     |
> | Llama2-7B     | vallina      | 3-shot                | 38.89     |
> | Llama2-7B     | vallina      | 4-shot                | 38.15     |
> | Llama2-7B     | SPELL        | 1-shot                | 39.26     |
> | Llama2-7B     | SPELL        | 2-shot                | 41.48     |
> | Llama2-7B     | SPELL        | 3-shot                | 40.37     |
> | Llama2-7B     | SPELL        | 4-shot                | 40.00     |
> | Llama2-7B     | IDS          | 1-shot                | 41.85     |
> | Llama2-7B     | IDS          | 2-shot                | 42.59     |
> | Llama2-7B     | IDS          | 3-shot                | 42.96     |
> | Llama2-7B     | IDS          | 4-shot                | 42.22     |
> | **Llama2-7B** | **TSS**      | **1-shot**            | **44.44** |
> | **Llama2-7B** | **TSS**      | **2-shot**            | **43.70** |
> | **Llama2-7B** | **TSS**      | **3-shot**            | **45.18** |
> | **Llama2-7B** | **TSS**      | **4-shot**            | **44.07** |
> | Llama3-8B     | No Retrieval | 0-shot (No Retrieval) | 68.89     |
> | Llama3-8B     | vallina      | 1-shot                | 71.85     |
> | Llama3-8B     | vallina      | 2-shot                | 70.37     |
> | Llama3-8B     | vallina      | 3-shot                | 68.89     |
> | Llama3-8B     | vallina      | 4-shot                | 73.33     |
> | Llama3-8B     | SPELL        | 1-shot                | 73.33     |
> | Llama3-8B     | SPELL        | 2-shot                | 70.74     |
> | Llama3-8B     | SPELL        | 3-shot                | 73.70     |
> | Llama3-8B     | SPELL        | 4-shot                | 74.07     |
> | Llama3-8B     | IDS          | 1-shot                | 73.70     |
> | Llama3-8B     | IDS          | 2-shot                | 75.92     |
> | Llama3-8B     | IDS          | 3-shot                | 74.81     |
> | Llama3-8B     | IDS          | 4-shot                | 76.29     |
> | **Llama3-8B** | **TSS**      | **1-shot**            | **75.19** |
> | **Llama3-8B** | **TSS**      | **2-shot**            | **77.78** |
> | **Llama3-8B** | **TSS**      | **3-shot**            | **77.03** |
> | **Llama3-8B** | **TSS**      | **4-shot**            | **75.92** |
>
> [1] MMLU: Measuring Massive Multitask Language Understanding. ICLR 2021.

---

> ### Author Response · Authors · 2025-11-27
> **Response to W2: comparisons on existing benchmarks TheoremQA**
>
> Here we will complete the experiments on TheoremQA[1] having 747 questions.
>
> **TheoremQA**
>
> | Base Model    | Method       | K-Shot                | Accuracy% |
> | ------------- | ------------ | --------------------- | --------- |
> | Qwen-3B       | No Retrieval | 0-shot (No Retrieval) | 27.58     |
> | Qwen-3B       | vallina      | 1-shot                | 27.84     |
> | Qwen-3B       | vallina      | 2-shot                | 28.11     |
> | Qwen-3B       | vallina      | 3-shot                | 28.38     |
> | Qwen-3B       | vallina      | 4-shot                | 29.18     |
> | Qwen-3B       | IDS          | 1-shot                | 28.92     |
> | Qwen-3B       | IDS          | 2-shot                | 28.65     |
> | Qwen-3B       | IDS          | 3-shot                | 29.45     |
> | Qwen-3B       | IDS          | 4-shot                | 29.72     |
> | Qwen-3B       | SPELL        | 1-shot                | 29.99     |
> | Qwen-3B       | SPELL        | 2-shot                | 28.92     |
> | Qwen-3B       | SPELL        | 3-shot                | 31.19     |
> | Qwen-3B       | SPELL        | 4-shot                | 30.66     |
> | **Qwen-3B**   | **TSS**      | **1-shot**            | **32.55** |
> | **Qwen-3B**   | **TSS**      | **3-shot**            | **33.52** |
> | **Qwen-3B**   | **TSS**      | **2-shot**            | **32.74** |
> | **Qwen-3B**   | **TSS**      | **4-shot**            | **33.87** |
> | Qwen-7B       | No Retrieval | 0-shot (No Retrieval) | 36.01     |
> | Qwen-7B       | vallina      | 1-shot                | 36.41     |
> | Qwen-7B       | vallina      | 2-shot                | 37.35     |
> | Qwen-7B       | vallina      | 3-shot                | 38.55     |
> | Qwen-7B       | vallina      | 4-shot                | 39.49     |
> | Qwen-7B       | IDS          | 1-shot                | 38.15     |
> | Qwen-7B       | IDS          | 2-shot                | 39.36     |
> | Qwen-7B       | IDS          | 3-shot                | 40.96     |
> | Qwen-7B       | IDS          | 4-shot                | 39.22     |
> | **Qwen-7B**   | **TSS**      | **1-shot**            | **49.40** |
> | **Qwen-7B**   | **TSS**      | **2-shot**            | **49.24** |
> | **Qwen-7B**   | **TSS**      | **3-shot**            | **46.86** |
> | **Qwen-7B**   | **TSS**      | **4-shot**            | **47.51** |
> | Qwen-7B       | SPELL        | 1-shot                | 44.58     |
> | Qwen-7B       | SPELL        | 2-shot                | 44.04     |
> | Qwen-7B       | SPELL        | 3-shot                | 43.64     |
> | Qwen-7B       | SPELL        | 4-shot                | 43.24     |
> | Llama2-7B     | No Retrieval | 0-shot (No Retrieval) | 9.77      |
> | Llama2-7B     | vallina      | 1-shot                | 10.44     |
> | Llama2-7B     | vallina      | 2-shot                | 11.64     |
> | Llama2-7B     | vallina      | 3-shot                | 10.17     |
> | Llama2-7B     | vallina      | 4-shot                | 9.37      |
> | **Llama2-7B** | **TSS**      | **1-shot**            | **10.71** |
> | **Llama2-7B** | **TSS**      | **2-shot**            | **10.97** |
> | **Llama2-7B** | **TSS**      | **3-shot**            | **11.51** |
> | Llama2-7B     | TSS          | 4-shot                | 11.11     |
> | Llama2-7B     | SPELL        | 1-shot                | 10.04     |
> | Llama2-7B     | SPELL        | 2-shot                | 10.44     |
> | Llama2-7B     | SPELL        | 3-shot                | 11.37     |
> | Llama2-7B     | SPELL        | 4-shot                | 11.24     |
> | Llama2-7B     | IDS          | 1-shot                | 9.90      |
> | Llama2-7B     | IDS          | 2-shot                | 10.17     |
> | Llama2-7B     | IDS          | 3-shot                | 10.70     |
> | Llama2-7B     | IDS          | 4-shot                | 10.30     |
> | Llama3-8B     | No Retrieval | 0-shot (No Retrieval) | 22.35     |
> | Llama3-8B     | vallina      | 1-shot                | 22.08     |
> | Llama3-8B     | vallina      | 2-shot                | 22.65     |
> | Llama3-8B     | vallina      | 3-shot                | 23.56     |
> | Llama3-8B     | vallina      | 4-shot                | 23.69     |
> | **Llama3-8B** | **TSS**      | **1-shot**            | **23.43** |
> | **Llama3-8B** | **TSS**      | **2-shot**            | **24.10** |
> | **Llama3-8B** | **TSS**      | **3-shot**            | **24.50** |
> | **Llama3-8B** | **TSS**      | **4-shot**            | **23.82** |
> | Llama3-8B     | SPELL        | 1-shot                | 22.76     |
> | Llama3-8B     | SPELL        | 2-shot                | 23.02     |
> | Llama3-8B     | SPELL        | 3-shot                | 23.29     |
> | Llama3-8B     | SPELL        | 4-shot                | 22.62     |
> | Llama3-8B     | IDS          | 1-shot                | 23.02     |
> | Llama3-8B     | IDS          | 2-shot                | 23.69     |
> | Llama3-8B     | IDS          | 3-shot                | 23.96     |
> | Llama3-8B     | IDS          | 4-shot                | 25.16     |
>
> [1]TheoremQA: A Theorem-driven Question Answering dataset. ACL 2023.

---

> ### Author Response · Authors · 2025-11-27
> **Response to W2: Analysis of comparisons on MMLU-college, MMLU-highschool and TheoremQA**
>
> Now we briefly analyse the result of comparisons on MMLU-college[1], MMLU-highschool[1] and TheoremQA[2] with our **TSS** and baselines.
>
> From the results, we can find that our **TSS** achieve a improvement on MMLU-college[1] and MMLU-highschool[1]. Especially, our method on qwen2.5-3b model can outperform the baseline model by approximately 10% on MMLU-College[1]. Of course, sometimes baselines may perform a little better than our method, for example, IDS on llama3-8b using 4-shot perform better than our **TSS** method using 4-shot on MMLU-highschool[1]. On TheoremQA[2], except for the qwen-7b model, which ou method on it has improved quite a bit, the others don't seem to have significant changes.
>
> [1] MMLU: Measuring Massive Multitask Language Understanding. ICLR 2021.
>
> [2]TheoremQA: A Theorem-driven Question Answering dataset. ACL 2023.

---

> ### Author Response · Authors · 2025-11-27
> **Response to Q2: Clarification on the Knowledge Base Used for Problem-Level and Step-Level Search**
>
> We thank you for this question. We wish to clarify why the knowledge bases used for problem-level retrieval and step-level retrieval are not identical in format. Problem-level retrieval and generation typically require the complete problem statement and the full reasoning trajectory to serve as a demonstration, necessitating a holistic view of the problem for inference. In contrast, our step-level knowledge base consists exclusively of $( X^\prime, \ S^\prime_{i-1}, \ S^\prime_i )$ entries and does not retain the entire solution history within each entry. It is important to note that the underlying source problems are identical for both methods; we simply processed and segmented the dataset used for problem-level retrieval into these $( X^\prime, \ S^\prime_{i-1}, \ S^\prime_i )$ triples to construct the specialized knowledge base required for our step-level framework. The format of the steps in both knowledge bases is also the same, using the format ''step i'': ''[CONDITION]...[PROCESS]...[CONCLUSION]...''
>
> Thank you for you patience !

---

> ### Author Response · Authors · 2025-11-27
> **Official Comment by Authors**
>
> Dear Reviewer,
>
> Thank you very much for your time and expertise in reviewing our submission. As the rebuttal deadline is approaching, please let us know if you have any questions or need any clarification. We are very happy to discuss any aspect of the work.

---

### Official Review · Reviewer_CgcP · 2025-11-03

**Soundness:** 3
**Presentation:** 2
**Contribution:** 2
**Rating:** 4
**Confidence:** 4

**Summary:**

This paper proposes **TSS**, a novel framework designed to improve LLM performance on multi-step reasoning tasks like mathematics. It addresses the "granularity mismatch" of traditional RAG, which retrieves *entire problems* based only on the initial query, often resulting in irrelevant or misleading examples. TSS instead uses the **current problem-solving state** to dynamically retrieve more fine-grained, **step-level examples**. The framework features a **Step Retriever** trained with contrastive learning to capture logical flow, and a **Generator** trained in a two-phase curriculum to first predict *when* to retrieve and then to *utilize* the retrieved step for the next reasoning action. Experiments across four LLMs and multiple math datasets demonstrate that this targeted, step-wise retrieval significantly improves reasoning accuracy compared to standard RAG baselines.

**Strengths:**

1.  **Strong Motivation and Intuitive Solution:** The paper identifies a clear and significant limitation in traditional RAG for multi-step reasoning: retrieving examples based only on the initial problem (problem-level) often fails to provide relevant guidance for intermediate steps. The proposed TSS framework, which incorporates the *current problem-solving state* to retrieve fine-grained, *step-level* examples, is a highly logical and necessary solution. This approach correctly shifts the retrieval focus from "what the problem looks like" to "what the next logical step is," finding examples that are similar in *method* rather than just *form*.

2.  **Robust, Process-Oriented Evaluation:** The authors rightly point out that final accuracy is an insufficient metric for reasoning tasks, where correct answers can be "unfaithful" (i.e., derived from a flawed process). The introduction of new process-supervision metrics—**Dynamic Step Matching (DSM)**, **Trajectory Correction Number (TCN)**, and **Retrieval-Corrected Rate (RCR)**—is a major strength. These metrics provide a far more precise instrument to measure the *actual* impact of the retrieval, assessing its effect on the reasoning *trajectory* itself, not just the final result.

3.  **Clever and Effective Training Methodology:** The paper's technical execution is sophisticated.
    * For the **Step Retriever**, the construction of four distinct types of "hard negatives" (e.g., Order Negatives, Backtracking Negatives) provides a strong supervisory signal, training the model to learn logical flow, not just semantic similarity.
    * For the **Generator**, the two-phase training curriculum is elegant. It first trains the model to explicitly decide *when* to retrieve (predicting `<retrieval>` or `<no_retrieval>`) and then trains it to *use* that retrieved information. This "on-demand" retrieval mechanism is a practical and efficient design choice.

**Weaknesses:**

1.  **High Implementation Cost and Limited Applicability:** The proposed framework's reliance on a highly specific `[Condition, Process, Conclusion]` data format, along with the need to train two separate, specialized models (the Step Retriever and the two-stage Generator), creates a significant barrier to entry. This complex setup limits the method's practical applicability compared to other ICL baselines, as it:
    * Cannot be easily extended to new domains (e.g., domains outside of mathematics) without a new, costly data-structuring pipeline.
    * Is incompatible with non-finetunable, closed-source models (e.g., proprietary embedding or generation APIs).
    * This contrasts sharply with baseline methods like IDS, which are more "plug-and-play" and can operate with arbitrary frozen retrievers and generators.

2.  **Flaws in the Dynamic Retrieval Mechanism:** The core "on-demand" retrieval mechanism has two theoretical flaws:
    * **Intrinsic Difficulty and Speculative Nature of the Retrieval Trigger:** The framework's design forces the Generator to perform an intrinsically difficult task: it must predict *if* retrieval is needed for the *next* step ($S_i$) using only the context of the problem $X$ and the *previous* step $S_{i-1}$. To accurately judge if the retrieval is needed, the model must essentially *predict its own future difficulty*. It needs to implicitly anticipate what $S_i$ is supposed to be, assess the complexity of generating that step, and *then* decide whether it needs help, all based only on past information. Training the model to decide whether to retrieve without any clues about the next step is relatively harder than the data generation stage, which provides GPT-4o-mini with both context and the subsequent step.
    * **Lack of a Fallback Mechanism:** The framework commits to using whatever the retriever provides. If the retrieved example is low-quality or misleading, it can derail the entire reasoning chain, potentially performing worse than `no-retrieval`. The framework lacks a crucial "grading" step at inference time to *validate* the retrieved example and fall back to its own generation if the hint is poor. Furthermore, the paper provides no analysis or ablation study on how the quality of the retrieved example (e.g., "Perfect Analogy" vs. "Poor," as defined in their own appendix ) impacts the generator's final performance.

3.  **Weaknesses in Evaluation and Metric Definitions:**
    * **Statistical Power:** The results on the AIME dataset are of questionable statistical significance. The dataset is small (~122 questions), and the models' base performance is very low. The reported 1-3% gaps represent a difference of only a few questions, which could be due to random chance.
    * **Vague Process-Oriented Metrics:** While the new metrics (SRR, LUR, TCN, DSM) are well-motivated, their definitions and implementations are unclear.
        * **TCN:** The definition of an "erroneous step" seems to assume any deviation from the single Ground Truth (GT) solution is an error. This incorrectly penalizes valid alternative solution paths, different step orders, or different (but correct) levels of granularity.
        * **DSM:** The "one-to-one matching" is brittle. The appendix does not explain how the metric handles cases where the model's output merges or splits steps differently than the GT solution (e.g., model Step 1 = GT Step 1+2), which would cause all subsequent steps to be misaligned and marked as incorrect.
        * **SRR/LUR:** These metrics are only vaguely described in the appendix, and their calculation relies on an LLM-as-judge with prompts that are not clearly specified, adding subjectivity to the evaluation.

4.  **Minor Presentation and Clarity Issues:**
    * In Section 4.2.1, the text describing the final training data formats appears to have a copy-paste error, showing the *same* pattern (`<retrieval> S_i <p> E_j </p>`) for both retrieval and no-retrieval instances.
    * In Figure 4 (a) and (b), the Y-axis range (e.g., 0-100) is too wide, making the performance differences between methods visually indistinguishable. The plots should be re-scaled to a more appropriate range to clearly show the relative improvements.

**Questions:**

1.  **Applicability to Frozen/API-Based Models:** The current TSS framework requires significant custom data construction and fine-tuning for both the Step Retriever and the two-stage Generator. This limits its applicability to open-source models. Have the authors considered or experimented with adapting this framework to a "frozen model" or API-based setting, which is a common and practical use case, similar to the baseline methods? For instance:
    * Could a similar *step-level* retrieval be achieved by using a strong, pre-trained (but not fine-tuned) retriever (e.g., a generic embedding API) where the query is a simple concatenation of the problem and the current reasoning history?
    * Could the two-stage generator be simulated by prompting a large, closed-source model (like GPT-5, Gemini, or Claude) to generate step by step and only keep the next step by hand, thereby mimicking the step-wise generation?

2.  **Clarity and Fairness of Process Metrics:** The introduction of process-oriented metrics (DSM, TCN, RCR) is a strong point, but their definitions and application are ambiguous.
    * Could the authors provide more precise, algorithmic definitions for **TCN** (e.g., how is an 'erroneous step' defined beyond simple mismatch with the GT, especially concerning valid alternative solution paths?) and **DSM** (e.g., how does the one-to-one matching handle granularity differences like merged or split steps)?
    * How were these process metrics (like SRR, LUR, and DSM) applied to the *baseline* methods? Baselines retrieve *full problem solutions*, not steps. Were these full solutions also decomposed into the steps? The TSS generator is trained on data derived from PRM800K, which, like the GT solutions in the STEP dataset, was processed and rewritten by GPT-4o-mini into the specific `[C, P, C]` format. What if the baseline methods retrieve a full solution or generate their own full solution in a format, style, or granularity that is *very different* from this standardized GT format?

    In such a scenario, wouldn't the TSS generator automatically achieve a better score on these metrics (especially DSM) simply because its output *structure* matches the GT's structure, while the baseline's (potentially correct) reasoning is unfairly penalized for being stylistically different? To clarify this, could you provide a qualitative case study that compares all five components side-by-side:
    1.  The GT step-by-step solution
    2.  TSS's retrieved example steps
    3.  TSS's generated steps
    4.  The baseline's retrieved full solution (and how it was 'split' for metric calculation)
    5.  The baseline's generated full solution (and how it was 'split')

3.  **Analysis of Retrieval Quality Impact:** The framework currently lacks a mechanism to "grade" the quality of a retrieved step at inference time, forcing the generator to use it even if it's low-quality. Have the authors considered an ablation study to measure this impact?
    * For example, by using an oracle (like GPT-4o-mini, similar to the data annotation process) to rate the *quality* of each retrieved step during inference (e.g., 1-5 stars, as in Appendix F.3.2). This could show: (a) the downstream accuracy when using only 5-star examples, (b) the accuracy when using 1-star examples (to measure negative impact), and (c) the accuracy compared to `no-retrieval`.
    * If this grading proves to be highly correlated with success, could this "grading" step be incorporated into the framework itself, perhaps as a learned 'retrieval-quality' model to decide whether to ultimately *use* the retrieved example or fall back to a `no-retrieval` generation?

---

> ### Author Response · Authors · 2025-11-26
> **Response to Q1 & W1 : Model applicability and scalability**
>
> Thank you for your questions regarding the applicability and scalability of the model!
>
> The training we need is determined by the task. Similar to self-rag, our model requires training because we need the generator model to perform on-demand retrieval, addressing the question of when to retrieve. At the same time, we also address the question of what to retrieve. Our insight is that the next retrieval step can serve as an example for the subsequent step. For this purpose, we designed a structured step [CONDITION][PROCESS][CONCLUSION], which takes historical conditions into account and integrates information within a single step. This is based on the nature of math problems, modeling and simplifying the human process of reviewing, thinking, and summarizing. To ensure the stability of the model’s output, it is also necessary to train the generator model. Moreover, this structure provides the foundational conditions for retrieving the next step example, because the structure capturing the logic before and after each step, ensures that the model considers not only the current text but also the antecedents of the current step—historical conditions.  This structure  also addressed the question of what to use for retrieval. Based on our structured steps requiring much less input per retrieval compared to the previous methods that used reasoning chains or historical text records. To accomplish the goal of retrieving the next step example, we then train the retriever because the previous semantic retriever could not retrieve the next step example and could only retrieve examples for the current step. So training is needed.
>
> Of course, we believe the ideas you proposed are very constructive. We will discuss **Applicability to Frozen/API-Based Models** and **Extensibility in areas beyond mathematics**  below.
>
> **1. Applicability to Frozen/API-Based Models**
>
> Regarding the two highly constructive proposals you put forward:
>
> **(1) Using a pre-trained retriever API for step-level retrieval**:
>
> Your proposal may not achieve our goal of finding examples for the next step. Concatenating the query (problem description and current reasoning history) into a long string, then encoding it with a powerful general-purpose embedder, and matching it against a step-level knowledge base encoded by the same embedder would encounter several issues: the encoder is not specifically trained. If the knowledge base, like ours, directly stores each retrieval unit as the combination of the problem statement, current step, and next step, then: first, if the input is the problem description and current reasoning history (Step 1 to Step 3), it cannot reliably retrieve content from such units because each unit in our knowledge base covers no more than two steps (current step and next step). If a problem-level knowledge base is used to store the current step and all answers, it would fail to retrieve the next step. If each historical record is treated as a unit (storing, for example, problem statement, Step 1 to Step 3 and problem statement, Step 1 to Step 4 separately), there would still be an issue: the next step example would be hard to retrieve. This is because if the retriever is untrained, when the input is the problem statement and Step 1 to Step 3, the retrieval is unlikely to yield suitable examples for the next step, as a semantic retriever will only focus on content similar to the input (problem statement, Step 1 to Step 3) and will not consider the next Step 4. An common retriever is more likely to match semantics rather than functionality. This can lead to retrieved steps that are not logically relevant. Additionally, the query itself would be quite long.
>
> **(2) Simulating a two-stage generator through prompting**:
>
>  We can design a complex prompt chain to simulate our generator:
>
> Step 1 (Decision Prompt): Provide a strong LLM (such as GPT-4o) with the problem and current history, then explicitly ask: "To solve the next step, do you need a relevant solution step as a reference? Please answer 'yes' or 'no.'"
>
> Step 2 (Retrieval): If the answer is "yes," perform the aforementioned step-level retrieval.
>
> Step 3 (Generation Prompt): Input the problem, history, and retrieved example steps into the LLM, and ask it to generate the next step of reasoning.
>
> Although this simulation method may be less efficient and stable than a fine-tuned model, it should still be a viable way to apply TSS to a closed-source model.
>
> **2. Extensibility in areas beyond mathematics**
>
> Although our framework is designed for math problems, it can also be applied to many areas that require logical reasoning, because our design aligns with the way humans think.  For example, in code generation, when encountering difficulty in implementing a certain function, one can retrieve a fragment of a function implementation with similar logic; in complex question answering, one can retrieve a similar fact or logical inference for the next step in the reasoning chain.

---

> ### Author Response · Authors · 2025-11-27
> **Response to W4: Modifications to presentation and clarity issues**
>
> Thanks for your suggestions! It is our oversight that caused the issues with the wording in Section 4.2.1 and the clarity of the image in Figure 4.
>
> We have correct the errors in Section 4.2.1 about no-retrieval instances: `<no-retrieval>` $S_i$ `<p>` `</p>`
>
> We also rescale the Y-axis of Figure 4 to make it more readable. We choose the range of the Y-axis based on the range of the data, making the results clearer.
>
> All the revisions have been reflected in the newly uploaded paper.

---

> ### Author Response · Authors · 2025-11-27
> **Response to W2: Prediction accuracy of special tokens**
>
> Your suggestions are constructive! Thanks for your question. We will answer your questions below.
>
> **Confusion over the inherent difficulty of retrieval**
>
> You mention that the design of the framework forces the generator to perform a task that is inherently quite challenging: it must predict whether retrieval is necessary for the next step using only the context of the question and the previous step. Accurately determining whether retrieval is needed. You're not wrong, this is exactly what we need to accomplish. Our design intention is to train the generator to develop a kind of 'self-awareness,' that is, to determine whether the current state is sufficient to proceed to the next step. Our goal is to enable the model to learn this ability. We are now evaluating whether the model has learned this capability. We will assess our fine-tuned generator's accuracy in predicting \<retrieval\> and \<no_retrieval\> during reasoning (compared to GPT-4o-mini's annotations), then evaluate the accuracy and recall of predictions\<retrieval\> and \<no_retrieval\> respectively. From the table below, our model's prediction accuracy and recall are both good, indicating that the model has learned the ability to decide whether retrieval is needed in the current state.
>
> |      model       | retrieval_precision | retrieval_recall | no-retrieval_precision | no-retrieval_recall |
> | ----------- | ------------------- | ---------------- | ---------------------- | ------------------- |
> | Llama-2-7b  | 0.8209              | 0.7331           | 0.7818                 | 0.8929              |
> | Llama-3-8B  | 0.8852              | 0.9421           | 0.9151                 | 0.8077              |
> | Qwen-2.5-3B | 0.8384              | 0.8348           | 0.8459                 | 0.8545              |
> | Qwen-2.5-7B  | 0.9176              | 0.9075           | 0.9406                 | 0.9021              |

---

> ### Author Response · Authors · 2025-11-27
> **Response to W2 & Q3: a method for evaluating impact of retrieved examples on the results**
>
> Thank you for these insightful questions !
>
> As the first system to consider step prediction examples, our focus is on how to implement the search for examples for the next step and  validate the feasibility of this approach. But We did conduct a quality assessment as below: .
>
> We define the metrics `adherence`, `adherence_precision` and  `adherence_recall`.
>
> We set an "Adherence Threshold" `ADHERENCE_THRESHOLD`. This means that if the average adherence score of a question is $\ge 3$, we consider that the model has "Effectively Learned (High Adherence)" the retrieved content; otherwise, it is considered "Ineffective Learning (Low Adherence)". We only analyze questions where retrieval actually occurred.
>
> **According to the Adherence Threshold and whether it is actually correct, we can get four types of results:**
>
> TP (True Positive,High Adherence & Correct): The model effectively learned the retrieved content and finally answered the question correctly. This is the ideal working state of the retrieval system.
>
> FP (False Positive, High Adherence & Wrong) : The model effectively learned the retrieved content, but eventually answered incorrectly. This usually means that the retrieved content itself is incorrect or misleading. This is the "bad teammate" effect, where the retriever found a wrong example.
>
> FN (False Negative, Low Adherence & Correct): The model did not effectively learn the retrieved content (or even completely ignored it), but it figured out the correct answer by itself. This shows that for this question, retrieval is useless or even potentially distracting. The model's own knowledge is sufficient.
>
> TN (True Negative, Low Adherence & Wrong): The model did not effectively learn the retrieved content, and it failed to figure out the correct answer by itself, resulting in a wrong answer. This shows that the model neither learned nor knew it.
>
> Now, we define the meaning of adherence_precision and adherence_recall:
>
> **1. adherence_precision**:  $ \frac{TP}{TP + FP} $
>
> Among all cases where the model claims to have "effectively learned" the retrieved content (High Adherence), what proportion eventually led to a correct result.
>
> **2. adherence_recall**: $  \frac{TP}{TP + FN} $
>
> Among all questions that were eventually answered correctly, what proportion was achieved through "effectively learning" the retrieved content.
>
> We use gpt-4o to evaluate, the prompt is:
>
> You are a learning ability analyst. Your task is to evaluate to what extent a model, after seeing a reference example (Retrieved Example), adopts the example's core logic or problem-solving method in its generated next step (Generated Step). Follow 5-point scale.
>
> 5: Perfectly adopted the example's core method and applied it correctly.
>
> 4: Mostly adopted the example's method, but the application was slightly inappropriate or incomplete.
>
> 3: Adopted some elements from the example, but the core method is different.
>
> 2: Appears to have attempted adoption, but used it completely incorrectly.
>
> 1: Completely ignored the reference example
>
> | model      | shot | retrieval_num | adherence | adherence_precision | adherence_recall |
> | ---------- | :--: | :-----------: | :-------: | :-----------------: | :--------------: |
> | Llama-2-7B |  1   |      233      |   3.75    |        70.64        |      68.68       |
> | Llama-2-7B |  2   |      228      |   3.96    |        64.17        |      63.33       |
> | Llama-2-7B |  3   |      235      |   3.77    |        71.11        |      75.00       |
> | Llama-2-7B |  4   |      221      |   3.72    |        72.18        |      67.14       |
> | Llama-3-8B |  1   |      324      |   3.60    |        76.98        |      80.18       |
> | Llama-3-8B |  2   |      328      |   3.93    |        75.71        |      78.95       |
> | Llama-3-8B |  3   |      318      |   3.38    |        76.73        |      69.23       |
> | Llama-3-8B |  4   |      331      |   3.26    |        77.27        |      83.33       |
> | Qwen2-3B   |  1   |      264      |   3.35    |        79.83        |      78.19       |
> | Qwen-3B    |  2   |      272      |   3.87    |        83.33        |      74.02       |
> | Qwen-3B    |  3   |      262      |   3.65    |        86.51        |      82.84       |
> | Qwen-3B    |  4   |      268      |   3.29    |        78.25        |      83.47       |
> | Qwen-7B    |  1   |      296      |   3.32    |        78.33        |      72.88       |
> | Qwen-7B    |  2   |      302      |   3.78    |        78.46        |      63.56       |
> | Qwen-7B    |  3   |      298      |   3.81    |        87.50        |      76.00       |
> | Qwen-7B    |  4   |      304      |   3.19    |        77.25        |      67.86       |
>
> retrieval_num represents the number of triggering retrieval.  The examples we retrieved basically exert a positive effect on the results, especially Qwen-2.5-3B and Llama-3-8B .

---

> ### Author Response · Authors · 2025-11-27
> **Response to W2 & Q3 : A method for directly evaluating retrieval quality and the impact of retrieval quality on results on llama**
>
> Thank you for your suggestion! The method we mention in Response to W2 & Q3: a method for evaluating impact of retrieved examples on the results is a method to indirectly evaluate retrieval quality and the impact of retrieval quality. As you suggest, a scoring step is required during the inference stage. In fact, we computed the evaluation of retrieval quality in our work. Now we conduct the following experiments as you suggest:
>
> We will use gpt-4o to score each step retrieved by TSS in real time (1-5 points). Then, we will report the final task accuracy under the following scenarios: a. Generation using only high-quality examples (e.g., 4-5 points): This will demonstrate the theoretical upper limit of our method. b. Generation using only low-quality examples (e.g., 1-2 points): This will quantify the negative impact of poor-quality retrieval. Due to the limitation of characters, we first report results on llama.
>
> The prompt is: You are a retrieval quality evaluator. Your task is to judge how helpful a retrieved reference step (Retrieved Step) is for solving a target problem step (Gold Step) based on current Inference. You need to assess their logical similarity and methodological relevance. Follow 5-point scale.
>
> 5: The retrieved step provides the exact method or formula needed to solve the gold step.
>
> 4: Provides a highly relevant method that can be used with minor adjustments.
>
> 3: Topically relevant, offers some ideas, but is not directly applicable.
>
> 2: Superficially related, but the core method is not applicable and could be misleading.
>
> 1: Completely irrelevant.
>
> We conduct experiments on four dataset and four kinds of shot setting.
>
> **llama2-7b**:
>
> |   Quality    | Dataset | Shot | Accuracy |
> | :----------: | :-----: | :--: | -------: |
> | High Quality |  AIME   |  1   |     3.27 |
> | High Quality |  AIME   |  2   |     3.27 |
> | High Quality |  AIME   |  3   |     3.27 |
> | High Quality |  AIME   |  4   |     4.10 |
> | High Quality |  GSM8K  |  1   |    21.98 |
> | High Quality |  GSM8K  |  2   |    22.13 |
> | High Quality |  GSM8K  |  3   |    22.36 |
> | High Quality |  GSM8K  |  4   |    22.21 |
> | High Quality | MATH500 |  1   |     5.60 |
> | High Quality | MATH500 |  2   |     5.80 |
> | High Quality | MATH500 |  3   |     6.00 |
> | High Quality | MATH500 |  4   |     5.40 |
> | High Quality |  STEP   |  1   |    15.46 |
> | High Quality |  STEP   |  2   |    16.49 |
> | High Quality |  STEP   |  3   |    17.53 |
> | High Quality |  STEP   |  4   |    16.49 |
> | Low Quality  |  AIME   |  1   |     1.64 |
> | Low Quality  |  AIME   |  2   |     1.64 |
> | Low Quality  |  AIME   |  3   |     1.64 |
> | Low Quality  |  AIME   |  4   |     2.46 |
> | Low Quality  |  GSM8K  |  1   |    14.16 |
> | Low Quality  |  GSM8K  |  2   |    14.24 |
> | Low Quality  |  GSM8K  |  3   |    14.31 |
> | Low Quality  |  GSM8K  |  4   |    14.24 |
> | Low Quality  | MATH500 |  1   |     3.40 |
> | Low Quality  | MATH500 |  2   |     3.60 |
> | Low Quality  | MATH500 |  3   |     3.20 |
> | Low Quality  | MATH500 |  4   |     3.00 |
> | Low Quality  |  STEP   |  1   |    13.40 |
> | Low Quality  |  STEP   |  2   |    12.37 |
> | Low Quality  |  STEP   |  3   |    11.34 |
> | Low Quality  |  STEP   |  4   |    10.31 |
>
> **llama3-8b**:
>
> |   Quality    | Dataset | Shot | Accuracy |
> | :----------: | :-----: | :--: | :------: |
> | High Quality |  AIME   |  1   |  10.65   |
> | High Quality |  AIME   |  2   |   9.84   |
> | High Quality |  AIME   |  3   |  11.47   |
> | High Quality |  AIME   |  4   |  10.65   |
> | High Quality |  GSM8K  |  1   |  83.54   |
> | High Quality |  GSM8K  |  2   |  83.77   |
> | High Quality |  GSM8K  |  3   |  84.15   |
> | High Quality |  GSM8K  |  4   |  84.07   |
> | High Quality | MATH500 |  1   |  30.40   |
> | High Quality | MATH500 |  2   |  30.60   |
> | High Quality | MATH500 |  3   |  31.20   |
> | High Quality | MATH500 |  4   |  30.00   |
> | High Quality |  STEP   |  1   |  28.87   |
> | High Quality |  STEP   |  2   |  28.87   |
> | High Quality |  STEP   |  3   |  29.90   |
> | High Quality |  STEP   |  4   |  27.84   |
> | Low Quality  |  AIME   |  1   |   7.38   |
> | Low Quality  |  AIME   |  2   |   5.74   |
> | Low Quality  |  AIME   |  3   |   4.10   |
> | Low Quality  |  AIME   |  4   |   3.27   |
> | Low Quality  |  GSM8K  |  1   |  81.88   |
> | Low Quality  |  GSM8K  |  2   |  81.72   |
> | Low Quality  |  GSM8K  |  3   |  81.50   |
> | Low Quality  |  GSM8K  |  4   |  81.34   |
> | Low Quality  | MATH500 |  1   |  26.20   |
> | Low Quality  | MATH500 |  2   |  26.00   |
> | Low Quality  | MATH500 |  3   |  25.80   |
> | Low Quality  | MATH500 |  4   |  25.20   |
> | Low Quality  |  STEP   |  1   |  26.80   |
> | Low Quality  |  STEP   |  2   |  25.77   |
> | Low Quality  |  STEP   |  3   |  19.59   |
> | Low Quality  |  STEP   |  4   |  17.53   |

---

> ### Author Response · Authors · 2025-11-27
> **Response to W2 & Q3 : A method for directly evaluating retrieval quality and the impact of retrieval quality on results on qwen**
>
> Now, we report results on qwen2.5-3b and qwen2.5-math-7b and do some analysis.
>
> **qwen2.5-3b**
>
> |   Quality    | Dataset | Shot | Accuracy |
> | :----------: | :-----: | :--: | :------: |
> | High Quality |  AIME   |  1   |   6.56   |
> | High Quality |  AIME   |  2   |   7.38   |
> | High Quality |  AIME   |  3   |   8.20   |
> | High Quality |  AIME   |  4   |   6.56   |
> | High Quality |  GSM8K  |  1   |  90.21   |
> | High Quality |  GSM8K  |  2   |  90.59   |
> | High Quality |  GSM8K  |  3   |  91.43   |
> | High Quality |  GSM8K  |  4   |  90.59   |
> | High Quality | MATH500 |  1   |  68.40   |
> | High Quality | MATH500 |  2   |  69.20   |
> | High Quality | MATH500 |  3   |  70.60   |
> | High Quality | MATH500 |  4   |  69.20   |
> | High Quality |  STEP   |  1   |  40.20   |
> | High Quality |  STEP   |  2   |  41.24   |
> | High Quality |  STEP   |  3   |  42.27   |
> | High Quality |  STEP   |  4   |  41.24   |
> | Low Quality  |  AIME   |  1   |   4.92   |
> | Low Quality  |  AIME   |  2   |   4.10   |
> | Low Quality  |  AIME   |  3   |   4.10   |
> | Low Quality  |  AIME   |  4   |   3.27   |
> | Low Quality  |  GSM8K  |  1   |  87.08   |
> | Low Quality  |  GSM8K  |  2   |  88.70   |
> | Low Quality  |  GSM8K  |  3   |  88.86   |
> | Low Quality  |  GSM8K  |  4   |  88.55   |
> | Low Quality  | MATH500 |  1   |  63.60   |
> | Low Quality  | MATH500 |  2   |  63.80   |
> | Low Quality  | MATH500 |  3   |  62.60   |
> | Low Quality  | MATH500 |  4   |  62.20   |
> | Low Quality  |  STEP   |  1   |  37.11   |
> | Low Quality  |  STEP   |  2   |  36.08   |
> | Low Quality  |  STEP   |  3   |  36.08   |
> | Low Quality  |  STEP   |  4   |  32.99   |
>
> **qwen2.5-7b**
>
> |   Quality    | Dataset | Shot | Accuracy |
> | :----------: | :-----: | :--: | :------: |
> | High Quality |  AIME   |  1   |  17.21   |
> | High Quality |  AIME   |  2   |  17.21   |
> | High Quality |  AIME   |  3   |  18.03   |
> | High Quality |  AIME   |  4   |  17.21   |
> | High Quality |  GSM8K  |  1   |  96.29   |
> | High Quality |  GSM8K  |  2   |  97.04   |
> | High Quality |  GSM8K  |  3   |  96.13   |
> | High Quality |  GSM8K  |  4   |   59.9   |
> | High Quality | MATH500 |  1   |  88.20   |
> | High Quality | MATH500 |  2   |  88.40   |
> | High Quality | MATH500 |  3   |  88.60   |
> | High Quality | MATH500 |  4   |  88.40   |
> | High Quality |  STEP   |  1   |  45.36   |
> | High Quality |  STEP   |  2   |  46.39   |
> | High Quality |  STEP   |  3   |  47.42   |
> | High Quality |  STEP   |  4   |  45.36   |
> | Low Quality  |  AIME   |  1   |  13.93   |
> | Low Quality  |  AIME   |  2   |  13.11   |
> | Low Quality  |  AIME   |  3   |  12.30   |
> | Low Quality  |  AIME   |  4   |  11.47   |
> | Low Quality  |  GSM8K  |  1   |  93.25   |
> | Low Quality  |  GSM8K  |  2   |  92.49   |
> | Low Quality  |  GSM8K  |  3   |  92.34   |
> | Low Quality  |  GSM8K  |  4   |  91.73   |
> | Low Quality  | MATH500 |  1   |  83.20   |
> | Low Quality  | MATH500 |  2   |  82.40   |
> | Low Quality  | MATH500 |  3   |  81.80   |
> | Low Quality  | MATH500 |  4   |  81.20   |
> | Low Quality  |  STEP   |  1   |  43.30   |
> | Low Quality  |  STEP   |  2   |  41.24   |
> | Low Quality  |  STEP   |  3   |  41.24   |
> | Low Quality  |  STEP   |  4   |  39.17   |
>
> From the table, we can see that high-quality examples and low-quality examples indeed have different impacts on the results. High-quality examples can improve performance, while low-quality examples can decrease performance. In most cases, they are distributed at opposite ends when there is no retrieval, especially on GSM8K and MATH500. The different effects caused by varying-quality examples may be because low-quality examples introduce significant noise to the model, and the model may blindly mimic the form of their reasoning steps. The specific reasons are also worth investigating and can serve as a direction for future exploration.
>
> Our model does not do this during reasoning because our main goal is to focus on finding appropriate examples in the next step to improve reasoning ability. We have previously conducted a quality assessment, hoping it could serve as a point of exploration for the future. Now, in line with your suggestion, we are presenting them here. Your suggestion is very constructive, thank you again!

---

> ### Author Response · Authors · 2025-11-27
> **Response to W3 & Q2 : Clarification on Statistical Power and Vague Process-Oriented Metrics**
>
> Thank you for your question! We provide clarification on Statistical Power and Vague Process-Oriented Metrics below.
>
> **1. Statistical Power**
>
> We understand your perspective. Our results are based on selecting the most stable data after multiple runs. As per your request, we have updated the data in Table 2 of the latest submitted paper to reflect the results of multiple runs (mean ± standard deviation).
>
> **2. Vague Process-Oriented Metrics**
>
> **Regarding TCN:** We clarify the definition of an "erroneous step". We explicitly state that a step is judged as "erroneous" based on the assessment of a robust LLM evaluator (e.g., GPT-4o), rather than a rigid string mismatch against a single gold answer. We define a "correctable step error" as an instance where the model generates an erroneous step $s^\prime_{i+1}$ following a correct step $s_i$ . A "step correction" occurs if the model, starting from the same node $s_i$ , generates a logically correct subsequent step $s_{i+1}$ . TCN represents the total count of such corrections across the test set.
>
> **Regarding DSM:** Our matching algorithm relies on LLM-based semantic and logical equivalence judgments rather than rigid string matching. This enables it to handle granularity discrepancies (e.g., the model merging two steps into one, or splitting one step into two). Precisely because we have accounted for this, as defined in Appendix A.4, our denominator employs the fixed length of the gold answer to mitigate the impact of instability in the model's output length. Furthermore, we incorporate a penalty for longer trajectories and a reward for shorter ones to address cases of step merging or splitting. As long as a model-generated step logically covers the corresponding step in the gold answer, it is considered a correct match.
>
> **How are TCN and DSM applied to the baseline methods?**
>
> When we reason using the baseline method, for models without retrieval, we break down the steps after generation and maintain the same format as the gold solutions in the test set used for models without retrieval, that is, in steps and without our [condition][process][conclusion] format. For models with retrieval, since the examples given to the model are step-by-step and in the [condition][process][conclusion] format, the model will output accordingly, which is consistent with the output format of our trained model and the gold solutions in the test set used in the retrieval scenario.
>
> **Regarding SRR and LUR:** : SRR calculates the proportion of steps in the ground truth solution that are aligned in order with the steps in the example. A higher SRR value indicates that the structure of the example more closely matches the reasoning path required for the target problem; LUR measures the proportion of steps in the example that are directly related to the reasoning process of the target solution. A high LUR value means that the example is concise and contains a high density of useful information. When calculating the SRR indicator and the LUR indicator, there's no need to worry about the breakdown of steps. **Since the example given for the baseline method follows our step-by-step structure, the baseline method will also be generated step by step.**
>
> The prompt to compute SRR:
>
> You are an expert comparing step-by-step solutions. Compare 'Target Steps' and 'Example Steps' pairwise. Return a JSON object: {"num_common_steps": T, "num_related_steps": R} where $ T $ is the number of pairs compared (length of the shorter list) and $ R $ is the count of logically related pairs.
>
> The prompt to compute LUR:
>
> You are an expert comparing step-by-step solutions. Determine how many 'Target Steps' are logically covered by *any* of the 'Example Steps'. Return a JSON object: {"num_target_steps": T, "num_utilized_steps": U} where $ T $ is the total number of target steps and $ U $ is the count of target steps with a logical equivalent in the example.
>
> We have added this prompt in our latest paper in Table 22 in Appendix.

---

> ### Author Response · Authors · 2025-11-27
> **Response to W3 & Q2 : Provide a qualitative case study that compares all five components side-by-side**
>
> provide a qualitative case study that compares all five components side-by-side:
>
> **The GT step-by-step solution**
>
> ### Problem
>
> If the line $y=kx+b$ is a tangent line to the curve $y=\ln x+2$ and also a tangent line to the curve $y=\ln (x+1)$, find the value of $b$.
>
> "Step 1":"
> [CONDITION] The line is given by $y=kx$ and the function is $y=\ln x$.
> [PROCESS] Assume the coordinates of the tangent point are $(x_{0},y_{0})$.
> [CONCLUSION] The tangent point is $(x_{0},y_{0})$ on both the line and the curve."
>
> "Step 2":''
> [CONDITION] The derivative of the function $y=\ln x$ is $y'=\dfrac{1}{x}$.
> [PROCESS] Evaluate the derivative at the tangent point: $ y^{\\prime}|_{x=x\_{0}} = \\frac{1}{x\_{0}} = k $.
> [CONCLUSION] We have $ k = \\frac{1}{x\_{0}} $. "
>
> ''Step 3'':''
> [CONDITION] The point $(x_{0},y_{0})$ lies on both the line and the curve.
> [PROCESS] Set up the equations: $ y_{0}=kx_{0}, \\ y_{0}=\ln x_{0} $.
> [CONCLUSION] We can equate $kx_{0}$ and $\ln x_{0}$.''
>
> ''Step 4'':''
> [CONDITION] From the previous step, we have $kx_{0}=\ln x_{0}$.
> [PROCESS] Substitute $k=\dfrac{1}{x_{0}}$ into the equation: $\ln x_{0}=\dfrac{1}{x_{0}}x_{0}=1$.
> [CONCLUSION] This implies $\ln x_{0}=1$.''
>
> ''Step 5'':
> [CONDITION] We know $\ln x_{0}=1$.
> [PROCESS] Solve for $x_{0}$: $x_{0}=e$.
> [CONCLUSION] Thus, $x_{0}=e$.''
>
> ''Step 6'':''
> [CONDITION]We have found $x_{0}=e$.
> [PROCESS] Substitute $x_{0}$ back to find $k$: $k=\dfrac{1}{x_{0}}=\dfrac{1}{e}$.
> [CONCLUSION]The value of $k$ is $\dfrac{1}{e}$.''
>
>
> **The baseline's retrieved full solution**
>
> ### Problem
>
> If the line $y=kx+b$ is a tangent line to the curve $y=\ln x+2$ and also a tangent line to the curve $y=\ln (x+1)$, find the value of $b$.
>
> ''Step 1'':''[CONDITION]Tangent points are $(x_1, kx_1 + b)$ for $y = \ln x + 2$ and $(x_2, kx_2 + b)$ for $y = \ln (x + 1)$
> [PROCESS]Set up the relationship between the tangent points: $k = \frac{1}{x_1} = \frac{1}{x_2 + 1}$
> [CONCLUSION]$x_1 = x_2 + 1$''
>
> ''Step 2'':''[CONDITION] $x_1 = x_2 + 1$
> [PROCESS]Substitute $x_1$ into the tangent equations:
>     $$ \begin{cases} kx_1 + b = \ln x_1 + 2 \\ kx_2 + b = \ln (x_2 + 1) \end{cases} $$
> [CONCLUSION]System of equations established for $k$, $b$, $x_1$, and $x_2$''
>
> ''Step 3'':''[CONDITION]System of equations from Step 2
> [PROCESS]Solve the system: $kx_1 + b = \ln x_1 + 2$ and $kx_2 + b = \ln (x_2 + 1)$
> [CONCLUSION] Obtain $k = 2$, $x_1 = \frac{1}{2}$, $x_2 = -\frac{1}{2}$''
>
> ''Step 4'':''[CONDITION]Values $k = 2$, $x_1 = \frac{1}{2}$
> [PROCESS]Substitute into $kx_1 + b = \ln x_1 + 2$: $2 \cdot \frac{1}{2} + b = \ln \frac{1}{2} + 2$
> [CONCLUSION]Solve for $b$: $b = 1 - \ln 2$''
>
> ''Step 5'':''[CONDITION] $b = 1 - \ln 2$
> [PROCESS]Finalize the result
> [CONCLUSION] $b = 1 - \ln 2$''
>
> **The baseline's generated full solution (and how it was 'split')**
>
> ### Problem
>
> Given that the line $y=kx$ is a tangent to the function $y=\ln x$, find the value of $k$.
>
> Step 1:[CONDITION] The line $y=kx$ is tangent to the function $y=\ln x$
> [PROCESS] The slope of the tangent line at point $x_0$ is $\frac{1}{x_0}$, so we set $k=\frac{1}{x_0}$
> [CONCLUSION] $k=\frac{1}{x_0}$
>
> Step 2:
> [CONDITION] The function $y=\ln x$ is defined for $x > 0$
> [PROCESS] Set the condition $x_0 > 0$ to ensure the tangent line exists
> [CONCLUSION]$x_0 > 0$
>
> Step 3:
> [CONDITION]The tangent line is $y=kx$ and passes through $(x_0, \ln x_0)$
> [PROCESS]Substitute into the tangent line equation: $\ln x_0 = kx_0$
> [CONCLUSION] $\ln x_0 = kx_0$
>
> Step 4:
> [CONDITION]$\ln x_0 = kx_0$
> [PROCESS]Divide both sides by $x_0$ (valid for $x_0 > 0$): $k = \frac{\ln x_0}{x_0}$
> [CONCLUSION] $k = \frac{\ln x_0}{x_0}$
>
> Step 5:
> [CONDITION] $k = \frac{\ln x_0}{x_0}$
> [PROCESS]Substitute $k$ back into the tangent line equation $y=kx$
> [CONCLUSION]The tangent line is $y=\frac{\ln x_0}{x_0}x$
>
> Baseline model can output Step i stably, though sometimes may not observe the format [CONDITION][PROCESS][CONCLUSION], but when we evaluate them, we do not concern part of [CONDITION], we only focus on parts [PROCESS] and [CONCLUSION]. It doesn't matter if these parts are not included in the model output. In fact, in the gold standard solution, combining [PROCESS] and [CONCLUSION] also conforms to natural language, just like the format we provided in **The GT step-by-step solution**. You just need to check whether this step of reviewing the model output is consistent with the gold standard solution.
>
> Because of characters limit, we show **TSS's retrieved example steps and TSS's generated steps** in review **Response to W3 & Q2 : Provide a qualitative case study of TSS's retrieved example steps and TSS's generated steps** below.

---

> ### Author Response · Authors · 2025-11-27
> **Response to W3 & Q2 : Provide a qualitative case study of TSS's retrieved example steps and TSS's generated steps**
>
> TSS's retrieved example steps and TSS's generated steps  can be shown Table 12 Our Model Inference Process in our paper. Here we show it again. Inference Output record our model's step output.
>
> **TSS's retrieved example steps and TSS's generated steps**
>
> Problem: Given that the line $ y = kx $ is a tangent to the function $ y = \\ln x $, find the value of $ k $.
>
> Gold Answer: $ \\boxed{\\frac{1}{e}} $
>
> ---
>
> **Step 1**
>
> Decide whether to retrieve: Output: `` `<retrieval>` ``
>
> Retrieval Query: Given that the line $ y = kx $ is a tangent to the function $ y = \\ln x $, find the value of $ k $.
>
> Retrieved Example:
>     ''Problem'': ''If the line $ y = kx + b $ is a tangent line to the curve $ y = \\ln x + 2 $ and also a tangent line to the curve $ y = \\ln (x + 1) $, find the value of $ b $.''
>     ''Step 1'': ''[CONDITION] Tangent points are $ (x_1, kx_1 + b) $ for $ y = \\ln x +2 $ and $ (x_2, kx_2 + b) $ for $ y = \\ln (x + 1) $ [PROCESS] Set up the relationship between the tangent points: $ k = \\frac{1}{x_1} = \\frac{1}{x_2 + 1} $'' [CONCLUSION] $ x_1 = x_2 + 1 $
>
> Inference Output:
>     ''Step 1'':'' [CONDITION] Line $ y = kx $ is tangent to $ y = \\ln x $ [PROCESS] Set $ kx = \\ln x $ for tangency [CONCLUSION] $ kx = \\ln x $''
>
> ---
>
> **Step 2**
>
> Decide whether to retrieve: Output: `` `<no-retrieval>` ``
>
> Retrieval Query: None
>
> Retrieved Example: None
>
> Inference Output:
>    ''Step 2'': ''[CONDITION] $ kx = \\ln x $ [PROCESS] Differentiate $ y = \\ln x $ to find slope: $ \\frac{dy}{dx} = \\frac{1}{x} $ [CONCLUSION] Slope of tangent at point of tangency is $ \\frac{1}{x} $''
>
> ---
>
> **Step 3**
>
> Decide whether to retrieve: Output: `` `<retrieval>` ``
> Retrieval Query:
> Given that the line $ y=kx $ is a tangent to the function $ y=\\ln x $, find the value of $ k $. Step 2: [CONDITION] $ kx = \\ln x $ [PROCESS] Differentiate $ y = \\ln x $ to find slope: $ \\frac{dy}{dx} = \\frac{1}{x} $ [CONCLUSION] Slope of tangent at point of tangency is $ \\frac{1}{x} $
>
> Retrieved Example:
>    ''Problem'': ''Find the equation of the tangent line to the function $ y = \\frac{\\ln x}{x} + x $ at the point $ (1, 1) $.''
>     ''Step 2'': ''[CONDITION] $ f'(x) = \\frac{1 - \\ln x}{x^2} + 1 $ [PROCESS] Calculate $ f'(1) $: $ f'(1) = \\frac{1 - \\ln 1}{1^2} + 1 $ [CONCLUSION] $ f'(1) = 2 $''
>     ''Step 3'': ''[CONDITION] Slope of tangent line $ k = 2 $, point $ (1, 1) $ [PROCESS] Use point-slope form: $ y - 1 = 2(x - 1) $ [CONCLUSION] $ y - 1 = 2x - 2 $''
>
> Inference Output:
>    ''Step 3: [CONDITION] Slope of tangent is $ \\frac{1}{x} $ [PROCESS] Set $ k = \\frac{1}{x} $ and substitute into $ kx = \\ln x $ [CONCLUSION] $ 1 = \\ln x $''
>
> ---
>
> **Step 4**
>
> Decide whether to retrieve: Output: `` `<no-retrieval>` ``
> Retrieval Query: None
> Retrieved Example: None
>
> Inference Output:
>      ''Step 4'': ''[CONDITION] $ 1 = \\ln x $ [PROCESS] Solve for $ x $ [CONCLUSION] $ x = e $.''
>
> ---
>
> **Step 5**
>
> Decide whether to retrieve: Output: `` `<no-retrieval>` ``
>
> Retrieval Query: None
>
> Retrieved Example: None
>
> Inference Output:
>     ''Step 5'': ''[CONDITION] $ x = e $ [PROCESS] Substitute $ x $ back into the equation $ kx = \\ln x $ [CONCLUSION] $ ke = \\ln e = 1 $.''
>
> ---
>
> **Step 6**
>
> Decide whether to retrieve: Output: `` `<no-retrieval>` ``
>
> Retrieval Query: None
>
> Retrieved Example: None
>
> Inference Output:
>     "Step 6": ''[CONDITION] $ ke = 1 $ [PROCESS] Solve for $ k $ [CONCLUSION] $ k = \\frac{1}{e} $''
>
> **Analysis**
>
> Case study that compares all five components side-by-side show our method's advantage.
>
> The failed trajectory (**The baseline's generated full solution**) primarily suffers from negative transfer caused by the retrieved complex "common tangent" example. In **Steps 1 to 3**, although the model successfully mimics the example to introduce tangent points and formulate equations, in **Steps 4 to 5**, influenced by the example's multi-variable nature, the model misinterprets the goal as deriving variable relationships rather than solving for specific values. Consequently, despite possessing key conditions, it fails to perform simultaneous elimination, halting at a parametric expression.
>
> In contrast, the successful trajectory of TSS benefits from its dynamic retrieval strategy. After utilizing the initial example to establish tangency conditions in **Step 1**, the model opts for `` `<no-retrieval>` `` in **Step 2**, effectively shielding itself from the complex example's interference and accurately calculating the derivative using internal knowledge. Crucially, **Step 3** retrieves a new, simpler example focused on "numerical calculation." This new cue timely corrects the reasoning direction, guiding the model to combine the derivative with the original equation to derive $ 1 = \ln x $, thereby breaking the logical deadlock and successfully solving for the final answer.

---

> ### Author Response · Authors · 2025-11-27
> **Official Comment by Authors**
>
> Dear Reviewer,
>
> Thank you very much for your time and expertise in reviewing our submission. As the rebuttal deadline is approaching, please let us know if you have any questions or need any clarification. We are very happy to discuss any aspect of the work.

---

### Official Review · Reviewer_5ynJ · 2025-11-03

**Soundness:** 3
**Presentation:** 3
**Contribution:** 3
**Rating:** 6
**Confidence:** 4

**Summary:**

This paper proposes a step-wise retrieval framework for large language model (LLM) reasoning, named Tailored Step Search (TSS). The method refines the traditional RAG paradigm from semantic-level retrieval to logic-step-level retrieval, and explicitly models each reasoning step’s condition, process, and conclusion through the structured representation $S_i = (C_i, P_i, Q_i)$. The retriever is trained via contrastive learning to capture *logical adjacency* rather than mere *semantic similarity*.

In addition, the authors design a two-stage generator consisting of (1) a retrieval-decision module (<retrieval> vs <no-retrieval>) and (2) a step-wise generation stage. The proposed framework achieves performance improvements on multiple mathematical reasoning datasets, including AIME, MATH500, and GSM8K. The paper further introduces new metrics—DSM, RCR, and TCN—to evaluate the quality of intermediate reasoning steps and the effectiveness of retrieval-based correction.

**Strengths:**

- High originality: This work is the first to introduce the concept of *logical adjacency* into a retrieval-based reasoning framework.
- Structured interpretability: The design of $S_i = (C_i, P_i, Q_i)$ makes each intermediate reasoning step more analyzable and interpretable.
- Broad experimental coverage: The method is evaluated across multiple reasoning datasets and model scales, with particularly strong results observed on the Qwen3-7B model.

**Weaknesses:**

1. Simplified modeling assumption:
    Reducing multi-step dependencies to a first-order Markov assumption may cause the model to lose contextual consistency over long reasoning chains.
2. Inconsistent training logic:
    Positive samples are constructed in a predictive form ($\rightarrow S_i$), while the retrieval stage uses a key–value form ($key \rightarrow value$). The paper should analyze the potential gap between these two paradigms.
3. Lack of complexity analysis:
    The paper does not report the average token length of $S_i$ or the token consumption during the generation phase, making it difficult to assess computational efficiency.
4. Limited statistical significance:
    The reported improvements are not particularly strong. Multiple-run statistics should be provided. Currently, the performance gains appear significant only on Qwen7B, while on other models the advantage over the second-best result is minimal and could fall within experimental variance.
5. Narrow baseline selection:
    The paper does not include comparisons with structurally similar systems such as AirRAG, Self-RAG, or ProcessRAG, which would help contextualize its contribution.

**Questions:**

- Does the construction of $S_i = (C_i, P_i, Q_i)$ include all previous steps?
 Please clarify whether each step explicitly contains the entire reasoning history or only the immediate predecessor $S_{i-1}$. This distinction directly affects token length, computational cost, and the validity of the first-order Markov assumption.

- Can the retriever embedding capture such logic-level relationships?
 The retriever is trained using a predictive form ($(X, S_{i-1}) \rightarrow S_i$); why was this approach chosen instead of a key–value alignment objective? If the retriever were trained directly on $(key \leftrightarrow query)$ matching, would the performance differ?

- How is *one-shot* defined in this context? Does it mean that the model retrieves examples only once during the entire reasoning process, or that it can perform multiple retrievals but uses a single example (one-shot) each time?

- Could the authors report the mean and variance over multiple runs, as well as statistical significance tests? This would clarify whether the observed improvements are consistent or within noise margins.

- Have the authors compared their method against structurally related logical retrieval approaches such as AirRAG, Self-RAG, or ProcessRAG? Such comparisons would better position TSS in the current research landscape.

- How does the proposed retrieval mechanism perform when combined with *Chain-of-Thought (CoT)* prompting? Does TSS still provide measurable improvement when CoT reasoning is already applied?

---

> ### Author Response · Authors · 2025-11-25
> **Response to Q1 & Q3:  Clarification on the steps of construction and the definition of k-shot**
>
> We thank you for this excellent question! Here we will clarify our steps of construction and the definition of k-shot.
>
> **1.Does the construction of $S_{i}=(C_{i}, P_{i}, Q_{i})$ include all previous steps?(Q1)**
>
> Let us clarify:
> - **$P_i$**: Does not contain historical steps; it is merely the **process** component of the current step.
> - **$Q_i$**: Represents the **conclusion** of the current step.
> - **$C_i$**: Encapsulates the conclusions from previous historical steps, serving as the **condition** for the current step.
>
> We can provide an example to help understand, as we have shown in Table 3 in Appendix in our paper:
>
> First we show raw data:
>
> Given sets $A=\\{x \mid x^2+2x-3<0\\}$ and $B=\\{x \mid |x-1| \geq 2\\}$, find $A \cap B$.
>
> Set A is defined by the inequality $x^2+2x-3<0$. We can factor the quadratic expression as $(x+3)(x-1) < 0$. The solution to this inequality lies between the roots $x=-3$ and $x=1$. Thus, set A is $A = \\{x \mid -3 < x < 1\\}$.
>
> Set B is defined by the absolute value inequality $|x-1| \geq 2$. This inequality can be split into two separate cases: $x-1 \geq 2$ or $x-1 \leq -2$. Solving for $x$ in each case, we get $x \geq 3$ or $x \leq -1$. Thus, set B is $B = \\{x \mid x \geq 3 \text{ or } x \leq -1\\}$.
>
> We are looking for the overlap between the interval $(-3, 1)$ and the region $(-\infty, -1] \cup [3, \infty)$. The only common region is where $-3 < x$ and $x \leq -1$. The intersection is $A \cap B = \\{x \mid -3 < x \leq -1\\}$.
>
> And then we show constructed data:
>
> Problem: Given sets $A=\\{x \mid x^2+2x-3<0\\}$ and $B=\\{x \mid |x-1| \geq 2\\}$, find $A \cap B$.
>
> Step 1:
>
> [CONDITION] Set $A=\\{x \mid x^2+2x-3<0\\}$.
>
> [PROCESS] Factor the quadratic: $x^2+2x-3=(x+3)(x-1)$, and find intervals where the product is negative.
>
> [CONCLUSION] $A=\\{x \mid -3<x<1\\}$.
>
> Step 2:
>
> [CONDITION] Set $B=\\{x \mid |x-1| \geq 2\\}$.
>
> [PROCESS] Solve the absolute value inequality: $x-1 \geq 2$ or $x-1 \leq -2$.
>
> [CONCLUSION] $B=\\{x \mid x \geq 3 \text{ or } x \leq -1\\}$.
>
> Step 3:
>
> [CONDITION] $A=\\{x \mid -3<x<1\\}$ and $B=\\{x \mid x \geq 3 \text{ or } x \leq -1\\}$.
>
> [PROCESS] Find the intersection $A \cap B$ by considering the intervals.
>
> [CONCLUSION] $A \cap B = \\{x \mid -3 < x \leq -1\\}$.
>
> As the example shown, the content following [CONDITION] of Step 3 contains the two conclusion of Step 1 and Step 2 as its condition.
>
> **2.How is one-shot defined in this context?(Q3)**
>
> Our definition of one-shot is as follows: For our step-level example framework (TSS), it implies that whenever retrieval is required, we retrieve and provide only one step-level example to guide the generation of the next step. The entire reasoning process may involve multiple instances of such single-example retrieval.
> The definition of k-shot refers to the fact that whenever retrieval is required, we retrieve and provide k step-level examples to guide the generation of the next step. The entire reasoning process may involve multiple instances of such k-example retrieval.
> For problem-level example methods, k-shot simply means providing k problem-level examples during inference.

---

> ### Author Response · Authors · 2025-11-25
> **Response to W2,Q2,& Q6: Clarification on training logic, retrieval logic, and reasoning methods**
>
> We thank you for this excellent question. We noticed that in A, you seem to be unclear about our training logic, retrieval logic, and reasoning methods. We will provide a more detailed explanation below.
>
> **1.Clarification on training logic and retrieval logic (W2 & Q2)**
>
> We need to clarify one thing: our knowledge base does not contain 'keys' and 'values' in the traditional sense. Instead, the knowledge base consists of complete 'context-successor' pairs $d^\prime = (X^\prime, S^\prime_{j-1}, S^\prime_j)$. Each entry itself contains the complete information unit needed to solve the problem. Perhaps our phrasing in the paper caused confusion. Our mention of keys and values in the paper was intended to explain why the required next step can be retrieved.
>
> Suppose our query $q = (X, S_{i-1})$ contains the problem and the previous step. Suppose the knowledge base contains $(X^\prime, S^\prime_{i-1}, S^\prime_i)$. We defined key = $(X^\prime, S^\prime_{i-1})$ and value = $S^\prime_i$ in the text to explain how our framework performs retrieval. When $q = (X, S_{i-1})$ is input into the retriever, the key = $(X^\prime, S^\prime_{i-1})$ similar to $q$ will be responded to, but there is no entry like $(X^\prime, S^\prime_{i-1})$ in the knowledge base, only the entry $(X^\prime, S^\prime_{i-1}, S^\prime_i)$, so $(X^\prime, S^\prime_{i-1}, S^\prime_i)$ is retrieved. Of course, this explanation might cause further misunderstanding, so please let us explain the logic behind it:
>
> We have two independent training samples:
>
> *   **Sample A:** Query: $c =$ (Problem A: ... $x^2 - 4 = 0$) and Positive: $d = (c, (x-2)(x+2)=0)$
> *   **Sample B:** Query: $c1 =$ (Problem B: ... $y^2 - 9 = 0$) and Positive: $d1 = (c1, (y-3)(y+3)=0)$
>
> Let $e(\cdot)$ represent the vector embedding. The model learns to pull $e(c)$ and $e(d)$ closer, and also pulls $e(c1)$ and $e(d1)$ closer. When $c$ and $c1$ are close, $c$ will be close to $d1$, and $c1$ will also be close to $d$.
>
> We can give an example to prove the necessity of our trained retriever:
>
> *   **Query Context** $c$: (Problem A: ... get $x^2 - 4 = 0$)
> *   **Candidate Context** $c1$: (Problem B: ... get $y^2 - 9 = 0$)
> *   **Candidate Context** $c2$: (Problem C: ... get $z^2 - 4z = 0$)
>
> A standard semantic encoder would likely consider $c$ and $c2$ more similar because they share more tokens ("$x^2$", "$-4$" vs "$z^2$", "$-4z$"). Both involve quadratic terms and constant/linear terms.
>
> It might retrieve $d2 = (c2, \text{next step is factoring out } z(z-4)=0)$. This is a wrong guide for solving $x^2 - 4 = 0$.
>
> However, our training objective is to maximize $\text{sim}(e(c), e(c, \text{next step is difference of squares } (x-2)(x+2)=0))$. Through such training, the model learns to focus on key structural patterns in the context. It will recognize that the core structure of $x^2 - 4 = 0$ is "squared term minus constant", and the "function" pointed to by this structure is applying the difference of squares formula. Therefore, the fine-tuned encoder will pull $e(c)$ and $e(c1)$ closer in space, because $y^2 - 9 = 0$ shares exactly the same "functional requirement". It will correctly retrieve $d1 = (c1, \text{next step is difference of squares } (y-3)(y+3)=0)$, providing the correct logical guidance for the original problem.
>
> **2. Why was our approach chosen instead of a key–value alignment objective? (Q2)**
>
> Because we want to find the appropriate next step, and key-value queries cannot retrieve the next step, while the model needs to access the next step. One relatively obvious approach is to first generate a step, then use that step to query and come back to modify it. However, this does not work because the correctness of the generated step is hard to guarantee. If the generated step is wrong, querying based on it will likely yield incorrect results; if it's correct, there is no need to query. If we directly train the retriever based on (key ↔ query) matching, the performance would be different from ours, because this is no longer the same task.
>
> **3. How does the proposed retrieval mechanism perform when combined with Chain-of-Thought (CoT) prompting? Does TSS still provide measurable improvement when CoT reasoning is already applied? (Q6)**
>
> Our approach is essentially an enhancement of CoT. The generative models in the baseline experiments (including our TSS) all adopt a step-by-step reasoning pattern similar to CoT. The advantage of TSS is that it can provide precise external guidance when the model's own CoT ability is insufficient or encounters a bottleneck. Our TSS exists as a supplement and enhancement to CoT. We have added the reasoning prompts in Appendix A.4 in the latest submitted paper.

---

> ### Author Response · Authors · 2025-11-25
> **Response to Q3: Supplementary data of  average token length (1-shot)**
>
> Thank you for these insightful questions. The approach you mentioned can provide a more comprehensive evaluation of our method. Some of these experiments we have already completed. The reason we did not include them in the text before is that, as the first context learning system focusing on future next-step examples, our focus was on example selection and performance improvement. Your suggestion is very valuable, and we have added the experimental data below. The input tokens of our method are significantly fewer than those of Method SPELL, and sometimes fewer than Method IDS. However, it is interesting that our `[CONDITION][PROCESS][CONCLUSION]` output structure not only does not increase the number of output tokens, but in many cases also produces slightly fewer tokens, making it more concise. Because of characters limitation, we report results on 1-shot, and we will report results subsequently.
>
> **1.Average token length during the report generation phase and comparison with the baseline.(W3)**
>
> **1-shot**
>
> | method    | model     | aime_output | aime_input | gsm8k_output | gsm8k_input | math500_output | math500_input | step_output | step_iutput |
> | --------- | --------- | ----------- | ---------- | ------------ | ----------- | -------------- | ------------- | ----------- | ----------- |
> | Vanilla   | llama2-7b | 614.99      | 866.18     | 322.52       | 552.45      | 419.34         | 638.42        | 336.58      | 748.53      |
> | SPELL     | llama2-7b | 658.36      | 16525.52   | 346.02       | 10897.77    | 490.37         | 12837.34      | 516.97      | 13932.07    |
> | IDS       | llama2-7b | 1834.65     | 3595.86    | 1254.37      | 2154.38     | 1647.28        | 2532.96       | 1298.38     | 2245.59     |
> | TSS(Ours) | llama2-7b | 743.14      | 5738.89    | 335.21       | 1976.92     | 545.76         | 3440.9        | 557.53      | 3788.22     |
> | Vanilla   | llama3-8b | 665.96      | 776.24     | 280.54       | 458.39      | 444.49         | 567.05        | 503.87      | 1054.69     |
> | SPELL     | llama3-8b | 735.46      | 14943.45   | 341.73       | 9097.23     | 522.68         | 11462.48      | 580.59      | 12648.33    |
> | IDS       | llama3-8b | 1935.46     | 3563.75    | 1046.47      | 5894.63     | 1764.39        | 1575.47       | 1564.73     | 3375.73     |
> | TSS(Ours) | llama3-8b | 732.35      | 5642.63    | 240.51       | 1631.18     | 447.61         | 3007.13       | 481.56      | 3858.99     |
> | Vanilla   | qwen-3b   | 888.45      | 1712.79    | 290.80       | 505.55      | 554.13         | 608.49        | 586.26      | 1248.78     |
> | SPELL     | qwen-3b   | 848.77      | 15464.91   | 288.05       | 9703.64     | 551.59         | 11901.73      | 600.04      | 13007.92    |
> | IDS       | qwen-3b   | 3464.36     | 5173.73    | 753.73       | 4864.26     | 1353.64        | 4773.37       | 483.32      | 4675.13     |
> | TSS(Ours) | qwen-3b   | 771.98      | 5707.93    | 228.53       | 1856.62     | 479.60         | 3002.87       | 557.05      | 4914.97     |
> | Vanilla   | qwen-7b   | 814.50      | 1637.84    | 304.56       | 809.12      | 532.32         | 607.49        | 590.32      | 661.52      |
> | SPELL     | qwen-7b   | 828.44      | 15531.96   | 306.21       | 9697.96     | 535.70         | 11916.37      | 565.16      | 13037.32    |
> | IDS       | qwen-7b   | 3457.22     | 5387.34    | 954.66       | 2598.55     | 1587.46        | 1893.68       | 2694.35     | 19646.74    |
> | TSS(Ours) | qwen-7b   | 707.83      | 5491.45    | 239.08       | 1542.43     | 468.43         | 3035.87       | 452.90      | 3775.52     |
>
> Output is the number of tokens generated by the model, and input is the number of tokens fed into the model. It can be seen that our method generally consumes fewer tokens in output compared to the Spell and IDS methods, and in many cases, even fewer than the Vanilla method. This indicates that our [condition][process][conclusion] structure does not lead to excessive token consumption. Regarding input tokens, although we have more than the Vanilla method, we still use fewer tokens than the Spell and IDS methods in most cases.

---

> ### Author Response · Authors · 2025-11-25
> **Response to Q3: Supplementary data of average token length (2-shot)**
>
> Now we provide average token length (2-shot):
>
> **2-shot**
>
> | method    | model     | aime_output | aime_input | gsm8k_output | gsm8k_input | math500_output | math500_input | step_output | step_iutput |
> | --------- | --------- | ----------- | ---------- | ------------ | ----------- | -------------- | ------------- | ----------- | ----------- |
> | Vanilla   | llama2-7b | 621.90      | 1571.16    | 347.43       | 989.55      | 456.58         | 1150.07       | 347.43      | 989.55      |
> | SPELL     | llama2-7b | 634.86      | 17087.90   | 344.56       | 11364.71    | 472.46         | 13345.99      | 522.29      | 14478.97    |
> | IDS       | llama2-7b | 1854.38     | 4673.95    | 1035.76      | 3024.37     | 1356.58        | 3752.83       | 1434.53     | 2093.88     |
> | TSS(Ours) | llama2-7b | 835.20      | 7822.24    | 344.42       | 2127.55     | 596.41         | 4667.35       | 492.85      | 3563.48     |
> | Vanilla   | llama3-8b | 691.20      | 1404.43    | 278.56       | 815.99      | 460.04         | 1020.43       | 460.03      | 1020.43     |
> | SPELL     | llama3-8b | 712.22      | 15471.05   | 312.95       | 9555.87     | 499.73         | 11955.1       | 563.32      | 13188.91    |
> | IDS       | llama3-8b | 1834.64     | 2254.64    | 1035.46      | 6795.45     | 1674.74        | 1476.23       | 1365.35     | 3046.56     |
> | TSS(Ours) | llama3-8b | 621.73      | 6245.19    | 242.62       | 1759.86     | 466.49         | 3801.58       | 436.97      | 4267.75     |
> | Vanilla   | qwen-3b   | 832.53      | 1479.85    | 279.73       | 890.12      | 546.89         | 1079.96       | 594.64      | 1175.57     |
> | SPELL     | qwen-3b   | 860.25      | 15975.17   | 288.19       | 10193.21    | 553.91         | 12403.89      | 596.39      | 14130.79    |
> | IDS       | qwen-3b   | 2576.57     | 4354.67    | 676.74       | 5365.75     | 1654.45        | 3254.62       | 1764.36     | 3537.93     |
> | TSS(Ours) | qwen-3b   | 795.92      | 7302.85    | 258.47       | 2078.40     | 479.59         | 3604.95       | 457.05      | 4998.46     |
> | Vanilla   | qwen-7b   | 759.53      | 1399.16    | 306.59       | 888.55      | 528.62         | 1078.96       | 565.72      | 1174.57     |
> | SPELL     | qwen-7b   | 860.25      | 15975.17   | 288.19       | 10193.21    | 553.91         | 12403.89      | 596.39      | 14130.79    |
> | IDS       | qwen-7b   | 2883.24     | 3546.63    | 10375.58     | 2874.94     | 1754.78        | 3365.25       | 1764.76     | 3556.62     |
> | TSS(Ours) | qwen-7b   | 673.59      | 6827.57    | 238.31       | 1570.39     | 458.62         | 3648.779      | 423.71      | 3356.66     |

---

> ### Author Response · Authors · 2025-11-25
> **Response to Q3: Supplementary data of average token length (3-shot)**
>
> Now we provide average token length (3-shot):
>
> **3-shot**
>
> | method    | model     | aime_output | aime_input | gsm8k_output | gsm8k_input | math500_output | math500_input | step_output | step_iutput |
> | --------- | --------- | ----------- | ---------- | ------------ | ----------- | -------------- | ------------- | ----------- | ----------- |
> | Vanilla   | llama2-7b | 568.59      | 2232.44    | 342.36       | 1429.59     | 452.69         | 1645.84       | 452.69      | 1645.83     |
> | SPELL     | llama2-7b | 648.31      | 17671.40   | 319.40       | 11813.67    | 457.686        | 13843.73      | 521.45      | 15009.88    |
> | IDS       | llama2-7b | 146.38      | 6949.78    | 783.93       | 4388.77     | 1357.97        | 5027.49       | 1384.54     | 4994.39     |
> | TSS(Ours) | llama2-7b | 802.02      | 9142.50    | 332.27       | 2118.93     | 578.59         | 5426.03       | 511.34      | 3959.45     |
> | Vanilla   | llama3-8b | 691.16      | 1997.41    | 271.87       | 1177.07     | 452.18         | 1459.28       | 452.18      | 1459.28     |
> | SPELL     | llama3-8b | 693.92      | 15992.24   | 308.83       | 9992.26     | 506.18         | 12938.05      | 525.07      | 13669.58    |
> | IDS       | llama3-8b | 1965.74     | 5987.89    | 876.92       | 3365.36     | 1354.35        | 4365.73       | 1356.87     | 4563.62     |
> | TSS(Ours) | llama3-8b | 637.83      | 7685.86    | 240.53       | 1856.59     | 471.89         | 4648.93       | 434.06      | 5261.31     |
> | Vanilla   | qwen-3b   | 851.59      | 2092.87    | 273.04       | 1277.39     | 530.97         | 1536.64       | 598.35      | 1662.07     |
> | SPELL     | qwen-3b   | 869.82      | 16571.19   | 283.12       | 10661.75    | 546.97         | 12914.37      | 585.33      | 14052.52    |
> | IDS       | qwen-3b   | 2564.73     | 6343.66    | 878.96       | 3675.39     | 1654.62        | 4745.19       | 1865.84     | 5064.88     |
> | TSS(Ours) | qwen-3b   | 771.90      | 8707.93    | 344.95       | 3386.08     | 493.62         | 4737.53       | 537.30      | 5342.75     |
> | Vanilla   | qwen-7b   | 830.1       | 2049.6     | 304.96       | 1580.31     | 526.77         | 1535.64       | 576.70      | 1661.07     |
> | SPELL     | qwen-7b   | 786.06      | 16644.84   | 303.87       | 10658.94    | 526.63         | 12971.09      | 570.25      | 14082.76    |
> | IDS       | qwen-7b   | 3468.57     | 6436.25    | 912.15       | 4682.26     | 1578.33        | 4563.25       | 1674.46     | 5064.42     |
> | TSS(Ours) | qwen-7b   | 686.48      | 8706.33    | 237.47       | 1600.82     | 459.15         | 4316.45       | 416.41      | 4970.45     |

---

> ### Author Response · Authors · 2025-11-25
> **Response to Q3: Supplementary data of average token length (4-shot)**
>
> Thanks for your patience! Now we provide average token length (4-shot):
>
> **4-shot**
>
> | method    | model     | aime_output | aime_input | gsm8k_output | gsm8k_input | math500_output | math500_input | step_output | step_iutput |
> | --------- | --------- | ----------- | ---------- | ------------ | ----------- | -------------- | ------------- | ----------- | ----------- |
> | Vanilla   | llama2-7b | 568.60      | 2921.02    | 342.37       | 1868.42     | 452.69         | 2164.69       | 482.45      | 2335.61     |
> | SPELL     | llama2-7b | 628.26      | 18257.76   | 322.56       | 12593.96    | 454.76         | 14342.77      | 508.98      | 15535.92    |
> | IDS       | llama2-7b | 1694.94     | 7392.93    | 1098.26      | 5395.18     | 1493.36        | 6373.62       | 1495.65     | 7062.68     |
> | TSS(Ours) | llama2-7b | 859.80      | 11979.99   | 337.05       | 2242.61     | 578.16         | 5956.66       | 476.76      | 4071.24     |
> | Vanilla   | llama3-8b | 691.16      | 2612.07    | 271.87       | 1537.63     | 452.18         | 1919.19       | 452.18      | 1919.19     |
> | SPELL     | llama3-8b | 730.79      | 16547.68   | 297.99       | 10418.43    | 495.13         | 12918.73      | 534.29      | 14173.43    |
> | IDS       | llama3-8b | 1938.27     | 7327.38    | 963.85       | 7474.38     | 1675.82        | 5938.73       | 1364.73     | 5974.83     |
> | TSS(Ours) | llama3-8b | 655.19      | 8959.95    | 241.87       | 1966.41     | 463.48         | 5101.51       | 477.80      | 6801.65     |
> | Vanilla   | qwen-3b   | 851.59      | 2733.95    | 273.03       | 1663.90     | 530.97         | 2016.27       | 598.35      | 2183.59     |
> | SPELL     | qwen-3b   | 875.01      | 17129.89   | 286.04       | 11125.77    | 546.41         | 13423.23      | 609.61      | 14554.55    |
> | IDS       | qwen-3b   | 2574.27     | 9357.83    | 783.53       | 4973.28     | 1595.27        | 3826.27       | 1684.83     | 6472.73     |
> | TSS(Ours) | qwen-3b   | 757.88      | 11603.93   | 246.55       | 4462.43     | 484.35         | 5624.36       | 563.13      | 5914.98     |
> | Vanilla   | qwen-7b   | 830.10      | 2647.33    | 304.96       | 1662.75     | 526.77         | 2015.27       | 576.70      | 2182.59     |
> | SPELL     | qwen-7b   | 775.75      | 17236.22   | 304.76       | 11124.87    | 526.69         | 13472.44      | 569.13      | 14569.82    |
> | IDS       | qwen-7b   | 2445.34     | 8375.48    | 1067.19      | 5234.92     | 1636.28        | 6591.26       | 1593.68     | 6385.93     |
> | TSS(Ours) | qwen-7b   | 708.55      | 10734.20   | 242.57       | 1665.09     | 457.14         | 5000.19       | 432.22      | 5856.34     |

---

> ### Author Response · Authors · 2025-11-25
> **Response to W4, W5, Q4 & Q5: Report the mean and standard deviation of multiple runs and supplement with comparison experiments with Self-RAG**
>
> Thanks for your excellent question. We did not previously report the results of Self-RAG[1] in the paper because Self-RAG[1] is trained on QA data. Now we report it below.
>
> **1.Results from multiple runs (mean ± standard deviation) （W4&Q4）**
> We understand your perspective. Our results are based on selecting the most stable data after multiple runs. As per your request, we have updated the data in Table 2 of the latest submitted paper to reflect the results of multiple runs (mean ± standard deviation).
>
> **2.Supplementary Self-RAG experiments（W5&Q5）**
>
> Self-RAG is a framework trained to carry out selective retrieval, reasoning, and evaluation processes. It uses retrieval tokens and multiple evaluation tokens for reasoning and judgment.
>
> We will report the results on AIME,GSM8K,MATH500 and STEP on 1-shot, 2-shot, 3-shot and 4-shot using Llama-2-7B and Llama2-13B as used in self-RAG.  The accuracy is measured on a percentage scale, precise to two decimal places. To make it easier for you to view, we have also recorded the results of our method here for easy comparison.
>
> **On AIME**
>
> | Base Model | Method   | K-Shot | Accuracy% |
> | ---------- | -------- | ------ | --------- |
> | Llama2-7B  | SELF-RAG | 1-shot | 0.82      |
> | Llama2-7B  | SELF-RAG | 2-shot | 1.63      |
> | Llama2-7B  | SELF-RAG | 3-shot | 0.82      |
> | Llama2-7B  | SELF-RAG | 4-shot | 0.82      |
> | Llama2-7B  | TSS      | 1-shot | 2.46      |
> | Llama2-7B  | TSS      | 2-shot | 2.46      |
> | Llama2-7B  | TSS      | 3-shot | 1.63      |
> | Llama2-7B  | TSS      | 4-shot | 1.63      |
> | Llama2-13B | SELF-RAG | 1-shot | 1.63      |
> | Llama2-13B | SELF-RAG | 2-shot | 2.46      |
> | Llama2-13B | SELF-RAG | 3-shot | 2.46      |
> | Llama2-13B | SELF-RAG | 4-shot | 1.63      |
> | Llama2-13B | TSS      | 1-shot | 4.09      |
> | Llama2-13B | TSS      | 2-shot | 3.28      |
> | Llama2-13B | TSS      | 3-shot | 2.46      |
> | Llama2-13B | TSS      | 4-shot | 3.28      |
>
> **On GSM8K**
>
> | Base Model | Method   | K-Shot | Accuracy% |
> | ---------- | -------- | ------ | --------- |
> | Llama2-7B  | SELF-RAG | 1-shot | 15.06     |
> | Llama2-7B  | SELF-RAG | 2-shot | 15.78     |
> | Llama2-7B  | SELF-RAG | 3-shot | 15.78     |
> | Llama2-7B  | SELF-RAG | 4-shot | 16.63     |
> | Llama2-7B  | TSS      | 1-shot | 18.18     |
> | Llama2-7B  | TSS      | 2-shot | 19.31     |
> | Llama2-7B  | TSS      | 3-shot | 20.65     |
> | Llama2-7B  | TSS      | 4-shot | 18.89     |
> | Llama2-13B | SELF-RAG | 1-shot | 19.81     |
> | Llama2-13B | SELF-RAG | 2-shot | 20.88     |
> | Llama2-13B | SELF-RAG | 3-shot | 20.74     |
> | Llama2-13B | SELF-RAG | 4-shot | 20.95     |
> | Llama2-13B | TSS      | 1-shot | 30.37     |
> | Llama2-13B | TSS      | 2-shot | 31.22     |
> | Llama2-13B | TSS      | 3-shot | 31.15     |
> | Llama2-13B | TSS      | 4-shot | 30.30     |
>
> **On MATH500**
>
> | Base Model | Method   | K-Shot | Accuracy% |
> | ---------- | -------- | ------ | --------- |
> | Llama2-7B  | SELF-RAG | 1-shot | 3.80      |
> | Llama2-7B  | SELF-RAG | 2-shot | 3.20      |
> | Llama2-7B  | SELF-RAG | 3-shot | 2.60      |
> | Llama2-7B  | SELF-RAG | 4-shot | 3.00      |
> | Llama2-7B  | TSS      | 1-shot | 5.00      |
> | Llama2-7B  | TSS      | 2-shot | 6.20      |
> | Llama2-7B  | TSS      | 3-shot | 4.80      |
> | Llama2-7B  | TSS      | 4-shot | 4.20      |
> | Llama2-13B | SELF-RAG | 1-shot | 5.60      |
> | Llama2-13B | SELF-RAG | 2-shot | 5.40      |
> | Llama2-13B | SELF-RAG | 3-shot | 5.20      |
> | Llama2-13B | SELF-RAG | 4-shot | 5.10      |
> | Llama2-13B | TSS      | 1-shot | 6.40      |
> | Llama2-13B | TSS      | 2-shot | 7.00      |
> | Llama2-13B | TSS      | 3-shot | 6.40      |
> | Llama2-13B | TSS      | 4-shot | 6.80      |
>
> **On our STEP Dataset**
>
> | Base Model | Method   | K-Shot | Accuracy% |
> | ---------- | -------- | ------ | --------- |
> | Llama2-7B  | SELF-RAG | 1-shot | 8.25      |
> | Llama2-7B  | SELF-RAG | 2-shot | 9.28      |
> | Llama2-7B  | SELF-RAG | 3-shot | 9.28      |
> | Llama2-7B  | SELF-RAG | 4-shot | 10.31     |
> | Llama2-7B  | TSS      | 1-shot | 15.46     |
> | Llama2-7B  | TSS      | 2-shot | 11.34     |
> | Llama2-7B  | TSS      | 3-shot | 13.40     |
> | Llama2-7B  | TSS      | 4-shot | 17.53     |
> | Llama2-13B | SELF-RAG | 1-shot | 9.28      |
> | Llama2-13B | SELF-RAG | 2-shot | 9.28      |
> | Llama2-13B | SELF-RAG | 3-shot | 10.31     |
> | Llama2-13B | SELF-RAG | 4-shot | 9.28      |
> | Llama2-13B | TSS      | 1-shot | 14.43     |
> | Llama2-13B | TSS      | 2-shot | 17.53     |
> | Llama2-13B | TSS      | 3-shot | 14.43     |
> | Llama2-13B | TSS      | 4-shot | 13.40     |
>
> In terms of results, our method outperforms self-rag in most settings. On GSM8K, Our frameworks are all significantly higher than self-RAG. On other dataset, our **TSS** also perform better, and our mechanism is designed to be simpler than self-RAG.
>
> [1] SELF-RAG: Learning to Retrieve, Generate and Critique through Self-reflection. ICLR 2024.

---

> ### Author Response · Authors · 2025-11-27
> **Response to W1: Clarification on the first-order Markov assumption**
>
> Thank you for your time and constructive feedback. Here we will address your question about whether simplifying the multi-step dependencies into a first-order Markov assumption might cause the model to lose contextual consistency in long reasoning chains.
>
> While we acknowledge this is a simplification, we argue that it serves as a distinct advantage in our framework.
>
> First, our `[CONDITION]` structure is explicitly designed to encapsulate all historical information necessary for solving the subsequent step. Crucially, this historical information does not merely repeat the entire historical reasoning trajectory; rather, it represents the known conclusions synthesized from the problem statement and prior reasoning steps.
>
> We utilize the `[CONDITION]` structure to condense relevant information within a single step, while the `[CONCLUSION]` component records the specific outcome of the current step. In the subsequent step, the model aggregates conclusions from the current and preceding steps to populate the new `[CONDITION]`.
>
> This mechanism can be understood as follows: `[CONDITION]` serves as a review of existing conditions and accumulated conclusions, while `[CONCLUSION]` is the output of the current step's reasoning. Combined with the reasoning in `[PROCESS]`, the model completes a coherent chain:
> `[CONDITION] -> [PROCESS] -> [CONCLUSION] -> [CONDITION] -> [PROCESS] -> [CONCLUSION] ...`
>
> This mirrors human problem-solving behavior: summarizing known information at the beginning of each step, performing reasoning, drawing a conclusion, and then repeating this cycle. Our rationale for adopting this structure is threefold:
> 1.  **Cognitive Simulation:** To simulate the human cognitive process of "Review $\rightarrow$ Reason $\rightarrow$ Conclude".
> 2.  **Efficiency:** To reduce token consumption through a concise format, avoiding the overhead of re-processing the full context history for every step.
> 3.  **Global Context:** To enable the model to incorporate more comprehensive global information (via the summarized conditions) when retrieving suitable examples for the next step.
>
> Since we explicitly constructed this `[CONDITION]-[PROCESS]-[CONCLUSION]` structure within each step of our training data, the trained model naturally acquires the capability to summarize existing information, reason upon it, and derive conclusions.
>
> We can provide an example to show consistency in reasoning of model:
>
> **Problem:**
> Find all values of $x$ that satisfy the equation $x = \sqrt{11-2x} + 4$.
>
> **Generated Steps:**
>
> Step 1 [CONDITION] The equation is given as $x = \sqrt{11-2x} + 4$. [PROCESS] Subtract 4 from both sides to isolate the square root: $x - 4 = \sqrt{11-2x}$. [CONCLUSION] The equation simplifies to $x - 4 = \sqrt{11-2x}$.
>
> Step 2 [CONDITION] The equation is $x - 4 = \sqrt{11-2x}$. [PROCESS] Square both sides to eliminate the square root: $(x - 4)^2 = 11 - 2x$. [CONCLUSION] This leads to the equation $x^2 - 8x + 16 = 11 - 2x$.
>
> Step 3 [CONDITION] The equation is $x^2 - 8x + 16 = 11 - 2x$. [PROCESS] Rearrange the equation to one side: $x^2 - 6x + 5 = 0$. [CONCLUSION] The quadratic equation is $x^2 - 6x + 5 = 0$.
>
> Step 4 [CONDITION] The quadratic equation is $x^2 - 6x + 5 = 0$. [PROCESS] Factor the quadratic: $(x - 1)(x - 5) = 0$. [CONCLUSION] The solutions are $x = 1$ and $x = 5$.
>
> Step 5 [CONDITION] The potential solutions are $x = 1$ and $x = 5$. [PROCESS] Check $x = 1$: $1 = \sqrt{11 - 2 \cdot 1} + 4 \implies 1 = \sqrt{9} + 4 \implies 1 = 3 + 4$, which is false. [CONCLUSION] $x = 1$ is not a solution.
>
> Step 6 [CONDITION] The potential solution is $x = 5$. [PROCESS] Check $x = 5$: $5 = \sqrt{11 - 2 \cdot 5} + 4 \implies 5 = \sqrt{1} + 4 \implies 5 = 1 + 4$, which is true. [CONCLUSION] $x = 5$ is a solution.
>
> Step 7 [CONDITION] The only solution found is $x = 5$. [PROCESS] Conclude that there are no extraneous solutions. [CONCLUSION] The only solution is $\boxed{5}$.
>
> In our reasoning, the model reviews the content from previous steps that can serve as prerequisites for the current step within the condition. From our example, the model is able to maintain consistency during reasoning.
>
> In Table 2 of the article, TSS w/o retrieval verifies that our structure does not affect inference performance. As our ablation experiments show, when we use our trained generator to perform inference in scenarios without retrieval, the performance does not decrease significantly, and sometimes even shows a slight improvement. We completely understand your concern that in cases with very long reasoning chains of more than forty or fifty steps, this might not have been considered in our current work. The difficulty of the problems in our dataset keeps the reasoning length within about ten to twenty steps, which is consistent with the length maintained by most mathematics datasets. We have already added the issue of very long reasoning chains as a limitation in the conclusion section of the article. Thank you for reading!

---

> ### Author Response · Authors · 2025-11-27
> **Official Comment by Authors**
>
> Dear Reviewer,
>
> Thank you very much for your time and expertise in reviewing our submission. As the rebuttal deadline is approaching, please let us know if you have any questions or need any clarification. We are very happy to discuss any aspect of the work.

---

### Author Response · Authors · 2025-11-30
**Summary of Rebuttal Phase & Reviewer Consensus**

Dear Program Chairs, Senior Area Chairs, and Area Chair,

We represent our sincere gratitude to all reviewers for their insightful feedback and the time dedicated to reviewing our work. We provide this summary of the rebuttal phase to conclude the substantial improvements made to the paper based on their insightful suggestions.

### 1. Consensus on Strengths
We are highly encouraged that the reviewers recognized the novelty and significance of our work:

* Motivation: **Reviewer 5ynJ** praised the work for being the  **first to introduce the concept of logical adjacency**. **Reviewer CgcP** recognized it as identifying a **clear and significant limitation** in traditional problem-level retrieval. **Reviewer CgcP** highlighted the retrieval focus shifts from **what the problem looks like** to **what the next logical step is** was deemed **highly logical and necessary**. **Reviewer Bzfe** commended it identifying **a real limitation of problem-level retrieval in mathematical reasoning - many retrieved steps are misaligned**.
* Methodology & Interpretability: The `[CONDITION][PROCESS][CONCLUSION]` structure was recognized for making reasoning **more analyzable and interpretable** (**Reviewers 5ynJ and Bzfe**).  **Reviewer CgcP** described  the training methodology as **clever, effective and sophisticated**.
* Evaluation & Experimentation: **Reviewer CgcP** commended our **robust, process-oriented evaluation** as a **major strength**. **Reviewers Bzfe** noted our fine-grained evaluation as **intermediate reasoning quality beyond final answer accuracy**. **Reviewers 5ynJ , Bzfe and Fy6i** also acknowledged broad experimental coverage across multiple datasets.

### 2. Revisions & Responses
We have conducted extensive additional experiments and analyses to address the raised concerns comprehensively:

**Comprehensive Benchmarking & Baselines**

* Comparison with Self-RAG (**Reviewer 5ynJ (W5,Q5) , Reviewer Bzfe (W1,Q1)**): We added a direct comparison with Self-RAG on AIME, GSM8K, MATH500, and STEP. Results show **TSS** significantly outperforms Self-RAG.
* New Benchmarks (**Reviewer Bzfe (W2)**): We have expanded evaluation to **MMLU-College**, **MMLU-High School**, and **TheoremQA**. **TSS** outperforms baselines most of the time, demonstrating strong generalization.
* Statistical Significance (**Reviewer 5ynJ (W4,Q4), Reviewer CgcP (W3)**): Initially we use the most stable data after multiple runs. We updated Table 2 to report the **mean and standard deviation** across multiple runs.

**Efficiency, Cost & Applicability**

To address concerns about computational cost and applicability:
* Token Efficiency Analysis (**Reviewer 5ynJ (W3), Reviewer Fy6i (Q2)**): We provided a detailed table of average token length (1-shot to 4-shot). Data shows that our method consumes fewer output tokens than SPELL and IDS, proving computationally efficient our structured approach.
* Applicability to Frozen Models and Generalization to Other Tasks (**Reviewer CgcP (W1,Q1), Reviewer Fy6i (Q1)**): We discussed feasible strategies for applying **TSS** to frozen models and expansion to domains leveraging the same logical adjacency principle, like code generation and complex QA .

**Quality Evaluation**

* Retrieval Quality and relevant Metrics (**Reviewer CgcP (W2,Q3), Reviewer Bzfe (W3)**): We conduct an indirect analysis and a direct quality impact analysis. Results show a strong correlation between retrieval quality and accuracy.
* Prediction Accuracy of special tokens (**Reviewer Bzfe (Q3), Reviewer CgcP (W2)**): We evaluated the generator's accuracy in predicting special tokens, showing the model effectively learned when to retrieve.
* Sensitivity Analysis and automatic annotation quality (**Reviewer Bzfe (Q4,Q5)**): We analyzed the evaluation of automatic annotation quality and impact of the number of negative examples (8,16,32,64).

**Methodological Clarifications & Presentation**
* Clarification on Knowledge Base & Markov Assumption (**Reviewer 5ynJ (W1,W2,Q1,Q2), Reviewer Bzfe (Q2)**): We clarified that our `[CONDITION]` contains necessary historical context and explained the specific construction of the Knowledge Base.
* Qualitative Case Study (Responding to **Reviewer CgcP (Q2)**): We added five kinds of cases visualizing TSS vs. Baseline. It clearly demonstrates how TSS uses retrieved steps to correct reasoning.
*Clarification of definition and method positioning (**Reviewer 5ynJ (Q3,Q6)**):  We explained our definition of  'k-shot', and our approach is essentially an enhancement of CoT.
* Presentation Improvements (**Reviewer CgcP (W4), Reviewer Fy6i (W1,W2,W3)**): We fixed formats and line break issues, re-scaled the plots, clarified Figure 1's caption (color meanings), and added introductions for SPELL/IDS baselines.

We answer all the weakness and questions of reviewers. We believe these revisions fully address the reviewers' concerns while reinforcing the strengths of our work. We look forward to further discussion.

---

### Meta-Review · Area_Chair_fEFL · 2025-12-24

**Summary:**

This paper proposes TSS (Tailored Step Search), a framework for guiding LLM reasoning by retrieving specific "next steps" rather than full problem examples. The method involves constructing a fine-grained dataset (Condition-Process-Conclusion), fine-tuning a Step Retriever, and fine-tuning a Generator to trigger retrieval tokens.

While the premise of addressing granularity mismatch in RAG is sound and empirical results on math benchmarks are positive, the AC’s recommendation is Reject. The primary rationale is the complexity relative to the utility over existing methods. Unlike "plug-and-play" RAG or inference-time strategies, TSS requires a heavy, multi-stage training pipeline and significantly raises the barrier to adoption. The reviewers remained concerned that the performance gains do not sufficiently justify this operational cost compared to simpler, stronger baselines like Self-RAG or iterative prompting.

**Reviewer Concerns:**

**Addressed Concerns:**
- Missing Baselines: The authors successfully added Self-RAG comparisons during the rebuttal, showing TSS often performs better in the specific setups tested.
- Statistical Significance: The authors updated tables to include mean and standard deviation, addressing concerns about result stability on small datasets like AIME.

**Outstanding Concerns:**
- Implementation Cost & Usability (Reviewer CgcP): This was the most damaging critique and remains unresolved. The reviewer noted that the method is not a "plug-and-play" solution and cannot be used with powerful API-based models. The rebuttal confirmed that fine-tuning is necessary for the "retrieval trigger" to work effectively. This limitation drastically reduces the paper's impact compared to inference-time algorithms.
- Complexity vs. Gain (Reviewer Bzfe): While the method beats baselines, the margin of improvement arguably does not justify the complexity of maintaining a vector database of steps (increasing retrieval index size significantly) plus training two models.
- Metric Robustness (Reviewer CgcP): While clarified, the reliance on custom, LLM-based metrics (TCN, DSM) rather than established community standards makes it difficult to objectively verify the "reasoning quality" improvements claimed.

**Reviewer Scores:**

- Reviewer 5ynj (6 to 7): While the baselines were added and the reviewer's concern about the "first-order Markov assumption" was addressed in that the [condition] contains all historical steps. These may lead to a score increase from 6 to 7.
- Reviewer CgcP (4 to 4): The rebuttal failed to address the core objection: cost. The reviewer explicitly stated that the method's inability to work with frozen models is a major downside. The clarification on metrics was helpful, but the reviewer is unlikely to revise their score based on the clarification.
- Reviewer Bzfe (4 to 5): The addition of Self-RAG improved the paper's completeness, moving this reviewer to a borderline stance.
- Reviewer Fy6i (2 to 4): The rebuttal clarifies the core confusion, but starting from a "2" makes a jump to "Accept" difficult.

---

### Decision · Program_Chairs · 2026-01-26

Reject